# Future supply and demand of net primary production in the Sahel

Florian Sallaba[1], Stefan Olin[1], Kerstin Engström[1], Abdulhakim M. Abdi[1], Niklas Boke-Olén[1], Veiko Lehsten[1], Jonas Ardö[1], Jonathan W. Seaquist[1]

[1]Department of Physical Geography and Ecosystem Science, Lund University, Lund, 22362, Sweden

*Correspondence to*: Jonathan W. Seaquist (jonathan.seaquist@nateko.lu.se)

**Abstract**

In the 21[st] century, climate change in combination with increasing demand, mainly from population growth, will exert greater pressure on the ecosystems of the Sahel to supply food and feed resources. The balance between supply and demand, defined as the annual biomass required for human consumption, serves as a key metric for quantifying basic resource

shortfalls over broad regions.

Here we apply an exploratory modelling framework to analyze the variations in the timing and geography of different NPP (net primary production) supply-demand scenarios, with distinct assumptions determining supply and demand, for the 21[st] century Sahel. We achieve this by coupling a simple NPP supply model forced with projections from four representative concentration pathways, with a global, reduced-complexity demand model driven by socio-economic data and assumptions

derived from five shared socio-economic pathways.

For the scenario that deviates least from current socio-economic and climate trends, we find that per capita NPP begins to outstrip supply in the 2040s, while by 2050, half the countries in the Sahel experience NPP shortfalls. We also find that despite variations in the timing of the onset of NPP shortfalls, demand cannot consistently be met across the majority of scenarios. Moreover, large between-country variations are shown across the scenarios where by the year 2050, some

countries consistently experience shortage or surplus, while others shift from surplus to shortage. At the local level (i.e. grid cell) hotspots of total NPP shortfall consistently occur in the same locations across all scenarios, but vary in size and magnitude. These hotspots are linked to population density and high demand. For all scenarios, total simulated NPP supply doubles by 2050 but is outpaced by increasing demand due to a combination of population growth and adoption of diets rich in animal products. Finally, variations in the timing of onset and end of supply shortfalls stem from the assumptions that

underpin the shared socio-economic pathways rather than the representative concentration pathways.

Our results suggest that the UN sustainable development goals for eradicating hunger are at high risk for failure. This emphasizes the importance of policy interventions such as the implementation of sustainable and healthy diets, family planning, reducing yield gaps, and encouraging transfer of resources to impoverished areas via trade relations.

## 1 Introduction

The global demand for food is projected to increase by up to double by 2050 (compared to the year 2005) due to rapid population growth and changes in dietary preferences (Hertel, 2015; Tilman et al., 2011). As a consequence, global agricultural supply needs to increase substantially in order to satisfy this demand (Ray et al., 2013). Agricultural practices can be intensified with technological investments (i.e. mechanization, irrigation and fertilization) to increase yields but these are costly and often lead to environmental degradation (Foley et al., 2005). As opposed to agricultural intensification, the amount of agricultural land can be expanded in order to meet future demand. This results in changing land use and land cover (LULCC), for example from natural vegetation to cropland. Approximately 35% of the total ice-free land surface is used for agriculture (Ramankutty et al., 2008). Agricultural land (grassland and cropland) expanded by 3% globally between 1985 and 2005 and is expected to further increase, especially in the tropics (Foley et al., 2011). The production of the most common crops (e.g. cereals, oil crops, and vegetables) increased by nearly 80% over the past four decades (FAOSTAT, 2015; Foley et al., 2011), mostly due to increases in yield (Kastner et al., 2012) and to a smaller extent by LULCC (Foley et al., 2011). Despite the large increase in agricultural production, global food security is not ensured (due to access and distribution challenges (e.g. Brown, 2016; Pinstrup-Andersen, 2009)), as there are presently 792 million people chronically undernourished across the planet, a third of which are in Africa (FAOSTAT, 2015).

The Sahel region of sub-Saharan Africa is one of the most technologically underdeveloped regions in the world, where yield gaps are explained by low and variable rainfall combined with low soil fertility (Yengoh and Ardö, 2014). The population by-and-large relies on rain-fed farming practices including subsistence agriculture, cash crops, pastoralism and agro-pastoralism. The population has a high reliance on their own land, where 95% of food produce is for domestic consumption, (Abdi et al., 2014; Running, 2014). The vulnerability of the population to variations in agricultural supply due to frequent drought undermines wealth accumulation, which would otherwise provide a buffer in drought years (Barbier et al., 2009). Additionally, poor transportation infrastructure inhibit the trade and distribution of food resources (Olsson, 1993). Between the late-1960s to the early 1990s, the Sahel experienced a protracted dry period in which severe droughts caused fluctuating levels of food supply leading, in some cases, to severe humanitarian crises. The devastating droughts in 1972/73 and 1983/84 induced complete crop failure leading to the largest famines in the recent history of the Sahel (Ibrahim, 1988). The latest major drought to hit the region was in 2002. As of 2013, over 11 million people across the region were considered to be food insecure (United Nations, 2013).

NPP estimates from the MODIS (Moderate Resolution Imaging Spectroradiometer (MODIS) suggest that the Sahel region experienced a near-constant rate of crop productivity between 2000 and 2010, while population grew at a rate of 3.1% over the same period (Abdi et al., 2014). Abdi et al. (2014) also showed that 19% of the NPP supply in the Sahel was able to satisfy demand for the year 2000 but this increased to 41% in 2010 due to a 31% increase in the population. Since the NPP demand increased at an annual rate of 2.2% over the period while the supply was near constant, the near doubling in NPP demand implies, in relative terms, that there was less NPP supply to service the increase in population. This raises the

question of whether demand could consistently outstrip supply in the future and underscores the importance for developing tools for analyzing potential future supply and demand that could be of use for policy makers. Indeed, the balance between supply and demand (annual biomass required for human consumption) serves as a key metric for quantifying basic resource shortfalls over broad regions (Abdi et al., 2014; Running, 2014).

Developing such tools requires coupling of specific models that address different sectors, such as a model for supply and a model for demand that can be run across multiple future climate, socio-economic and $CO_2$ concentration scenarios. However, the supply-demand system in the Sahel is complex and the future cannot be precisely evaluated. This is because there are many uncertainties associated with the assumptions that underpin the natural and socioeconomic drivers that lead to particular supply-demand balances. As such, an exploratory modelling approach is required, where an emphasis is placed on

a structured analysis across a range of outcomes. This approach capitalizes on future indeterminacy for developing adaptive policy insights (e.g. Kwakkel and Pruyt (2013)). As the goal of exploratory frameworks is not prediction, they often employ parsimonious or simplified versions of more complex models (often referred to as meta-models in the latter case) that run across a range of scenarios (e.g. Harrison et al. (2016)). Another benefit of using such simplified models lies in the ease to which they can be coupled to other sectoral models (e.g. Kebede et al. (2015)).

In this study we couple a simple supply model (Biome-based Meta-model Ensemble - BME) with a demand model (Parsimonious Land Use Model - PLUM) to compute NPP supply-demand balance for a set of 21st century Sahel scenarios covering different climate, $[CO_2]$ and socio-economic trajectories in an exploratory modelling framework. Our overall aim is to quantify variations in the timing and geography of NPP supply and demand in the Sahel in association with these trajectories. Three different aggregation levels are considered, including Sahel, the national, and –local (cell level with a

spatial resolution of $0.5^o$ x $0.5^o$). Thereafter we discuss those natural and socio-economic factors that lead to changes in the balance between supply and demand throughout the $21^{st}$ century, as accounted for by the coupled models. The Sahel-level analysis focuses on the total impact of the different future climatic and socio-economic pathways and its timing on supply and demand and asks the fundamental question of whether the Sahel as a whole, could potentially be self-sufficient. By contrast, the country-level analysis focusses on a level relevant for policy, international relations, and aid agencies. Finally,

the local-level analysis identifies potential hotspots of supply shortage occurring at sub-national levels. We restrict our analyses to localized supply-demand only in order to flag those areas that would require the lateral transfer of supply from elsewhere via trade or aid. This would provide a first order boundary condition for further studies or for use by policy makers. As a consequence, specifically accounting for the myriad of political, social and cultural factors that affect lateral transfer, access to, and distribution of supply is beyond the scope of this study.

## 2 Materials and Methods

### 2.1 Modelling framework

In the current study, we couple two sectoral models to assess the future supply and demand trajectories for the Sahel region. We divided the modelling framework into three parts (Fig. 1), where the first part describes NPP supply; the second
encapsulates NPP demand, while the third combines the two.

### 2.1.1 NPP supply

Supply is dependent on vegetation growth, and can be quantified as net primary production (NPP), which is defined as the difference in gross photosynthetic assimilation of carbon and carbon loss due to autotrophic respiration, per area per unit time (Foley, 1994). NPP is an established measure of ecosystem productivity indicating how much energy is available for
all life on Earth. We estimated future plant productivity of the Sahel with the BME (Biome-based NPP meta-models). The BME is a rapid biome-based NPP meta-model that emulates the performance of the more complex model LPJ-GUESS (Lund-Potsdam-Jena General Ecosystem Simulator, Smith et al., 2014), but in a simplified, and more time-efficient manner. LPJ-GUESS is a state-of-the-art dynamic global potential natural vegetation model that incorporates carbon and nitrogen interactions (Smith et al., 2014). LPJ-GUESS (carbon cycling only)shows good skill in predicting NPP at regional and global
scales (Hickler et al., 2008; Tang et al., 2010). We developed the BME using LPJ-GUESS NPP simulations driven by several climate and $CO_2$ concentration perturbations (see Table A1). The biome definition in BME is taken from the Major Biome classification (MBC) (Reich and Eswaran, 2002), which stratifies the terrestrial biosphere into 13 biomes based on soil moisture and temperature regimes. We chose this biome definition because it represents a trade-off between global biosphere classifications that either have too many biomes or too few, compared to other stratifications (Kottek et al., 2006;
Metzger et al., 2013; Olson et al., 2001). The trade-off also allowed for a reasonably accurate reproduction of vegetation dynamics, compared with LPJ-GUESS. For our study, we parameterized BME for the four major biomes of the Sahel: a) desert tropical, b) desert temperate, c) tropical semi-arid and d) tropical humid (Fig. 2). A recent study by Gonzalez et al. (2010) shows that climate change has the potential to shift biomes by the end 21[st] century. For simplicity, we therefore assumed static biomes that persist during climatic changes encountered during the modelling period (year 2000-2100). A
detailed description of the BME implementation is provided in Appendix A.1.

We also evaluated LPJ-GUESS (e.g. Olin et al., 2015) and BME performance (magnitudes, trends and interannual variability) by first implementing a global biome-by-biome-level validation, where results from the Sahel are highlighted. We then compare BME estimates with LPJ-GUESS NPP simulations (including LPJ-GUESS managed land, in order to gauge the effect of agriculture on NPP, keeping in mind that BME is based on a model of potential natural vegetation) that
were excluded from BME parameterization. Finally, we compare BME estimates against MODIS-derived NPP (2000-2006) (Running, 2004), as well as country-level censuses of crop yield trends from Rey et al. (2013). We also include a comparison

with LPJ-GUESS C (carbon cycling only), a version that has been previously validated at the global scale (e.g. Hickler et al., 2008).The evaluation covered the entire Sahel region and was run from 1970 to 2006 (see Appendix A.2).

We forced BME with climate data (spatial resolution 0.5 x 0.5 degrees) from five GCMs (General Circulation Models, including HADLEY, GFDL, IPSL, MIROC and NorESM), and [$CO_2$] based on four RCPs (Representative Concentration Pathways, including 2.6, 4.5, 6.0 and 8.5) to estimate annual total NPP in kg dry-weight m$^{-2}$ yr$^{-1}$ (DW, dry-weight). We used climate data derived from runs across the 4 RCPs for each of the 5 models. We then calculated annual means of the five GCM NPP yields, resulting in four NPP time-series (covering each RCP) each spanning from 2000 to 2100. By averaging the GCM based NPP estimates we decreased the data amount while reducing spatial and temporal variability stemming from individual GCMs. In the next step, we summed the annual NPP estimates over the grid cell area in m$^2$ using the latitude of each grid cell centre. Additionally, we used annual land use projections from Hurtt et al. (2011) to calculate the total area of pasture and cropland in each grid cell. This allowed us to estimate annual total $NPP_{supply}$ (kg cell$^{-1}$ yr$^{-1}$) for pasture and cropland separately. We estimated crop- and grassland scaling factors for each country by dividing the PLUM-predicted land-use area with the total land-use area provided by the Hurtt et al. (2011) dataset (Table C1). We then applied the scaling factors to the Hurtt et al. (2011) land-use data and multiplied the resulting crop- and grassland areas with the NPP estimates to obtain annual $NPP_{cereal\_supply}$ and $NPP_{grazing\_supply}$ (kg DW cell$^{-1}$ yr$^{-1}$). We addressed potential developments in the wider use of existing agricultural technology that result in higher plant productivity with a technology improvement factor, where this factor is used to decrease the yield gap. The technology improvement factor is the aggregate result of parameterizing three technology related parameters (trends in technology, change in yield with GDP per capita, as well as how agricultural management practices are transferred both within and between countries) that are consistent with the scenario storyline of each SSP. Parameter ranges have been empirically determined based on analysis of data between the years 1995 and 2005. Yield gaps are not necessarily closed, but are decreased (see Engström eta al., 2016 for more detail).We then used country-wide yield gap fractions provided by PLUM spanning from 2000 to 2100 (Engström et al., 2016b; Licker et al., 2010). The yield gap fractions are country-specific and dependent on technological development in each scenario, and are thus consistent with the SSP storylines (Engström et al., 2016b). For example, a scenario with strong technological change has large decreasing yield gaps while a scenario with slow technological change has slowly, or stagnating (or even increasing) yield gaps. Here, we calculated yearly technology improvement factors by dividing the inverse yield gap fraction (i.e. 1-yield gap fraction) of the respective year with the inverse yield gap of the starting year (i.e. 2000). Thereafter, we applied the annual technology improvement factors to the $NPP_{cereal\_supply}$ (kg cell$^{-1}$ yr$^{-1}$) of the respective year and country.

Finally, we used root-to-shoot ratio (R:S) to remove below ground biomass NPP of croplands (we exclude tubers and groundnuts) and pasture from our NPP estimates, since this component cannot generally be appropriated by humans or by the majority of animals. For croplands, we assumed common agricultural practice across the Sahel region and therefore applied a region-wide R:S=0.1 (Jackson et al., 1996). This a reasonable R:S since crops produce low root biomass compared to the above ground biomass. Moreover, we extracted the consumable parts of the above ground NPP by using a region-wide crop harvest index of 0.235, which is the average of reported harvest indices for maize, millet, sorghum and wheat (Haberl et

al., 2007; Wirsenius, 2000). In contrast to crops, grasslands produce more below ground NPP in relation to above ground NPP (R:S >1) (Jackson et al., 1996). Therefore we considered the climatic limitations of individual biomes by extracting above ground NPP (for grasslands): a) desert tropical R:S=2.8; b) desert temperate R:S=1.1; c) tropical semi-arid R:S=2.8; and d) tropical humid R:S=1.6 (IPCC, 2006; Mokany et al., 2006).

### 2.1.2 NPP demand

For the calculation of NPP demand only, the parts of NPP that are available for direct consumption (excluding e.g. NPP preserved in e.g. national parks) are here considered. Future NPP demand can be projected applying a set of consistent assumptions for future societal and economic developments, described in socio-economic scenarios. We simulated future NPP demand for each country of the greater Sahel with PLUM, which is based on a conceptual model of socio-economic processes that determine global agricultural land-use change (Engström et al., 2016c). These processes include population and economic development, the consumption of cereal, milk and meat dependent on economic development and lifestyle/diet choice and the development of cereal yields dependent on technological change. PLUM is driven by country-level population and gross domestic product (GDP) data, and a range of parameters that characterize the development of the socio-economic processes mentioned above. PLUM was evaluated against historic (1991-2010) consumption and land-use data at the country scale and was shown to reproduce land-use change and consumption patterns at the global aggregated scale (Engström et al., 2016c). Due to the model's relative simplicity and the limited number of scenario parameters it is suited for scenario studies and was used to quantify uncertainty ranges for global cropland scenarios based on the Shared Socio-economic Pathways (SSPs) (Engström et al., 2016b). Mean cropland change for the five scenarios resulted in 963-2280 Mha cropland by 2100 compared to 1503 Mha cropland in 2000. The parameter-settings resulting in the uncertainty ranges for each scenario are described in Engström et al. (2016b) and the reported mean values were used in the current study. For more details see Engström et al. (2016b). In the version of PLUM applied in our study, we introduced an additional parameter which characterizes the increasing intensification of the livestock production systems in scenarios with strong increase in milk and meat consumption (Engström et al., 2016a). This process was previously not included in PLUM, but it was later identified to lead to an underestimation of land requirements for scenarios with strong increases in milk and meat consumption (Engström et al., 2016b).

We forced PLUM with the five socio-economic scenarios from 2000-2100 (see box outlined in red in Part 2 of Fig. 1) taken from the SSPs, but it is important to remember that is it also coupled to the BME (see dashed arrow in Fig. 1) through annual country-level total NPP estimates for cropland. Aggregation of BME NPP estimates was implemented as described in Engström et al. (2016b), except that cropland fractions in 2000 from MIRCA dataset were replaced with Hurtt et al. (2011) cropland fractions from 2000-2100.

Finally, we defined the demand of NPP as compounds that are necessary for human livelihood in the Sahel region, following the $NPP_{demand}$ approach of Abdi et al. (2014). However, our approach differs from Abdi et al. (2014) by distinguishing

between the demand of cereal- and pasture products. PLUM outputs were combined to determine $NPP_{cereal\_demand}$ as given in Eq. (1) and $NPP_{grazing\_demand}$ (see Eq. A9 in the Appendix A.3).

$$NPP_{cereal\_demand} = NPP_{food} + NPP_{feed} \qquad (1)$$

where $NPP_{cereal\_demand}$ is the total amount of annual NPP needed for human appropriation via cropland; $NPP_{food}$ (ton country[-1]) is the NPP needed for consumed cereals; and $NPP_{feed}$ (ton country[-1]) is the amount of cereal based fodder to support the region's livestock population. $NPP_{grazing\_demand}$ is the NPP needed for sustaining the livestock by grazing (ton country[-1]). Furthermore, we converted $NPP_{cereal\_demand}$ and $NPP_{grazing\_demand}$ to per capita demand (kg person[-1]) using country population projections of the corresponding year in the SSP. A detailed methodology of the PLUM output combinations to satisfy Eq. (1) is given in Appendix A.

In the following step, we disaggregated the annual per capita $NPP_{cereal\_demand}$ and $NPP_{grazing\_demand}$ from country to 0.5 degree grid cell resolution in order to facilitate the spatial analysis of NPP supply and demand at the grid cell level. For that we multiplied annual per capita demands with gridded population data (0.5 x 0.5 degree resolution) of the corresponding years. The disaggregated annual $NPP_{cereal\_demand}$ and $NPP_{grazing\_demand}$ (kg cell[-1] yr[-1]) are therefore weighted by population density (i.e. population centers achieve high demand).

**2.1.3 NPP Supply-Demand Balance**

In the next step, we combined the $NPP_{supply}$ (i.e. RCP based) with the $NPP_{demand}$ (i.e. SSP driven) using a SSP-RCP likelihood matrix (Engström et al. (2016b), see Table 1) in order to facilitate the analysis of the NPP supply and demand balance. To create the likelihood matrix, a qualitative probability was assigned to describe the likelihood of a SSP resulting in a RCP (Engström et al., 2016b). The qualitative likelihood estimates are based on experts' judgements, ranging from "very low" to "very high" and were translated to quantitative probabilities (Engström et al., 2016b). For the analysis, we considered SPP-RCP combinations with likelihoods above > 0.05 (> very low likelihood).

Next, we computed cereal-based (i.e. $NPP_{cereal\_balance}=NPP_{cereal\_supply}-NPP_{cereal\_demand}$) and grazing (i.e. $NPP_{grazing\_balance}=NPP_{grazing\_supply}-NPP_{grazing\_demand}$) balances. In order to combine the balances meaningfully we defined four rules as outlined in Table 2. Rule no. 1 states that a deficit of cereal products ($NPP_{cereal\_balance}<0$) cannot be balanced with surplus of plant growth on grassland ($NPP_{grazing\_balance} \geq 0$) because grassland products are inappropriate for direct human consumption, resulting in all grazing surplus being disregarded. Rule no. 2 regulates the treatment of cereal and grazing surplus occurring simultaneously, where pasture NPP surplus ($NPP_{grazing\_balance} \geq 0$) is ignored but the cereal-based NPP surplus ($NPP_{cereal\_balance}\geq 0$) is retained. This surplus is of interest because it can potentially balance NPP shortages in adjacent grid cells as well as on the country level. Rule no. 3 permits the combination of cereal ($NPP_{cereal\_balance}<0$) and grazing ($NPP_{grazing\_balance}<0$) deficits in order to quantify the total NPP shortage of the grid cell. The last rule allows supplementation of grazing-based shortages ($NPP_{grazing\_balance}<0$) with cereal surplus ($NPP_{cereal\_balance}\geq 0$).

## 2.2 Scenarios

In the current study, we combine four Representative Concentration Pathways (RCPs) with five SSPs which are the latest future climate, [$CO_2$] and socio-economic projections (O'Neill et al., 2014; van Vuuren et al., 2011; van Vuuren et al., 2013) from the Intergovernmental Panel on Climate Change (IPCC) Fifth Assessment Report (AR5) framework. Each RCP represents a different cumulative measure of future human greenhouse gases (GHG) emissions and is defined by their radiative forcing targets for the year 2100, and which range from 2.6 to 8.5 W m$^{-2}$ (van Vuuren et al., 2011). For each RCP, we obtained climate data from the Inter-Sectoral Impact Model Intercomparison project (ISI-MIP), containing climate simulations of five General Circulation Models (GCMs) for each RCP (Hempel et al., 2013). (GCMs : (Collins et al., 2013; Dufresne et al., 2013; Dunne et al., 2013; Iversen et al., 2013; Watanabe et al., 2011)). The climate data (0.5 x 0.5 degrees resolution) was bias corrected by the ISI-MIP approach that preserves trends in absolute changes in monthly temperature, and relative changes in monthly precipitation amounts (Hempel et al., 2013). For future socio-economic developments, the SSPs consider different narratives of future population levels, urbanization scenarios and economic development (O'Neill et al., 2017; van Vuuren et al., 2013) as summarized in Table 3.

No mitigation strategies are assumed and resulting scenarios are thus reference scenarios. Furthermore, for each of the considered SSP and RCP combinations, we used a distributed population projection dataset at 1 km$^2$ from Boke-Olén et al. (2017). The population dataset was created by Boke-Olén et al. ( 2017) to match both the RCP specific urban fractions from Hurtt et al. (2011) and SSP country urban and rural population counts. Hence, one population dataset exists for each SSP and RCP combination used in this study. We resampled (summed) the population dataset to the same spatial resolution as the climate data (0.5 x 0.5 degrees) and grid cells with population count below 3000 people per grid cell (~ one person per 1km$^2$) were excluded following Abdi et al. (2014).

Additionally, variation in NPP supply estimates originating from the five GCMs was retained for an estimate of supply uncertainty to be included in the analysis. Uncertainty estimates for NPP demand associated with each SSP were derived from the results of Engström et al. (2016b) and applied here. In their study, conditional probability ranges were defined for twelve PLUM input parameters (reflecting uncertainties in SSP interpretation and quantification) in order to estimate uncertainty in a range of PLUM outputs.

## 2.3 Study area

The study area covers the African continent between roughly 5° and 25° northern latitude and stretches from the Red Sea to the Atlantic Ocean, hereafter referred to as the greater Sahel. Following Abdi et al. (2014), the area also includes the neighbouring countries of the Sahel belt (encompassing 21 countries see Table 4). Note that this study uses the African country definition for the year 2000 where South Sudan was a part of Sudan. The actual Sahel belt is described by an annual rainfall range between 100mm and 600mm (hatched area in Fig. 2). The Sahel is an arid and semi-arid region that separates the Sahara desert from the humid and tropical regions to the south. The northern parts of the region border the Sahara Desert

with low mean annual precipitation (<100mm) while the southern parts of the Sahel belt border the savannas of the tropical semi-arid biome, permitting increased plant productivity due to higher mean annual rainfall (~600mm). The southern parts of the study area cover the tropical semi-arid and tropical humid biomes with much higher mean annual precipitation amounts ranging from 600 to 1000 mm enabling larger vegetation growth. The study area is one of the poorest as well as most

5  technologically underdeveloped regions on the African continent (Chidumayo and Gumbo, 2010).

# 3 Results

In the following the results are presented at Sahel, country and local (grid cell) level. Results for the different scenario combinations are reported, but emphasis is given to the SSP2-RCP6.0 scenario, as this scenario deviates least from current socio-economic and climate trends at the global level. Additionally, Fig. 3a also provides a basis for interpreting Fig. 3b.

## 3.1 Sahel

Per capita demand exceeds supply in the early 2040s for SSP2-RCP6.0 after which a very high likelihood for shortfalls begins in 2070 (see black dots in Fig. 3a showing non-overlapping 95% confidence limits). By 2050, per capita demand almost doubles while per capita supply drops by almost 30% for the same scenario. Across the scenarios, differences in the timing of the start of persistent supply shortfalls with high statistical certainty are observed (see black dots in Fig. 3b). Three of these high likelihood shortfalls begin at 2050 or before (SSP5 scenarios – see black dots in Fig. 3b) while an additional six display shortfalls with high certainty by the end of the 21$^{st}$ century (black dots in Fig. 3a, b). Out of these nine, two scenarios never achieve a sustained run of shortfalls (SSP2-RCP6.0, SPP2-RCP8.5). In total, there is better than an even chance for shortfalls before 2050 for 9 scenarios (exceptions are SSP1-RCP2.6, SSP1-RCP6.0, and all SSP4 scenarios.

Variations in the timing of onset and end of supply shortfalls are generally greater between the SSPs than between the RCPs (Fig. 3b). For SSP2 and SSP3 scenarios, onsets of high likelihood supply shortfall range from the early 2050s to the mid-2070s (even chance from late 2030s to early 2050s). The SSP5 family shows the largest deficits of high likelihood shortfalls beginning in the 2040s-2050s (even chance from the early 2030s), and after several decades of deepening begin to diminish in the 2080s. Shortfalls with high certainty never emerge for SSP1 (even chance from the early 2050s) while the SSP4 scenarios show sustained but diminishing surplus throughout.

## 3.2 Country-level

For scenario SSP2-RCP6.0, per capita NPP balances generally show a decrease for all countries. Eleven countries (out of twenty-two) experience per capita shortages by 2050, up from two countries (Djibouti and Mauritania) in 2000. Ethiopia shows the most extreme shortfall while Togo the greatest surplus. The largest change amongst all countries (is exemplified by Niger which starts with a surplus in 2000 but ends up with a deficit by 2050. Conversely, Djibouti shows a small decrease in deficit over the period (Table 4).

Large changes in per capita NPP balance are caused by contrasting development of NPP supply and demand, as analyzed in the following two paragraphs. Despite large total NPP increases between 2000 and 2050 (SSP2-RCP6.0), per capita NPP supply decreases for almost all countries, the largest decreases being for Niger and Sudan while an increase is noted for Liberia.

Since all countries double or even triple their population counts from 2000 to 2050 (Table 4), large increases in demand occur over the 50 year period, while even per capita demand increases. By 2050, the largest increases in demand per capita are projected for Liberia, Ethiopia and Ghana by 2050 respectively (Table 4).

Generally, the differences in NPP balances across scenarios are high, with the largest variations attributed to the SSPs as opposed to the RCPs (Table C2), with two countries (Sierra Leone and Liberia) showing considerable variation across the scenarios (coefficients of variation > 2.0).

### 3.3 Local level

For SSP2-RCP6.0, the localities experiencing negative NPP balance expand and become more connected between 2000 and 2050. By 2050, a semi continuous band of low magnitude NPP shortage emerges (generally > -0.2 Mt dry weight yr$^{-1}$ per grid cell), stretching from the Atlantic Ocean to the Red Sea, between 15$^o$ and 20$^o$ N (Fig. 4b). In the east, this band extends down along the coast and wraps around the horn of Africa. A separate band of similar magnitude emerges toward the south, from just above 10$^o$ N, and stretching toward the east-southeast into Cameroon. Additionally, four separate locations of large magnitude shortfalls (> 1.5 Mt dry weight yr$^{-1}$ per grid cell) of varying extents emerge. The first hotspot (relatively small cluster of large magnitude shortfall) is located along the Nigerian coast, stretching from the metropolitan areas of Lagos to the densely populated area of the Niger delta (Fig. 4a, h1). The second hotspot is located in northern Nigeria, close to the city Kano (Fig. 4a, h2) while the third is located in the Ethiopian highlands of Eastern Africa (Fig. 4a, h3). Finally, the fourth covers the area of around Khartoum in the Sudan (Fig. 4a, h4). Elsewhere, very small pockets (e.g. 1 grid cell in size) of large magnitude NPP shortages (<-1.0 Mt DW yr$^{-1}$ per grid cell) are distributed unevenly across the region.

Both supply and demand increase over most localities for the SSP2-6.0 scenario from 2000 to 2050 (Fig. 4 c-d). For supply, largest increases (up to, and exceeding 1 Mt dry weight yr$^{-1}$ per grid cell) occur in those areas that already see large supply in 2000, including the southern parts of Ivory Coast and Ghana, and most of Nigeria and the southern part of Niger (Fig. 4c,d). Smaller increases occur throughout central Sudan and Ethiopia. Large magnitude increases (between 1 and > 2 Mt dry weight per year$^{-1}$ per grid cell) in demand are seen for distinct geographic regions, the largest patches covering coastal Nigeria, northern Nigeria-southern Niger, north-central Sudan around Khartoum, and Ethiopia (Fig. 4f). By-and-large, these correspond to the hotspots of supply shortfall identified in Fig. 4b. Smaller areas, sometimes no larger than one grid cell, are seen scattered across Sudan, Chad, the west coast, and south Sudan.

The general geographical patterns of NPP shortage remain persistent across all scenarios, including the four hotspots identified for SSP2-RCP6.0. The largest magnitude shortages are indicated for SSP5-RCP8.5 (Fig. B1).

## 4 Discussion

### 4.1 Sahel-level

World-wide cereal production in 2010 amounted to 2400 Mt and current food aid shipments to countries in the Sahel are below 1 Mt yr$^{-1}$ (FAOSTAT, 2016). At present about 260 million people are chronically undernourished in Africa (FAOSTAT, 2015) and this is despite the fact that we also estimate a per capita NPP surplus of 860 ($\pm$144) kg DW yr$^{-1}$ (corresponding to 309 ($\pm$52) Mt DW yr$^{-1}$) in the Sahel for the year 2000. This implies that current challenges are associated with other determinants such as access to and distribution of resources (Brown, 2016; Olsson, 1993; Pinstrup-Andersen, 2009). These challenges are set to increase in the future, particularly for scenarios with high social and economic inequalities (SSP4). Furthermore, the majority of all other scenarios show that by mid-century, the NPP surplus will be much reduced compared to the year 2000. According to the sustainable development goals, hunger and all forms of malnutrition should be eradicated by the year 2030 (UN, 2016), but under the current trend given by the SSP2-RCP scenarios, there is a risk that 15-25% (160 to 270 million people) of the population would not be able to be supported with NPP supply (on the basis of assumed adoption of diets rich in animal products, consistent with the SSP2 storyline) and are therefore at high risk for malnutrition by 2050.

Presently, the Sahel has a high reliance on their own land by producing 90% of domestic food consumption resulting in very little import or export of crops (Abdi et al., 2014). This implies that agricultural resources from global trade will need to increase considerably in order to reduce the future food shortages across the region. Participation in global markets and investments in infrastructure that enable trade of food commodities to ensure food security via trade will therefore be important (D'Odorico et al., 2014). However, it needs to be kept in mind that the simulated shortages partly occur due to steep increases in per capita consumption. For example, while reducing social inequities is clearly desirable (as embraced by the SSP5 RCP scenarios), from a sustainability perspective, it is questionable if this should mean that developing countries follow the development path of economically developed countries and adopt diets with very high consumption levels of animal products (O'Neill et al., 2017). The adoption of sustainable diets (i.e. reduced contribution of animal products to diets) has to be envisaged as a strategy consistent with efforts to reduce food demand to healthy and sustainable levels (Smith, 2013). This would be consistent with the SSP1 ('taking the green road' scenarios) where sustainable diets are adopted statistically significant shortages never develop (e.g. Fig. 3b).

### 4.2 Country-level

Beyond the import of agricultural products to the Sahel, inter-country trade of such resources will also need to become more important later in the 21$^{st}$ century. Trade relations between productive and high-demand countries should be encouraged (Ahmed et al., 2012). For instance, Cameroon, Ivory Coast, Chad and Togo produce NPP surplus for SSP2-RCP6.0 by 2050 which could be traded to neighbouring countries with NPP shortages (e.g. Nigeria). Across the scenarios, some countries showed continuous NPP shortfalls (e.g. Mauritania), while Ivory Coast and Guinea consistently produce NPP surplus (Table

C2). The large range of different climate conditions in the Sahel region implies that those countries within the tropical humid (and partly in tropical semi-arid) biome have larger potential NPP compared to countries in the desert temperate biome. We note that the closure of yield gaps by 2050 (for scenario SSP2-RCP6.0) would result in a change in mean per capita NPP balance from -107 kg DW yr-1 (see Table 3) to 9 kg DW yr$^{-1}$. Though the balance for many countries will still be negative,

the shortfall magnitudes would be reduced. . Decreased supply due to losses of food during harvest, transport and storage (i.e. household level) should be reduced through improvements of agricultural management, infrastructure and educational development (Godfray et al., 2010). For most countries however, the different socio-economic development pathways prescribed by the SSPs lead to high inter-scenario variability (having positive or negative balances depending on the scenario) and will determine if countries have the potential be a net exporter or importer of resources.

**4.3 Local-level**

At the local-level, robust NPP shortages across scenarios were found to be strongly linked to densely populated areas. For the example of SSP2-RCP6.0, by 2050, the number of grid cells with high population density (i.e. > 1 million population per 25 km x 25 km increased substantially compared to 2000 (see Fig. B4)). For instance, > 1 million people per grid cell trigger NPP shortages in Ethiopia while > 2 million people per grid cell induce NPP shortfalls in Nigeria for SSP2-RCP6.0 by 2050.

The NPP shortage hotspots in Nigeria and Ethiopia agree geographically with reported considerable NPP demand expansions in the 2000s (Abdi et al., 2014) indicating a combination of population growth and increased consumption as explanatory factors. Furthermore, the projected deepening and persistent shortages in urban areas underscore the hypothesis that the urban poor are especially at risk for food insecurity since they neither have the means to purchase food on the markets, nor the means to be self-sufficient due to limited land in densely populated areas (Lynch et al., 2001). Thus,

connecting productive hinterlands with metropolitan areas will need to be achieved (Owuor, 2007).

That the locations of the hotspots and the overall patterning of NPP shortfalls remain consistent across all scenarios narrows the number of future policy choices in the region for alleviating environmental insecurity despite the very different assumptions and uncertainties embedded in the scenarios and models (Kwakkel and Pruyt, 2013).

**4.4 Additional Perspectives**

Livestock mobilization is one way local populations generally employ to manage risk (e.g. Herrmann et al. (2014). This strategy may help regulate supply shortfalls locally, and over the short term. Even if the Sahel were to continue to green up (increase in NPP supply) this would  not necessarily imply an increase in the amount of usable NPP or an enhancement in health and well-being. Recent studies in  the Sahel show that much of the recent greening, at least in some regions, is due to undesirable shifts in species composition (e.g. Herrmann et al. (2014)), reductions in biodiversity and an increases in woody

biomass (e.g. Brandt et al. (2015)).Campbell et al. (2014) underscore the importance of family planning and education in the Sahel in order to curb population growth. Generating demand for various forms of birth control and gender empowerment

would be two key interventions that would work towards slowing population growth, improving health and facilitating income generation. These interventions would act to curtail supply shortfalls in the future.

## 4.5 Mechanisms of changes in future NPP supply and demand

### 4.5.1 NPP supply

In order to isolate the $CO_2$ (rainfall) effect on NPP increase for RCP6.0, we compared a simulation where rainfall ($CO_2$) was held constant with a simulation where both were held constant for the period 2000-2050 for all GCMs. We found that supply increases mostly due to $CO_2$ fertilization (see Fig. B2), with very little attributed to rainfall. However, yield gap closure from SSP2 contributes most to the increase in simulated NPP supply (Fig. B2).

The $CO_2$ fertilization effect increases with the magnitude of climate change and explains the smaller shortages in SSP-
RCP8.5 scenarios compared to SSP-RCP4.5 scenarios (Fig. 3b). The decreases in yield gap (applied to the NPP supply and demand balance through the technological improvement factor) are simulated with PLUM and are strongly dependent on scenario-driven assumptions for technological change. High rates of technological change explain the decreasing shortages at the end of the 21$^{st}$ century for SSP1-RCPs and SSP5-RCPs scenarios. For example, in the SSP1-RCP scenarios, the yield gap decreased from 0.55 in 2000 to 0.43 by 2050 in Nigeria and from 0.69 in 2000 to 0.56 by 2050 in Ethiopia. By contrast,
slow technological change in SSP3-RCP scenarios leads to very small decreases in yield gaps, e.g. for Nigeria to 0.54 by 2050 while no improvement at all was simulated for Ethiopia. Uncertainties in yield improvements driven by technological development are very large and critically dependent on investments as well as on infrastructural and political development in developing countries (Engström et al., 2016b; Licker et al., 2010; Mueller et al., 2012). Reducing yield gaps to 0.5 in Sub-Saharan countries can be achieved by intensified nutrient management, while decreases down to 0.25 require increased
irrigation and fertilization (Mueller et al., 2012). However, Elliott et al. (2014) underscore that freshwater limitations in the dryer regions of the globe could limit agricultural production, and even lead to the reversion of irrigated farmland to rainfed farmland, thereby negatively affecting food production. Conventional agricultural intensification, however, can result in environmental degradation, vulnerability to pests, and depletion of aquifers (Ceccato et al., 2007; Foley et al., 2005). Agricultural management should consider strategies of sustainable intensification while simultaneously considering
adaptation of agriculture to changing climates (Dile et al., 2013; Pretty, 2008, 2011).

An additional driver of NPP supply is the simulated increase in agricultural land area provided by PLUM (i.e. grass- and cropland – Fig. B5). However, the simplified representation of grassland in PLUM potentially underestimates the expansion of agricultural land into naturally vegetated areas, and thus the magnitude of total NPP supply. As with agricultural intensification, the expansion of agricultural land into natural forests and grasslands has the potential to produce negative
impacts on the environment and on climate (Canadell and Schulze, 2014; Foley et al., 2005; Pugh et al., 2015).4.5.2 NPP demand

Despite increases in future NPP supply, according to our results, the Sahel is likely to experience NPP shortages for most NPP scenarios due to strong increases in demand. Generally, the increasing NPP demand in the Sahel region can be explained by doubling to tripling population in the period 2000-2050 across the scenarios (Fig. B3a). However, changes in economy, lifestyle and consumption patterns as simulated with PLUM were shown to be the important drivers for large total NPP demand. For example, in the SSP5-RCP scenarios, per capita NPP demand almost triples (2000-2050, Fig. B3b), driven by the adoption of meat- and milk-rich diets and processed food as previously pointed out by (Kearney, 2010; Tschirley et al., 2015). Increased per capita NPP demand coupled with the doubling in population (2000-2050) leads to almost seven-fold increases in total NPP demand during the period 2000-2100 for SSP5-RCP scenarios. By contrast, for SSP4-RCP scenarios population triples (2000-2050), but widening income gaps and no improvements in diets in the poor population lead to declining per capita NPP demand (Fig. B3b) with a low increase (compared to other scenarios) in total NPP demand (doubling between 2000 and 2050, Fig. B3b). The relatively weak increase of total NPP demand in the SSP4-RCP scenarios is the underlying reason for a sustained NPP surplus in the scenarios. The NPP surplus per se is not an indicator for achieved food security, as suggested by the decreasing per capita demand (described above). By contrast, food insecurity will be likely more wide-spread than today according to the SSP4-RCP scenarios, aggravated by strong inequalities within the population that are likely to worsen food distribution and food access for the poor (Pinstrup-Andersen, 2009).

The uneven projected changes in per capita NPP demand across countries (Table C1) are partly due to contrasts in the evolution of drivers (e.g. income) for different countries, but also due to differing initial conditions for the different countries. In countries with initially higher per capita demand (e.g. Sudan) the potential to increase per capita demand is limited, while for countries with lower initial per capita demands (e.g. Ethiopia) the potential to increase demands is comparatively higher. Finally, the NPP demand estimates are limited by the assumption of cereals, meat and milk being proxies for food supply, which for countries with high shares of pulses and tubers in their average diet in particular, underestimates the NPP demand.

## 4.6 Uncertainties

We show that the deep uncertainties represented by the scenarios i.e. not knowing how drivers (e.g., population, technological change) will develop in the future (van Vuuren et al., 2008) are major sources of uncertainty leading to variations in our results (Fig. 3b). The variability in NPP supply and demand, originating from the five GCMs and uncertainties in SSP interpretation and quantification (see Engström et al. (2016b) and Table 1 and Table B1 ), respectively, allows us to assess, with high statistical confidence, when the onset of supply shortfalls begin and are sustained.

Additional uncertainty exists with respect to the total magnitude and trends of simulated NPP supply, given the lack of ground truth for the region, and that differences in NPP trends between other models is very large (e.g. Friend et al., 2014; Körner et al., 2006; Pugh et al., 2016; Rosenzweig et al., 2014). Indeed, recent observational evidence suggests that the effect of $CO_2$ fertilization on plant growth may be constrained by counteracting feedbacks associated with increasing atmospheric moisture demand and nutrient availability (e.g. Smith et al., 2016; Wieder et al. 2015). For example, NPP is

reduced under warmer and dryer conditions due to moisture stress, particularly in temperate and arid ecosystems. Future trends NPP trends in the Sahel could therefore be strongly determined by changes in the frequencies of wet years versus dry years, with the dry years counteracting the $CO_2$ fertilization effect. Furthermore, nutrient supply rates may not be able to keep up with extra demand associated with $CO_2$ fertilization, and leading to a depletion of soil nutrients, as current evidence suggests. This could also curtail the $CO_2$ fertilization effect, particularly in the more southerly parts of our study area, where nutrients tend to become a limiting factor. We performed a simple experiment negating the $CO_2$ fertilization effect in order to gauge its impact on supply-demand balance on all scenarios. For the SSP2-RCP6.0, per capita demand has an equal chance of exceeding per capita supply in 2036 for the SSP2-6.0 scenario as opposed to 2043 if $CO_2$ fertilization in included (Fig. B7), with a very high likelihood of continuous supply shortfall beginning in 2056, as opposed to 2073 with $CO_2$ fertilization. The effect on all other scenarios is an earlier shift to the onset of supply shortfalls, by about 10 years, compared to Fig. 3b (see Fig. B7). Supply shortfalls with high likelihood of occurrence (black dots showing non-overlapping 95% confidence intervals) are similarly shifted, and occur with greater consistency and frequency. All of this suggests that the NPP increases found in our current analysis are likely optimistic, due the potential overestimation of the $CO_2$ fertilization effect, as well as the fact that BME is based on potential natural vegetation.

Finally, we note that country-specific scaling factors used to convert PLUM output to per pixel changes using the Hurtt et al. (2011) data set for the year 2000 did not depart substantially from 1 (scaling factors for the larger countries were all within 10%, and the area weighted mean of the scaling factors was 0.95), but a few smaller countries in West Africa diverge by more than 25% (<0.80 or > 1.25) (see Table C1). We expect these to have only marginal influence on the results at the regional level, but could have a larger impact on localities along the West African coast (Fig. 4 and Fig. B1).

Other sources of uncertainty, such as model uncertainty stemming from the supply and demand models (Alexander et al., 2016) are not presently taken into account.

## 5 Conclusions

In the Sahel, population growth and climate change raise the question of whether the demand for NPP will outstrip supply during the 21[st] century. In order to address this question, we developed a reduced-complexity framework capable of generating a range of NPP supply-demand trajectories for different Sahel futures at the regional, country, and local levels of aggregation. These results are based on differing climate, [$CO_2$], and socio-economic scenarios supplied by different SSP and RCP combinations.

We conclude that the potential for NPP self-sufficiency in the Sahel will not likely be attainable later in the 21[st] century. The most likely consequence will be that hunger and malnutrition will become more widespread than it is currently, undermining the UN sustainable development goals. This highlights the importance of establishing strategies that address the reduction of NPP demand, increasing its supply as well as facilitating its access, particularly for the urban poor. The consistency of geographical shortfall patterns across all scenarios also suggests that, despite deep uncertainties associated with assumptions

about how the future unfolds and uncertainties associated with NPP supply magnitudes and trends, a relatively narrow range of policy interventions can be crafted.

Finally, we advance previous research by showing how NPP supply-demand balance (a key metric for quantifying resource shortfalls over large regions, but applied retrospectively in previous studies) can also be used to explore the impact of changing socio-economic and climate assumptions in the Sahel to support policy.

**Acknowledgements**

FS acknowledges support from the Helge Ax:son Johnsons Stiftelse. SO, KE, and JWS acknowledge support from the FORMAS Strong Research Environment project Land use today and tomorrow (LUsTT; dnr: 211-2009-1682), while JWS also acknowledges partial support from LUCID (www.lucid.lu.se), a FORMAS-funded Linnaeus Centre of Excellence at Lund University (dnr: 259-2008-1718). AMA and JA acknowledge funding provided by Swedish National Space Board (Rymdstyrelsen). Finally, VL acknowledges support from the BIODIVERSA project CONNECT via FORMAS.

**Appendices**

**Appendix A Methods**

**A.1 Biome based Meta-model Ensemble**

In this section, we describe the development of the biome based meta-model ensemble (BME) for the Sahel region. BME

consists of rapid NPP meta-models tailored for the desert temperate, desert tropical, tropical semi-arid and tropical humid biome. The BME is based on the dynamic vegetation model LPJ-GUESS (Smith et al., 2014) and NPP simulations following the methodology of Sallaba et al. (2015).

**A.1.1 LPJ-GUESS**

LPJ-GUESS (Smith et al., 2014) is a mechanistic model of plant physiological and biogeochemical processes that

incorporate ecosystem carbon and nitrogen cycles as well as water fluxes. The model uses a detailed individual- and patch-based representation of vegetation structure where individual plants differ in growth form, phenology, life history strategy and photosynthetic pathway, demography and resource competition. LPJ-GUESS is forced by various climate (i.e. solar radiation, temperature and precipitation), atmospheric $[CO_2]$, soil characteristics and nitrogen deposition. Vegetation is represented as plant functional types (PFTs) with different age cohorts interacting on patch level. Ten generalized trees and

two generalized grass functional types (i.e. C3 and C4 grass) following Smith et al. (2014) were used for global potential natural vegetation (PNV). Several patches (here 25) are applied in parallel within a grid cell with distinguished establishment of vegetation, fire impacts, random disturbance and mortality rate of different age cohorts (Sitch et al., 2003; Smith et al., 2001; Smith et al., 2014). We applied the LPJ-GUESS in cohort mode which represents individual PFTs in different age classes competing for resources (light, water and space) in a patch. We defined disturbance events with an expected return

interval of 100 years following Ahlström et al. (2015). We spun up each LPJ-GUESS simulations with a 500 years long phase of de-trended climate data and a particular $[CO_2]$ (unique for each simulation as outlined in Input data) in order to run the model from bare soil to a vegetation equilibrium state.

**A.1.2 Input Data**

We collected our BME development dataset with a random stratified selection of climate data using the Major Biome

classification (BMC) (Reich and Eswaran, 2002) on a 0.5°x0.5° spatial resolution. The BMC characterizes four biomes in the greater Sahel region based on soil moisture and soil temperature regimes (see Fig. 1). We chose randomly 2-5% of the total cells in each biome.

We overlaid the sampled cells with CRU TS. 3.0 climate data (Harris et al., 2014; Mitchell and Jones, 2005), which have the same spatial resolution. CRU data span from 1901 to 2006 providing monthly data of temperature, precipitation and

cloudiness. Soil texture characteristics were taken from the FAO global soil dataset (FAO, 1991) as described in Sitch et al.

(2003). Historical monthly nitrogen deposition rates were achieved from the Atmospheric Chemistry and Climate Model Intercomparison Project (ACCMIP) database of Lamarque et al. (2010) and processed as described by Smith et al. (2014). We developed climate and $[CO_2]$ scenarios based on a factorial approach where increasing monthly temperature, $[CO_2]$ and changing monthly precipitation amounts are varied multiple variables -at-a-time (i.e. MAT) (Smith and Smith, 2007). We set

maximum changes for each variable (see Table A1) in order to design reasonable climate and $[CO_2]$ scenario limits as described by Sallaba et al. (2015). We used CRU TS 3.0 climate data as the baseline time-series and superimposed the climate and $[CO_2]$ scenarios upon the baseline data while we held the nitrogen deposition rates according to the ACCMIP records. In total, we developed 100 scenarios (including baseline) for each CRU grid cell, which were then applied to simulate NPP in LPJ-GUESS. We assumed that grid cells maintain the biome membership even though the climate

conditions change during the LPJ-GUESS simulations since we consider transitions of vegetation biomes to be long-termed, 100 years.

**Table A1** Minimum and maximum stepwise changes of the climate variables and $[CO_2]$. The magnitudes of increases are related to how much a variable could be adjusted. Temperature was increased in four steps and the other variables in five steps resulting in 100 different climate change scenarios.

| Change attributes | Temperature change [°C] | Precipitation [% of baseline] | Atmospheric $CO_2$ [ppm] |
|---|---|---|---|
| **Minimum Value** | 0 | 50 | 350 |
| **Maximum Value** | 6 | 150 | 670 |
| **Magnitude of increase** | 2 | 25 | 80 |
| **No. of steps** | 4 | 5 | 5 |

### A.1.3 Biome meta-models

We followed the assumption that plant growth is controlled by climate conditions (Sallaba et al., 2015) and defined biome specific assumptions of ecosystem-climate interactions. As Sallaba et al. (2015) we assume that vegetation growth is controlled synergistically by temperature and precipitation. Under optimal climate conditions maximum plant growth can be

reached but decreases when temperature and/or precipitation are not at the optimum. In order to keep the meta-modelling framework as simple but efficient as possible, we limited the meta-model to three input climate surrogates that control plant growth: (1) annual precipitation ($P_{cum}$), (2) maximum temperature ($T_{max}$) and (3) minimum temperature ($T_{min}$) temperature. We followed the methodology of Sallaba et al. (2015) by defining functions of the climate surrogates that yield maximum NPP at baseline $[CO_2]$, combining these in a synergistic function and then adding the $CO_2$ fertilization effect.

For the meta-model development at baseline $[CO_2]$, we scaled the LPJ-GUESS NPP estimates between 0-1 (i.e. $NPP_{min}=0$ and $NPP_{max}=1$) using the highest NPP yield of each biome and combined them with the climate surrogates. The highest NPP yields of the biomes $Max_{biome}$ at baseline $[CO_2]$ are given in Table A3. We then extracted the climate surrogate - NPP value

combinations that yield highest NPP, assuming that maximum NPP yields can only be reached under optimal climate conditions (Sallaba et al., 2015).

For NPP as a function of temperature we assumed a hump-shaped curve relationship, which is based on the temperature-photosynthesis relationship (Sallaba et al., 2015). For $T_{max}$ we developed a function that is built upon the *beta*-distribution as given in Eq. (A1).

$$f(T_{max}) = \frac{\left(\frac{T-Lim_{min}}{Lim_{max}-Lim_{min}}\right)^{\partial-1}\left(1-\left(\frac{T-Lim_{min}}{Lim_{max}-Lim_{min}}\right)\right)^{\beta-1}}{\left(\frac{\Gamma(\partial)\Gamma(\partial\beta)}{\Gamma(\partial+\beta)}\right)}\,a \qquad (A1)$$

where $f(T_{max})$ calculates the NPP yield (relative) of the given temperature surrogate; $T$ is the value (°C) of $T_{max}$; $Lim_{min}$ and $Lim_{max}$ are the minimum and maximum temperature limits of the biome normalizing $T$ between 0 and 1; $\Gamma$ is the gamma function ; $\partial$ and $\beta$ describe the shape of the function and $a$ stretches the function along the ordinate (the amplitude).

For $T_{min}$ we developed a function that is identical to $T_{max}$ as given in Eq. (A2).

$$f(T_{min}) = \frac{\left(\frac{T-Lim_{min}}{Lim_{max}-Lim_{min}}\right)^{\partial-1}\left(1-\left(\frac{T-Lim_{min}}{Lim_{max}-Lim_{min}}\right)\right)^{\beta-1}}{\left(\frac{\Gamma(\partial)\Gamma(\partial\beta)}{\Gamma(\partial+\beta)}\right)}\,a \qquad (A2)$$

where $f(T_{min})$ estimates relative NPP and $T$ is the value (°C) of $T_{min}$. The function parameters of Eq. (A1) and (A2) are provided in Table A2.

For NPP as a function of precipitation we applied two function types because the dataset shows saturation as well as linear NPP growth with increasing precipitation amounts in the Sahelian biomes. Both function types let NPP increase with increasing precipitation amounts until $NPP_{max}$ is reached. Further increasing precipitation levels only yield $NPP_{max}$ because precipitation surplus is assigned as run-off and percolation, following the treatment of high precipitation levels in LPJ-GUESS (Gerten et al., 2004; Smith et al., 2014).

**Table A2** Parameter values for maximum temperature $\boldsymbol{f(T_{max})}$ in Eq. (A1) and minimum temperature $\boldsymbol{f(T_{min})}$ in Eq. (A2).

| Biomes | Temperature function in $f(T_{lim})$ | $Lim_{min}$ | $Lim_{max}$ | $\partial$ | $\beta$ | $a$ |
|---|---|---|---|---|---|---|
| **Desert tropical** | $f(T_{min})$ | 9.00 | 33.00 | 2.12 | 1.22 | 0.46 |
| **Desert temperate** | $f(T_{min})$ | -14.00 | 28.00 | 2.06 | 1.33 | 0.52 |
| **Tropical semi-arid** | $f(T_{min})$ | 4.00 | 33.00 | 2.27 | 1.57 | 0.52 |
| **Tropical humid** | $f(T_{max})$ | 13.00 | 36 | 1.47 | 1.49 | 0.68 |

In the tropical humid and tropical semi-arid biomes, we applied a saturation function where NPP grows rapidly with increasing precipitation until $NPP_{max}$ is reached, as given in Eq. (A3),

$$g(P_{cum}) = min\left(1, k - \frac{o}{P_{cum}l}\right) \qquad (A3)$$

where $g(P_{cum})$ estimates the cumulative precipitation NPP (relative); $P_{cum}$ is the annual cumulative precipitation; $k$ is the maximum relative NPP (here $NPP_{max}=1$) that limits the growth of the function; $o$ is a constant; $l$ determines the slope of the function and min() limits the linear function to $NPP_{max}=1$. If $P_{cum}$ is 0 mm than $g(P_{cum})$ is set to 0.

In the desert tropical and desert temperate biomes we defined NPP as a simple linear function of precipitation (see Eq. (A4)), which is limited to $NPP_{max}=1$ in order to consider the treatment of precipitation surplus in LPJ-GUESS (Gerten et al., 2004; Smith et al., 2014).

$$g(P_{cum}) = \min(1, mP_{cum}) \qquad\qquad (A4)$$

where $g(P_{npp})$ calculates the cumulative precipitation NPP (relative); $P$ is the annual cumulative precipitation; $m$ is the slope of the linear function; and min() limits the linear function to $NPP_{max}=1$. All parameter values of Eq. (A3) and (A4) are presented in Table A3. For the parameter values determination of the temperature and precipitation functions we randomly halved the biome training subsets (at $[CO_2]$ = 350 ppm) in analysis and validation parts, and then applied nonlinear least-squares model fit in MATLAB® (2015b). We chose the parameter values that yield the lowest root mean square error (RMSE) in the validation part following (Del Grosso et al., 2008).

**Table A3** Parameter values for cumulative precipitation functions in Eq. (A2) for the tropical biomes and Eq. (A3) for the desert biomes.

| Biomes | $k$ | $o$ | $l$ | $m$ |
|---|---|---|---|---|
| **Desert tropical*** | - | - | - | 0.0009 |
| **Desert temperate*** | - | - | - | 0.0014 |
| **Tropical semi-arid** | 1.84 | 4.29 | 0.18 | - |
| **Tropical humid** | 1.24 | 19.69 | 0.51 | - |

\* The asterisk indicates linear precipitation functions

We then combined the climate variable functions and investigated model complexity. We combined $f(T_{min})$, $f(T_{max})$ and $g(P_{cum})$ in seven groupings ranging from one function to multiplying all three climate functions to calculate NPP in each biome. We assessed model complexity with the Bayesian information criterion (BIC) (Burnham and Anderson, 2002; Schwarz, 1978) and model agreement with RMSE and the Wilmott index (DR) (Smith and Smith, 2007; Smith et al., 1997; Willmott et al., 2012). We chose the combinations with lowest BIC and best model agreement. In all biomes the best results were obtained by a combination of precipitation with either one temperature function (because $T_{max}$ and $T_{min}$ are potentially auto-correlated). The combination of $g(P_{cum})$ with $f(T_{max})$ gave the best results in the tropical humid biome while $g(P_{cum})$ combined with $f(T_{min})$ yielded the best results in the remaining biomes (see applied temperature function in Table A2).

In the next step, we combined the selected functions, converted the synergistic function from relative to absolute NPP (kg C m$^{-2}$ yr$^{-1}$) and rescaled the function to independent LPJ-GUESS NPP simulations in order to correct for differences in NPP magnitudes as given in Eq. (A4).

$$NPP_{base} = NPP_{scale}\left(\left(f(T_{lim})\, g(P_{cum})\right) Max_{biome}\right),\ f(T_{lim}) \in [f(T_{max}), f(T_{min})]\ \ (A4)$$

where $NPP_{base}$ is the estimate (kg C m$^{-2}$ yr$^{-1}$) at baseline [$CO_2$]; $f(T_{lim})$ is the temperature function used for the specific biome (either $f(T_{max})$ or $f(T_{min})$ - see Table A2); $Max_{biome}$ is maximum NPP yield of the biome at baseline [$CO_2$] for converting NPP from relative to absolute units; and $NPP_{scale}$ is the scaling factor to minimize the magnitude difference between LPJ-GUESS and BME estimates. The scaling factor is a ratio based on the mean of LPJ-GUESS NPP and the mean of biome meta-model NPP estimates from 1985-2006. In the tropical humid biome $f(T_{min})$ is set to 1 and in the remaining biomes $f(T_{max})$ is set to 1 based on the model complexity analysis. The parameter values are given in Table A4.

**Table A4** Parameter values of the synergistic function in Eq. (A4).

| Biomes | $Max_{biome}$ | $NPP_{scale}$ |
|---|---|---|
| **Desert tropical** | 1.25 | 1.46 |
| **Desert temperate** | 0.86 | 1.05 |
| **Tropical semi-arid** | 1.46 | 1.04 |
| **Tropical humid** | 1.56 | 0.97 |

We implemented the $CO_2$ fertilization effect on plant growth in the final meta-model function (see Eq. (A5)) by applying the same methodology as described in Sallaba et al. (2015) (assuming saturating NPP enhancement with increasing [$CO_2$]) but determined new parameters for each biome using linear fitting in MATLAB® (R2015b). We chose the parameters that yielded lowest RMSE are shown in Table A5.

$$NPP_{scenario} = \left( NPP_{baseline} \left( c \left( 1 - \frac{CO_{2\_baseline}}{CO_{2\_scenario}} \right) + 1 \right) \right) \qquad \text{(A5)}$$

Where $NPP_{scenario}$ is NPP (kg C m$^{-2}$ yr$^{-1}$) under elevated [$CO_2$] (ppm); $NPP_{baseline}$ is modelled NPP at baseline [$CO_2$]; $c$ is the slope; $CO_{2\_baseline}$ is the baseline [$CO_2$] of 350 ppm and $CO_{2\_scenario}$ is an [$CO_2$] > 350 ppm.

**Table A5** Parameter values of the $CO_2$ function in Sallaba et al. (2015) Eq. (5) therein.

| Biomes | $c$ |
|---|---|
| **Desert tropical** | -0.19 |
| **Desert temperate** | -0.63 |
| **Tropical semi-arid** | -0.70 |
| **Tropical humid** | -1.03 |

For each biome, we determined $CO_2$ fertilization function parameter values with a nonlinear least-squares model fit in MATLAB® (R2015b) choosing values yielding the lowest root mean square error (RMSE).

**A.2 Model Evaluation**

**A.2.1 Biome Level Model Validation**

We validate biome-level LPJ-GUESS and BME performance for estimating NPP of natural vegetation with NPP field-measurements from Michaletz et al. (2016) and Luyssaert et al. (2009) (see Sallaba et al., 2015) for the Major Biome Classification of Reich and Eswaran (2002) including the biomes found in the Sahel (desert temperate, tropical semi-arid and tropical humid – no observations were available for desert tropical). Note that since only two observations were available for our study area (see Fig. A1) this evaluation demonstrates the ability of both LPJ-GUESS and BME to replicate NPP for Sahel biomes found elsewhere in the world.

Before we combined the Michaletz et al. (2016) and Luyssaert et al. (2009) datasets, we removed sites with no records of combined above- and below-ground NPP measurements. After we merged the data, we checked the final assembly of NPP measurements for duplicates and removed them. The final dataset consists of 1561 samples (i.e. 1247 samples from Michaletz et al. (2016) and 314 samples from Luyssaert et al. (2009)) representing total NPP measurements across the terrestrial biosphere (sample sizes are 18, 6, and 12 for Sahel biomes of desert temperate, tropical semi-arid and tropical humid, respectively) from 1959-2006. Both LPJ-GUESS and BME were driven with CRU TS 3.21 climate data (Harris et al. 2014, Trenberth et al. 2014) that has global coverage across the time period.

We calculated mean values of the NPP field-measurements and the modelled NPP estimates located in the respective biomes, following Smith et al. (2014b). We aggregated to the biome-level to account for the difference in scale between in situ NPP measurements and modelled grid cell NPP estimates (being grid cell averages).

Finally, we determined the overall model performance, biome-by-biome, with the coefficient of determination ($R^2$ value) and the root mean square error (RMSE). Additionally, we investigated model agreement with performance ratios (hereafter referred to as 'Q') by dividing mean biome NPP estimates (for both models) with mean biome NPP observations. Model overestimation in comparison to in situ NPP measurements is indicated by Q > 1 and underestimation by Q < 1. Good model performance is classified with a Q range between 0.9-1.1 assuming an error of ± 10% following Sallaba et al. (2015). However, we further defined an acceptable model performance error range of ±20% (i.e. Q = 0.8-1.25) given the limitations of using LPJ-GUESS standard modelling protocol, PNV and CRU climate observations, and especially the simplicity of BME.

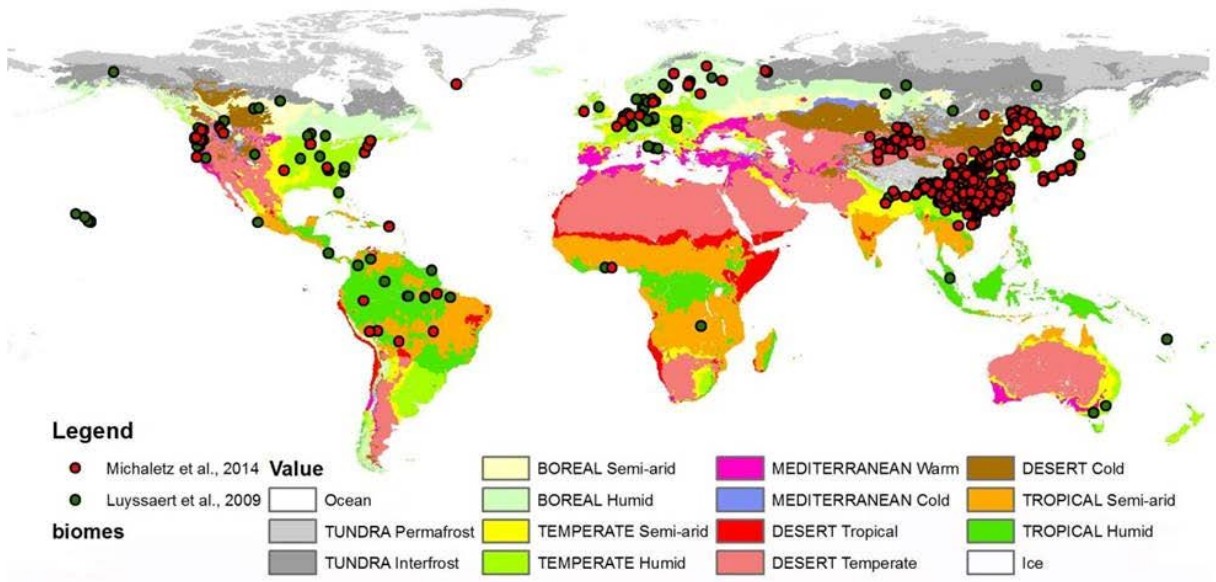

**Fig. A1** Map of the Major Biome Classification based on Reich and Eswaran (2002). The red and green points are the locations of the NPP field-data from Michaletz et al. (2016) and Luyssaert et al. (2009).

5   LPJ-GUESS performs reasonably well in simulating NPP at the overall biome level ($R^2 = 0.71$ and RMSE = 0.16) but the model performance varies notably across the biomes (see Fig. A2 and Table A6). In general, LPJ-GUESS yields acceptable model agreement in seven (with good performance in four biomes) out of thirteen biomes. At the same time, the model underestimates NPP in three biomes while it overestimates NPP in two biomes (Fig. A2).

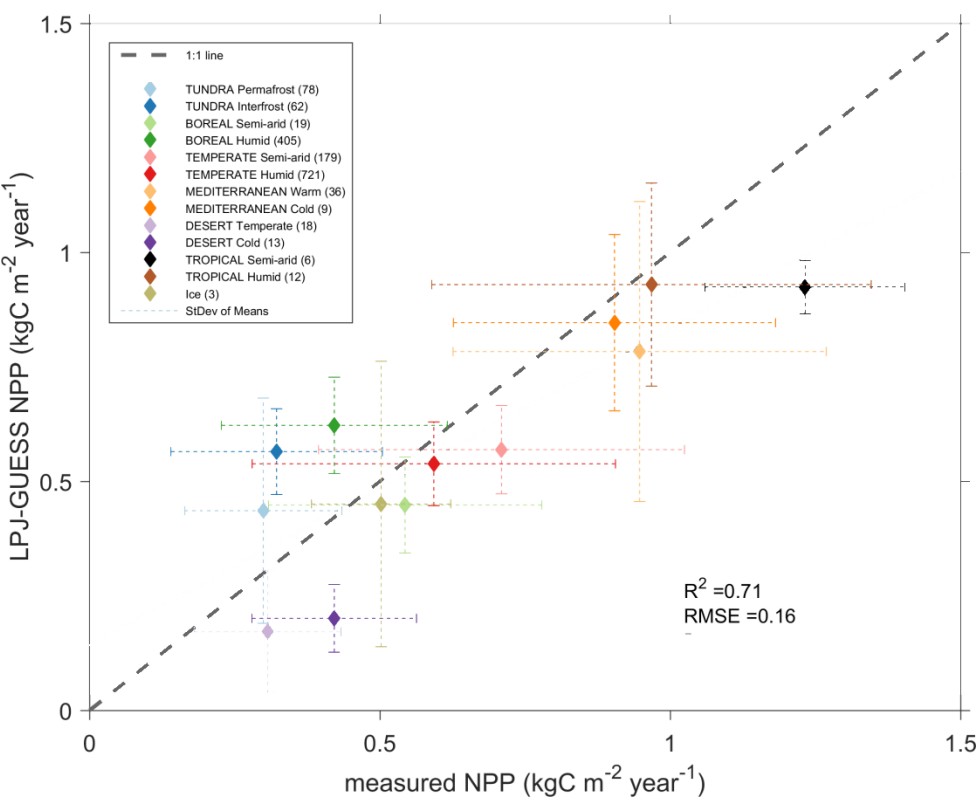

**Fig. A2** Comparison of LPJ-GUESS through NPP estimates and NPP field-measurements at the biome level using biome mean NPP values and their standard deviation. The different colours represent MBC biomes based on (Reich and Eswaran 2002). The number of NPP observations in each biome is given in the legend. Note that Sahel biomes Desert temperate, Tropical Semi-arid, and Tropical Humid.

**Table A6** Comparison between mean biome NPP field-measurements, LPJ-GUESS, BME NPP estimates; and their Q as model performance measure. Sahel biomes are underlined.

| Biome (sample size) | Field-data mean NPP [kg C m$^{-2}$ yr$^{-1}$] | LPJ-GUESS mean NPP [kgC m$^{-2}$ yr$^{-1}$] | LPJ-GUESS Q | BME mean NPP [kgC m$^{-2}$ yr$^{-1}$] | BME Q |
|---|---|---|---|---|---|
| **TUNDRA Permafrost (78)** | 0.30 | 0.44 | 1.46 | 0.24 | 0.79 |
| **TUNDRA Interfrost (62)** | 0.32 | 0.56 | 1.75 | 0.44 | 1.36 |
| **BOREAL Semi-arid (19)** | 0.54 | 0.45 | 0.83 | 0.49 | 0.91 |
| **BOREAL Humid (405)** | 0.42 | 0.62 | 1.48 | 0.56 | 1.32 |
| **TEMPERATE Semi-arid (179)** | 0.71 | 0.57 | 0.80 | 0.45 | 0.63 |
| **TEMPERATE Humid (729)** | 0.59 | 0.54 | 0.91 | 0.56 | 0.95 |
| **MEDITERRANEAN Warm (36)** | 0.95 | 0.78 | 0.83 | 0.52 | 0.55 |
| **MEDITERRANEAN Cold (9)** | 0.90 | 0.85 | 0.94 | 0.41 | 0.45 |
| **DESERT Temperate (18)** | 0.31 | 0.17 | 0.56 | 0.09 | 0.28 |
| **DESERT Cold (13)** | 0.42 | 0.20 | 0.48 | 0.24 | 0.57 |
| **TROPICAL Semi-arid (6)** | 1.23 | 0.92 | 0.75 | 0.84 | 0.68 |
| **TROPICAL Humid (12)** | 0.97 | 0.93 | 0.96 | 0.81 | 0.84 |
| **Ice (3)** | 0.50 | 0.45 | 0.90 | - | - |

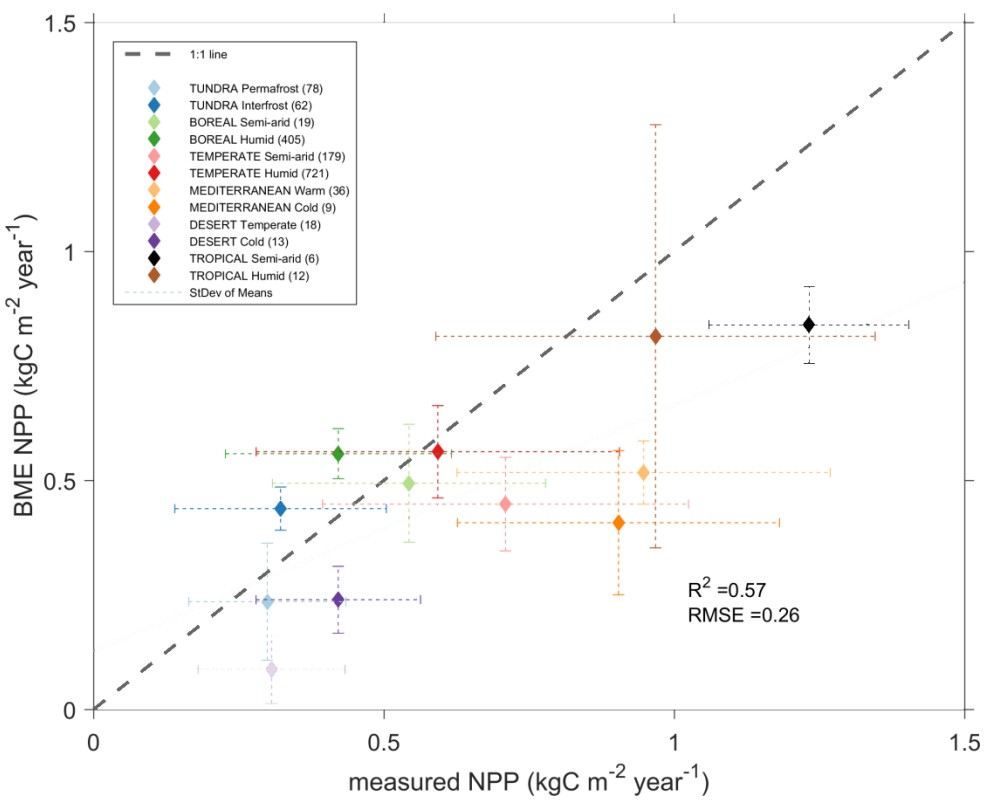

**Fig. A3** Comparison of BME NPP estimates and NPP field-measurements on biome level using biome mean values as well as biome standard deviation of the means. The different colours represent MBC biomes based on (Reich and Eswaran 2002). The number of NPP observations in each biome is given in the legend. Note that Greter Sahel biomes Desert temperate,

5     Tropical Semi-arid, and Tropical Humid.

For Greater Sahel biomes: LPJ-GUESS exhibits good skill in simulating NPP in the Tropical humid (Q = 0.96, see Table A6) where it also captures satisfactorily the variability of the NPP measurements. LPJ-GUESS underestimates NPP for the tropical semi-arid biome (Q = 0.75) showing reduced NPP variation compared to the observations. Performance is reduced

10     for Desert temperate (Q =0.56).

BME performance is acceptable at the overall biome level ($R^2$ = 0.57 and RMSE = 0.26) but varies substantially for individual biomes (see Fig. A3). Overall, BME model agreement is reasonable in four biomes (with good performance in two biomes). At the same time, BME overestimates NPP in two biomes while it underestimates plant growth in six biomes. The variability in in- situ NPP measurements cannot be captured by BME in the majority of biomes except in the tropical

15     humid and tundra permafrost biomes (see vertical and horizontal lines connected to the diamonds in Fig. A3).

For Greater Sahel biomes: BME yields acceptable agreement in estimating NPP in the tropical semi-arid and tropical humid biomes (Q = 0.84, 0.81 respectively) but accuracy drops more water limited biomes of desert temperate (Q = 0.28).

**Overall, BME mimics the behavior of LPJ-GUESS shown by a good model agreement of $R^2$ = 0.71 and moderate RMSE = 0.12 kg C m$^{-2}$ yr$^{-1}$ between the average biome NPP estimates of BME and LPJ-GUESS. Notable is that BME yields, on average, less NPP in the majority of biomes compared to the observations. A.2.2 BME Performance in the Sahel**

For the assessment of BME performance in the Sahel, we chose approximately 4000 CRU TS 3.0 grid cells that cover evenly distributed the Sahel region. We forced LPJ-GUESS with the CRU climate data and measured [$CO_2$] spanning from 1970-2006 and measured [$CO_2$] using the same modeling protocol as described in section A.1). The climate data were post-processed as in section A.1 and then applied to BME in order to estimate NPP. We employed several measures to gauge BME performance against LPJ-GUESS simulations. We calculated the BME's agreement (i.e. precision) with LPJ-GUESS simulations with the coefficient of determination ($R^2$ value) measuring the strength of linear association between the models; the root mean squared error (RMSE) gives the total difference between the models in NPP units (NPP kg C m$^{-2}$ year$^{-1}$) and the Wilmott index (DR) determines how well the plot of LPJ-GUESS simulations and BME NPP fit to a perfect agreement line ranging from -1 to 1 (1 = optimal value) (Smith and Smith, 2007; Smith et al., 1997; Willmott et al., 2012).

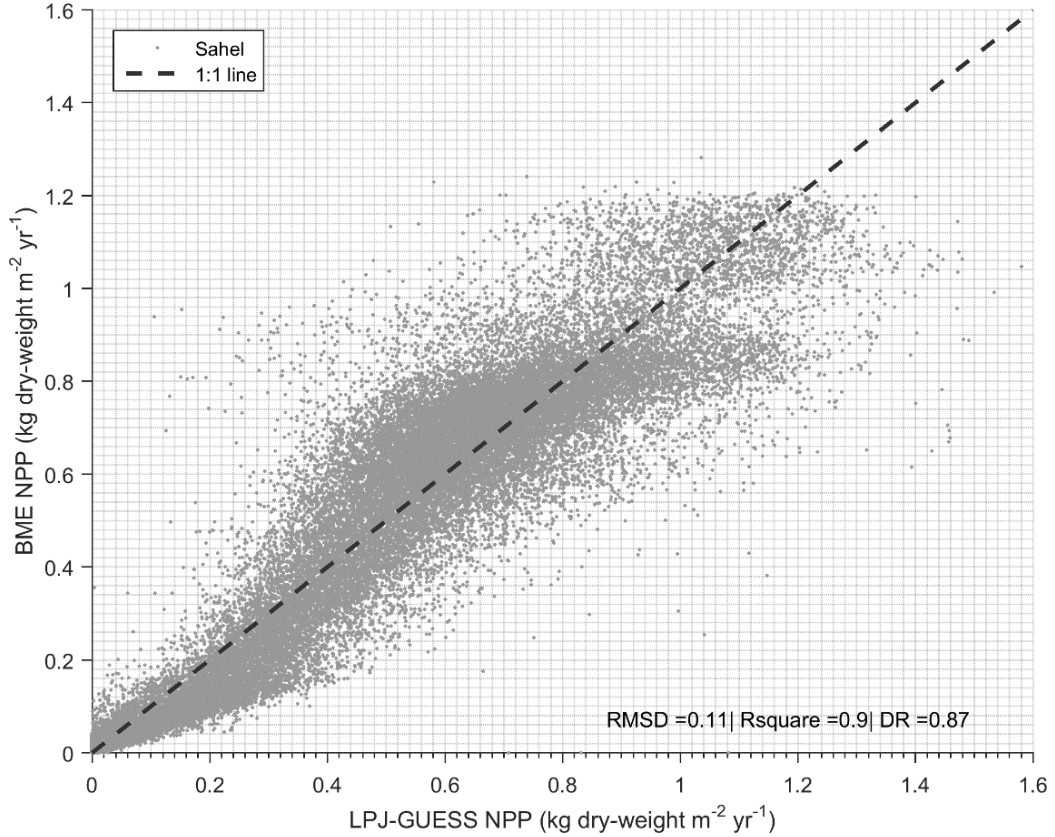

**Fig A4** Comparison between BME and LPJ-GUESS NPP estimates covering the Sahel region.

The comparison between BME and LPJ-GUESS NPP estimates (see Fig. A1) shows a good agreement $R^2$=0.9 and DR=0.87 while the RMSE=0.1 NPP kg DW m$^{-2}$ year$^{-1}$ shows notable total differences between the models.

We then calculate annual means of BME and LPJ-GUESS NPP (i.e. aggregating the entire Sahel region) for the time period in order to investigate whether BME follows the inter-annual variation of LPJ-GUESS NPP. As shown in Fig A5., BME NPP follows the inter-annual variation of LPJ-GUESS NPP. Both models yield depleted NPP in 1972/73 and 1983/84 showing the impact of devastating droughts that occurred in these years resulting in complete crop failure (Ibrahim, 1988). Furthermore, both models yield a dip in NPP in 2002 when the latest major drought befell the region (see Fig. A5) (Balogun

et al., 2013). In Fig. A5, we also include runs from LPJ-GUESS C (carbon cycling only), LPJ-GUESS ml (managed land) and MODIS derived NPP for comparison purposes.

In order to test how effectively the NPP of natural ecosystems can be can be used as a proxy for the NPP of agricultural ones we ran LPJ-GUESS managed land (Olin et al., 2015) for the period 1970 to 2006 and compared this to LPJ-GUESS (used to develop BME) for the entire Sahel region. The results (see Fig. A5) of this experiment show that mean NPP derived from

LPJ-GUESS ml over the region underestimates mean NPP derived from BME by 0.7% (0.006 dry-weight m$^{-2}$ yr$^{-1}$) and LPJ-GUESS by 2.4% (0.020 kg dry-weight m$^{-2}$ yr$^{-1}$), though all models show similar levels of interannual variability and trend (see Fig. A5). The implication of this experiment is that there is a demonstrable reduction in NPP when land management is taken into consideration, but the effect is relatively minor. Lindeskog et al. (2013) show that LPJ-GUESS managed land (C-version) overestimated actual yield derived from FAO country-level crop statistics and Smith et al. (2014b) also report that

natural systems are more productive than agricultural systems in sub-Saharan Africa. We conclude with that possibility that our results are in the upper range for NPP found in the Sahel.

We also compare total yearly means of NPP from BME and LPJ-GUESS to NPP derived from the MOD17A3 processing stream (using MOD17A3 data obtained from the NASA Earth Observation System repository at the University of Montana at www.ntsg.umt.edu) for the period 2000 to 2006 for the greater Sahel region (Running, 2004). We averaged resampled

MODIS NPP from 1km to the spatial resolution of the BME estimates (0.5 x 0.5 degrees) and excluded urban areas. We removed below-ground NPP and plant parts unable to be consumed by applying the same R:S and harvest index as described in Section 2.1.1. Lastly, we calculated mean values of MODIS NPP estimates from 2000 to 2010 for each grid cell covering the study area. Our results show that between 2000 and 2006 MODIS-derived NPP underestimate BME-derived NPP by 42% (difference of 0.38 kg dry-weight m$^{-2}$ yr$^{-1}$), on average (Figure A5). Ardö (2015) also reports that that average annual

MODIS NPP underestimates LPJ-GUESS (C version only, Fig. A5) for Africa for 2000-2010 and attributes this to the fact that autotrophic respiration is considerably higher for MODIS NPP compared to LPJ-GUESS, due to large temperature sensitivity in the MODIS algorithm, differences in the biome-specific parameterizations for MODIS as well as specification of plant functional types in LPJ-GUESS.

Country-level census yield trends (1989-2008) for 4 major crops from appendix Data S1 of Ray et al. (2013) for rice (Benin, Burkina Faso, Chad, Ghana, Guinea, Guinea-Bissau, Ivory Coast, Liberia, Mali, Nigeria, Senegal, Sierra Leone, Togo), maize (Benin, Burkina Faso, Cameroon, Chad, Ethiopia, Ghana, Guinea, Ivory Coast, Mali, Nigeria, Senegal, Togo), wheat (Cameroon, Chad, Eritrea, Ethiopia, Mali, Mauritania, Niger, Nigeria, Sudan) and soybean (Benin, Burkina Faso, and Nigeria) range from -5.98 to 2.80 (mean of -0.002), -0.94 to 4.08 (mean of 1.400), -2.58 to 3.1 (mean of 1.280) and 1.15 to 3.98 (mean of 2.280) respectively. Trends for BME, LPJ-GUESS, and MODIS NPP fall within most of the ranges for crop yield trends, showing yearly increases of 0.55% (BME), 0.58% (LPJ-GUESS), and 0.51% (MODIS) for the 7 year period of overlap. For the entire length of each series (1970-2006 for BME and LPJ-GUESS and 2000-2010 for MODIS), slopes indicate yearly increases of 0.40%, 0.40%, and 0.62% respectively. We note the number of uncertainties involved in this comparison (e.g. spatial/temporal sampling, and the fact that BME and MODIS represent natural vegetation and a mix of natural vegetation and crops, respectively).

### A.2.3 Concluding Remarks for Model Validation and Evaluation

In sum, a validation involving ground measurements for the same biomes found in the Sahel (but observations mostly from other locations) show that LPJ-GUESS and BME underestimate NPP, while a comparison with MODIS shows that LPJ-GUESS (and BME) overestimate total mean annual NPP in the greater Sahel region (2000-2006). Yet is widely acknowledged, natural systems are likely more productive than agricultural systems. But we also show that trends for BME, LPJ-GUESS, and MODIS mostly fall within trend ranges for country-level yield statistics (though sample size is low). We acknowledge that the uncertainties are significant. Differences in estimates between methods are due to a combination of spatial aggregation/sampling issues (e.g. low sample sizes for biomes typically found in the Sahel, that CRU data do not necessarily represent site-level climate, and the uncertain assessment below-ground and short-lived above-ground plant matter at the site level) as well differing assumptions between the MODIS processing stream and LPJ-GUESS (particularly respiration). We therefore conclude that BME and LPJ-GUESS replicate ground observations of NPP at similar orders of magnitude at the biome level, but may be overestimated due to the fact that natural systems are usually more productive than agricultural ones. This underscores the fact that BME and LPJ-GUESS should be restricted to biome-level applications (or coarser) while applications on the grid cell level should be limited to explorations of patterns and trends, which is the reason why we emphasize an aggregated level of analysis.

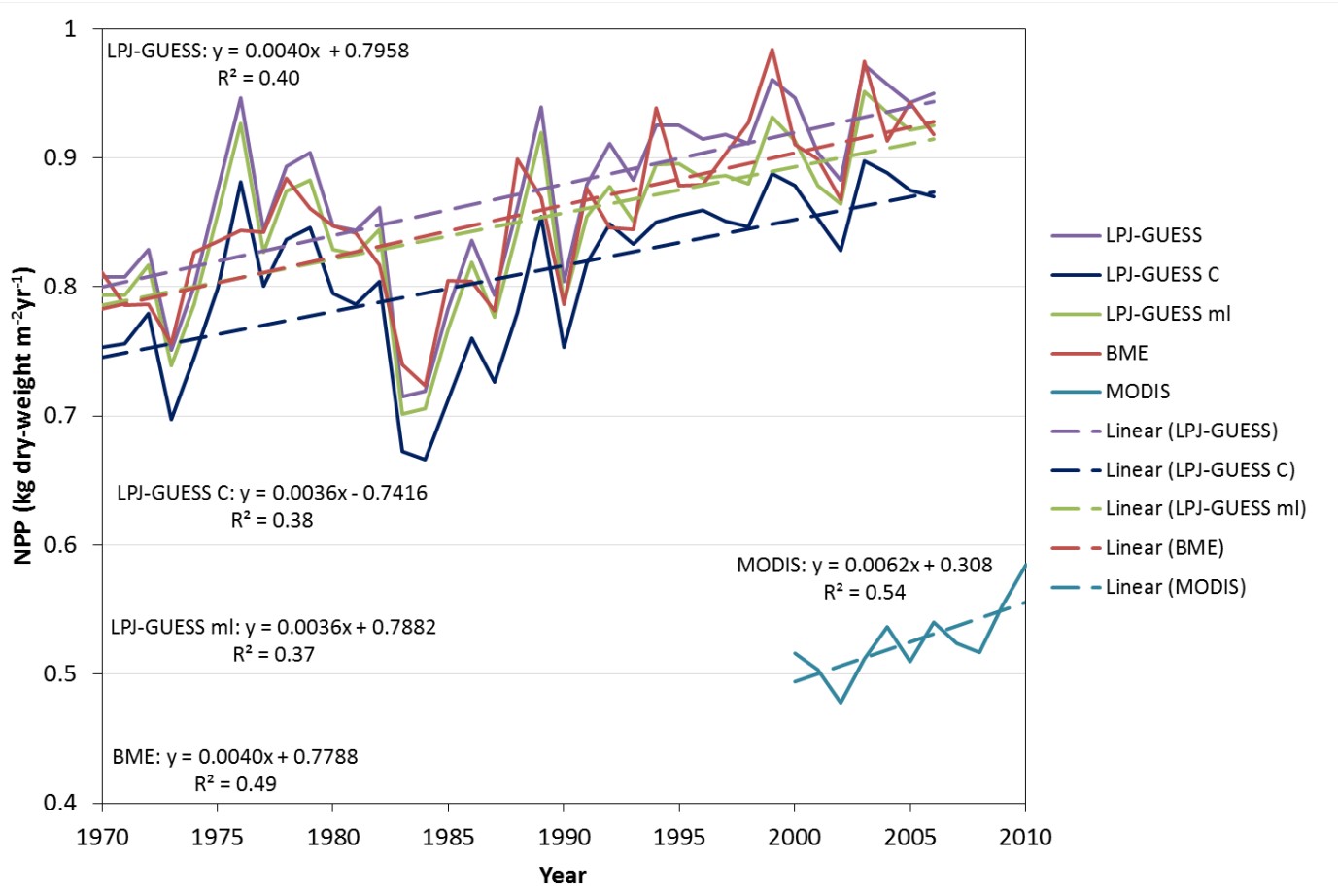

**Fig A5** Regional annual NPP Annual means of NPP for BME, LPJ-GUESS, LPJ-GUESS C (carbon only), LPJ-GUESS ml (managed land) (1970 to 2006) and MODIS (2000-2010) for the greater Sahel region.

### A.3 Estimation of NPP supply and demand

In this modelling framework, we followed the $NPP_{demand}$ definition Abdi et al. (2014) as given in Eq. (A6).

$$NPP_{demand} = NPP_{food} + NPP_{feed} + NPP_{residues} + NPP_{fuel} + NPP_{burned} \quad \text{(A6)}$$

Where $NPP_{demand}$ is the actual amount of annual NPP needed for human survival; $NPP_{food}$ is the NPP needed for consumed cereals, meat and milk production; $NPP_{feed}$ is the total amount of fodder to support the livestock population and $NPP_{residues}$ are agricultural by-products (after harvesting); $NPP_{fuel}$ describes fuelwood and charcoal from the region's dry woodlands and $NPP_{burned}$ represents the human-driven NPP loss from biomass burning of forest resources for land clearing due to land use change (Abdi et al., 2014).

We adapted Eq. (A6) to the current study's framework by dividing the demand into cereal (Eq. A7) and grazing (Eq. A8) based NPP, and PLUM outputs.

$$NPP_{cereal\_demand} = NPP_{food} + NPP_{feed} \qquad\qquad (A7)$$

where $NPP_{food} = cereal_{total} - cereal_{feed}$ (ton country$^{-1}$); $cereal_{total}$ (ton country$^{-1}$) is the total cereal consumption of human and livestock population provided by PLUM; $cereal_{feed}$ (ton country$^{-1}$) is the total cereal demand to sustain the livestock population (a direct PLUM ouput ); $NPP_{feed}$ (ton country$^{-1}$) is equal with $cereal_{feed}$; We then converted then $NPP_{cereal\_demand}$ to per capita (kg person$^{-1}$) using country population of the corresponding year in the SSP.

The amount of NPP needed to sustain the livestock by grazing that cannot be covered with $cereal_{feed}$ we applied Eq. (A7).

$$NPP_{grazing\_demand} = (1 - feed_{ratio}) * cereal_{feed}/feed_{ratio} \qquad\qquad (A8)$$

Where $NPP_{grazing\_demand}$ (ton country$^{-1}$) is the NPP obtained from grasslands for sustaining the livestock; $feed_{ratio}$ ranges between 0-1 (given by PLUM) and provides the proportion of how much $cereal_{feed}$ can meet the livestock demand of energy needed to sustain the livestock. Furthermore, we assumed that the Sahelian livestock is kept close to human populated areas and we therefore we converted $NPP_{grazing\_demand}$ to per capita (kg person$^{-1}$) using country population of the corresponding year in the SSP.

Furthermore, we eliminated $NPP_{fuel}$ in Eq. (A6) because we assumed that fuelwood doesn't directly contribute to the availability of food resources. Fuelwood is a vital variable since it is a necessity for processing cereals and meat but it cannot provide information about food resource availability. Moreover, we eliminated $NPP_{burned}$ in Eq. (A6) since it cannot be counted as an actual food resource in the particular year where the land-clearances occurs but it is an important indirect factor, determining how much food can be produced in the following years.

**Appendix B Figures**

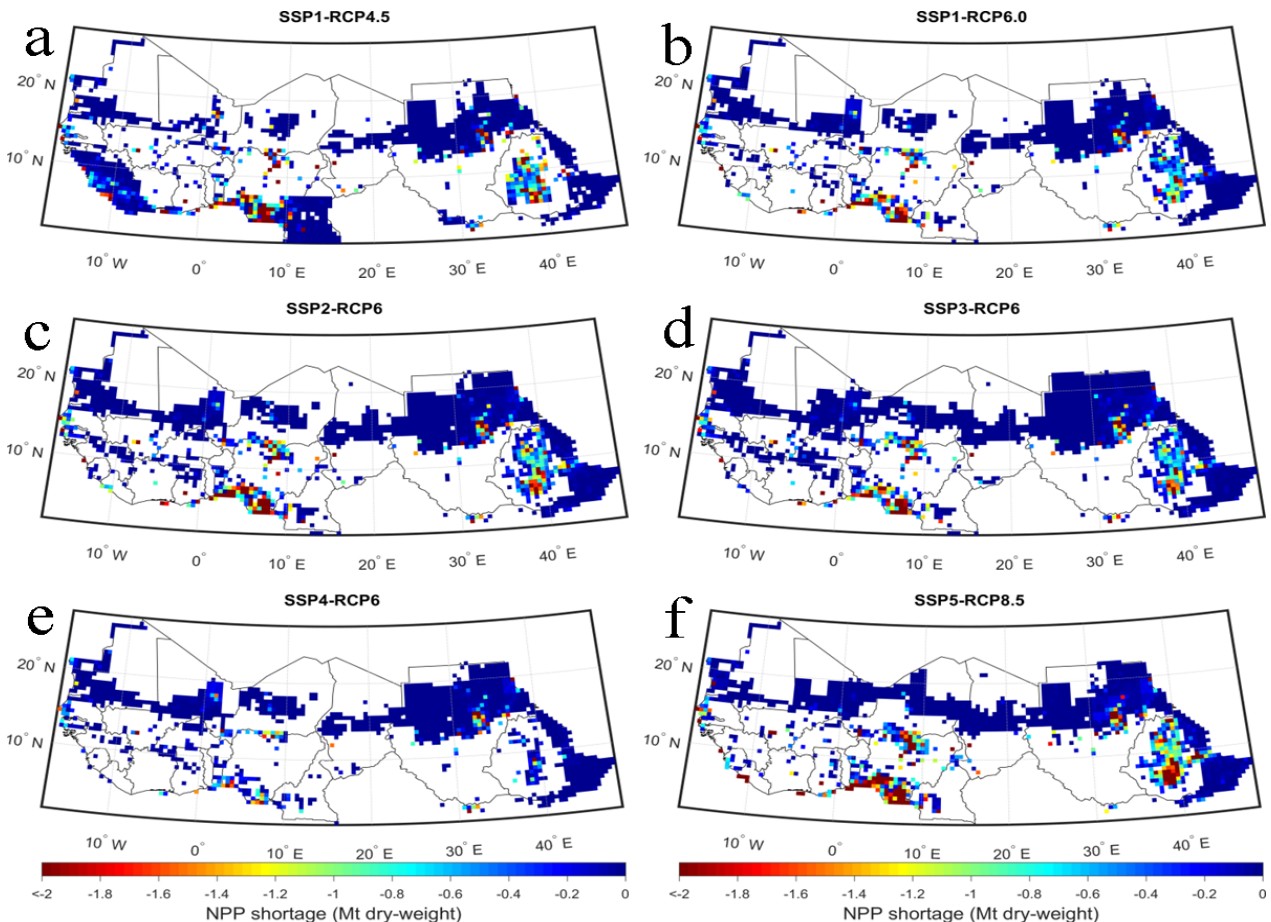

**Fig. B1** Spatial distribution of NPP shortage in 2050 for the six most likely SSP-RCP combinations.

The future socio-economic and climatic scenarios are ordered in the panels as following: a) SSP1-RCP4.5, b) SSP1-RCP6.0,

c) SSP2-RCP6.0, d) SSP3-RCP6.0, e) SSP4-RCP6.0 and f) SSP5-RCP8.5.

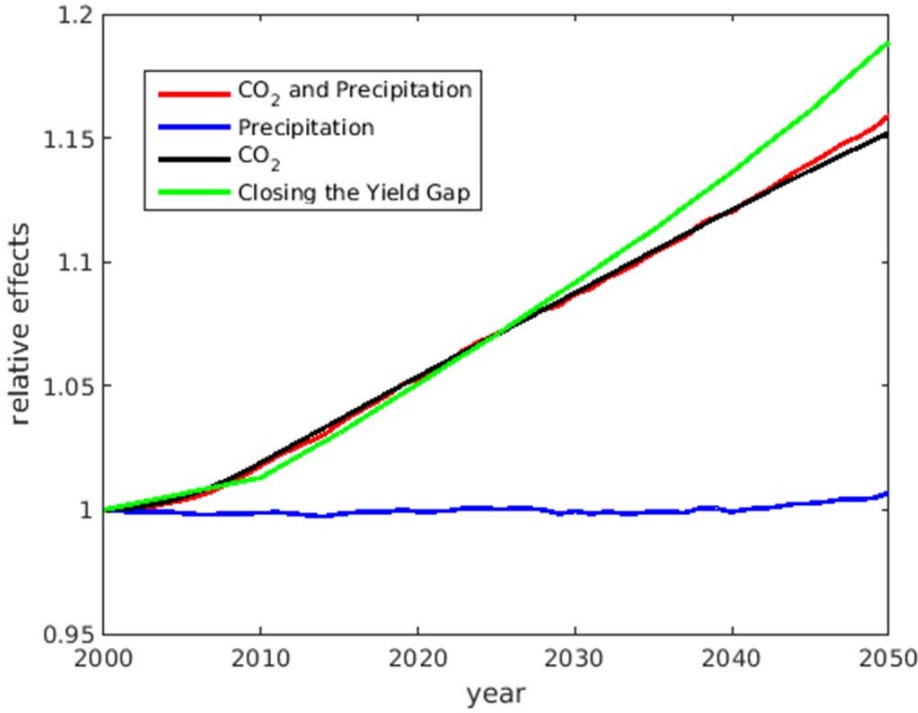

**Fig. B2**. The relative contributions of $CO_2$, precipitation and yield gap closure to the increase in NPP over the greater Sahel region, 2000-2050. Results for $CO_2$ and precipitation are from RCP 6.0 and yield gap is from SSP2. Simulated climate and $CO_2$ effects shown here are mean effects over the five GCMs (GFDL,MIROC,Hadley,NorESM, IPSL).

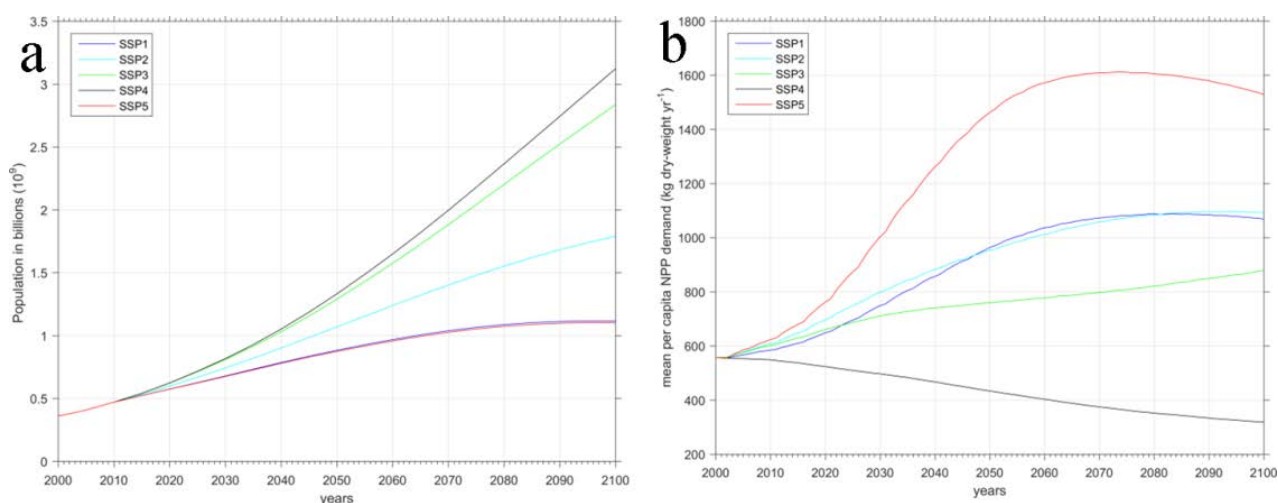

**Fig. B3** a) population growth scenarios of the greater Sahel region and b) mean per capita demand of Sahelian countries

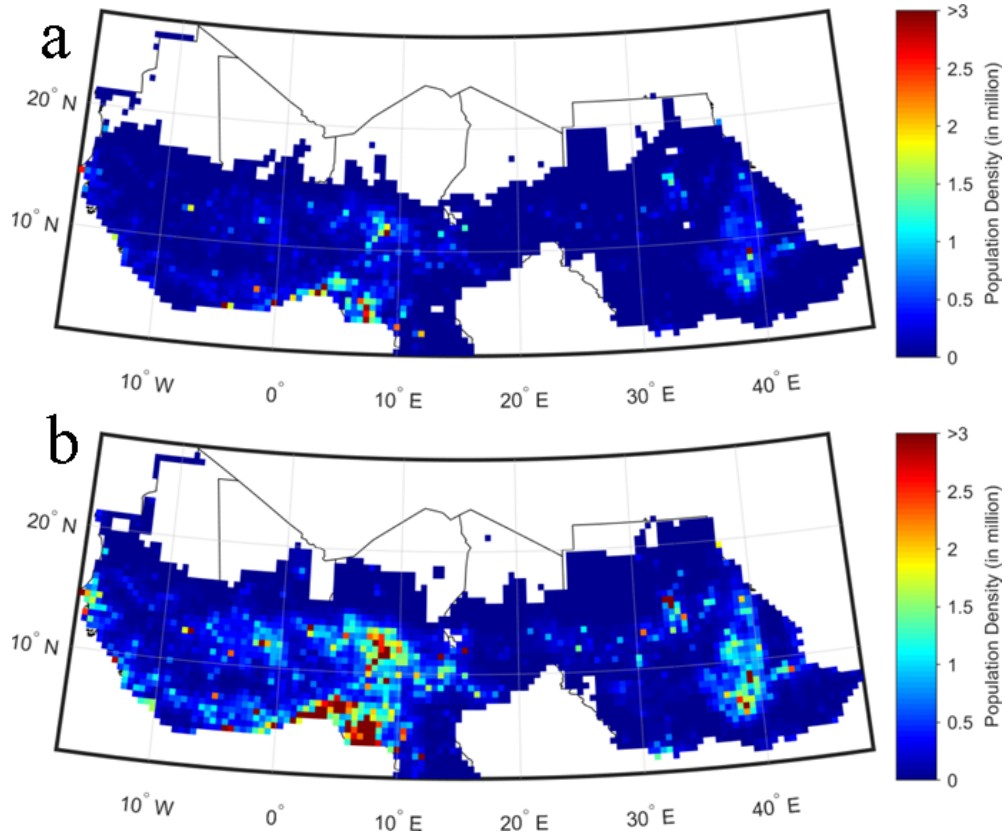

**Fig. B4** Distribution of population for SSP2-RCP6.0 for the years a) 2000 and b) 2050. Grid cells with less than one person per km² are excluded.

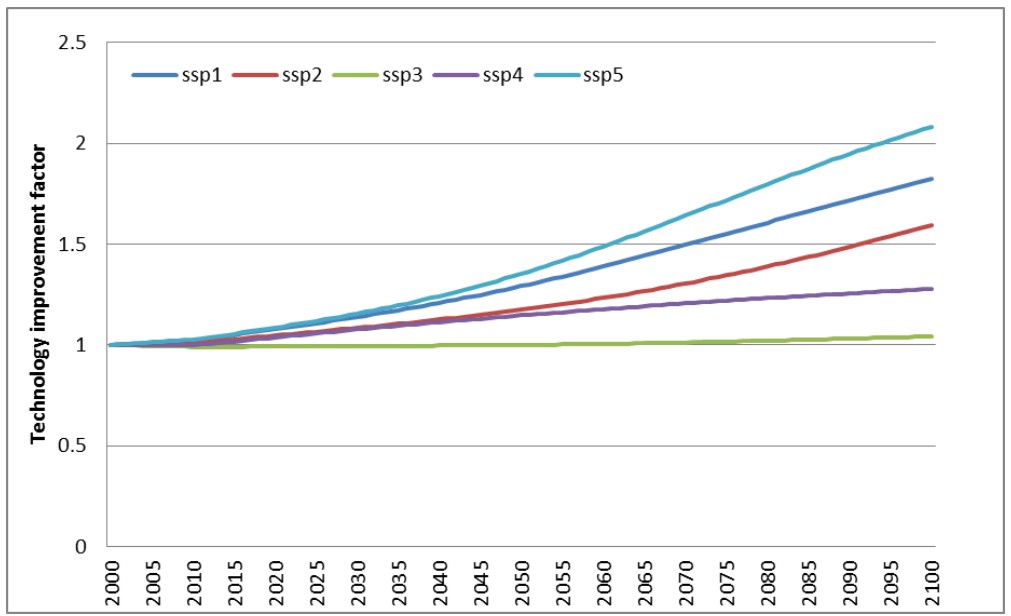

**Fig. B5** Development of mean technology improvement factor for all countries for the socio-economic pathways.

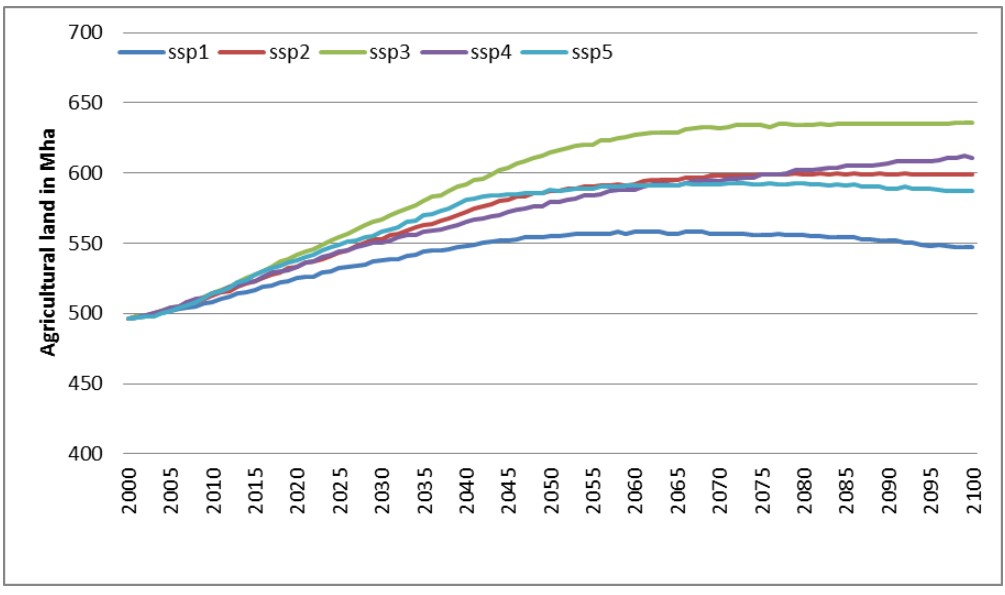

**Fig. B6** Expansion of total agricultural land, including grass- and cropland, in the Sahel for the socio-economic pathways

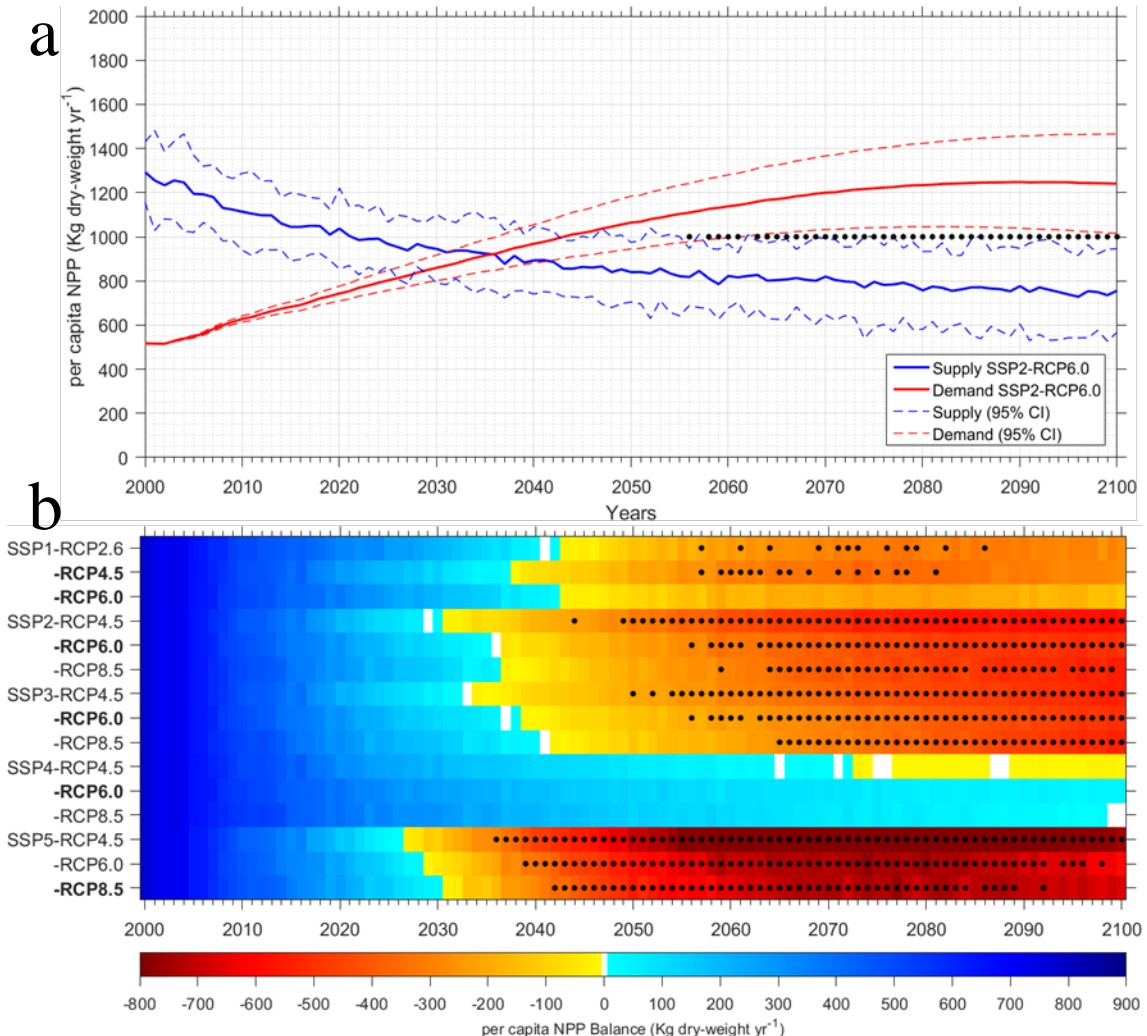

**Fig. B7** Per capita NPP supply, demand and balance for the greater Sahel (2000-2100) without $CO_2$ fertilization. **B7a)** shows NPP supply (red) and demand (blue). The solid curves illustrate the mean of the SSP2-RCP6.0 combination. The dashed blue curves show supply uncertainty (95% confidence interval around the mean) based on the five GCMs NPP results. The dashed red curves show demand uncertainty (95% confidence interval around the mean) based on the uncertainty related to the interpretation and quantification of SSP2. **B7b)** shows the different magnitudes of the NPP balance and the varying onsets of shortage across all SSP-RCP combinations. Black dots illustrate years with a shortage outside of the 95% confidence intervals. Combinations are grouped according to the socio-economic scenarios (y-axis). The RCPs are ordered from low to high radiative forcing in each SSP group. The temporal trajectory is shown along the x-axis and the colouring indicates the sign of the annual NPP balance. Blues show a surplus of the NPP supply while yellow to red represent small to very large the gaps between supply and demand). SSP-RCP combinations in bold indicate the most likely SSP-RCP pairs based on Table 1.

## Appendix C Tables

**Table C1** Per capita NPP supply and demand of countries in the greater Sahel region for 2000 and 2050. Portions of food and feed (including grazing) in per capita NPP demand for SSP2-RCP6.0. All NPP is given in dry-weight (DW). Hurtt:PLUM scaling factors and land areas (from FAO) are also included.

| Country | Per capita NPP supply [kg DW yr$^{-1}$] | | Per capita NPP demand [kg DW yr$^{-1}$] | | Food portions in per capita NPP demand [kg DW yr$^{-1}$] | | Feed portions in per capita NPP demand[kg DW yr$^{-1}$] | | Hurtt:PLUM scaling factors | Land Area from FAOSTAT |
|---|---|---|---|---|---|---|---|---|---|---|
| | 2000 | 2050 | 2000 | 2050 | 2000 | 2050 | 2000 | 2050 | 2000 | 1000 ha |
| **Benin** | 1341 | 607 | 474 | 874 | 99 | 92 | 375 | 782 | 0.89 | 11062 |
| **Burkina Faso** | 933 | 316 | 196 | 169 | 196 | 169 | 0 | 0 | 0.90 | 27360 |
| **Cameroon** | 2127 | 1173 | 387 | 717 | 90 | 82 | 297 | 635 | 1.04 | 47271 |
| **Chad** | 1878 | 1484 | 658 | 1157 | 120 | 116 | 538 | 1041 | 1.00 | 125920 |
| **Djibouti** | 0 | 0 | 134 | 120 | 134 | 120 | 0 | 0 | 0.00 | 2318 |
| **Eritrea** | 333 | 221 | 124 | 130 | 124 | 130 | 0 | 0 | 1.10 | 10100 |
| **Ethiopia** | 825 | 779 | 459 | 1439 | 135 | 157 | 323 | 1283 | 0.98 | 1000000 |
| **Gambia** | 1137 | 632 | 706 | 1082 | 168 | 142 | 539 | 940 | 1.58 | 1000 |
| **Ghana** | 1490 | 1291 | 274 | 1080 | 68 | 67 | 207 | 1013 | 1.03 | 22754 |
| **Guinea** | 1773 | 1697 | 402 | 1066 | 123 | 87 | 279 | 979 | 1.73 | 24572 |
| **Guinea Bissau** | 2319 | 1648 | 599 | 934 | 144 | 118 | 455 | 816 | 1.25 | 2812 |
| **Ivory Coast** | 1795 | 1549 | 282 | 811 | 95 | 75 | 188 | 736 | 0.98 | 31800 |
| **Liberia** | 1186 | 1312 | 212 | 1273 | 91 | 109 | 121 | 1164 | 0.91 | 9632 |
| **Mali** | 1929 | 1191 | 1111 | 1272 | 191 | 170 | 920 | 1102 | 0.97 | 122019 |
| **Mauritania** | 1129 | 1043 | 1530 | 1555 | 151 | 140 | 1379 | 1415 | 0.97 | 103070 |
| **Niger** | 3437 | 1426 | 1274 | 1540 | 210 | 202 | 1064 | 1338 | 1.01 | 126670 |
| **Nigeria** | 1059 | 719 | 321 | 923 | 139 | 139 | 182 | 784 | 1.04 | 91077 |
| **Senegal** | 925 | 539 | 556 | 837 | 155 | 137 | 401 | 699 | 0.74 | 19253 |
| **Sierra Leone** | 759 | 949 | 194 | 767 | 117 | 125 | 77 | 642 | 0.99 | 7162 |
| **Sudan** | 2517 | 1512 | 1530 | 1609 | 126 | 118 | 1404 | 1491 | 0.98 | 237600 |
| **Togo** | 2171 | 1491 | 271 | 653 | 127 | 124 | 144 | 529 | 1.10 | 5439 |
| **Mean**[1] | 1377 | 957 | 517 | 1064 | - | - | - | - | - | - |

5  [1] Weighted mean of per capita NPP measure using total population.

**Table C2** Per capita NPP balances, mean, standard deviation and coefficient of variation for all SSP-RCP combinations for the year 2050. All values are given in Kg NPP dry weight yr-1 except the coefficient of variation, which is dimensionless.

| Country | SSP1 | | | SSP2 | | | SSP3 | | | SSP4 | | | SSP5 | | | Summary statistics | | |
|---|---|---|---|---|---|---|---|---|---|---|---|---|---|---|---|---|---|---|
| | RCP2.6 | RCP4.5 | RCP6.0 | RCP4.5 | RCP6.0 | RCP8.5 | RCP4.5 | RCP6.0 | RCP8.5 | RCP4.5 | RCP6.0 | RCP8.5 | RCP4.5 | RCP6.0 | RCP8.5 | Mean | Std | CoV |
| **Benin** | -288 | -264 | -290 | -256 | -267 | -247 | -95 | -97 | -84 | 148 | 118 | 150 | -945 | -968 | -939 | -288 | 364 | 1.3 |
| **Burkina Faso** | 261 | 257 | 258 | 147 | 147 | 152 | 78 | 78 | 82 | 93 | 95 | 98 | 244 | 244 | 251 | 166 | 75 | 0.5 |
| **Cameroon** | 604 | 134 | 627 | 47 | 456 | 526 | 91 | 439 | 500 | 258 | 621 | 679 | -180 | 285 | 365 | 363 | 244 | 0.7 |
| **Chad** | 327 | 311 | 442 | 239 | 326 | 367 | 283 | 312 | 336 | 369 | 423 | 430 | -13 | 205 | 314 | 311 | 108 | 0.3 |
| **Djibouti** | -117 | -117 | -117 | -120 | -119 | -120 | -119 | -119 | -119 | -101 | -101 | -101 | -128 | -128 | -128 | -117 | 9 | 0.1 |
| **Eritrea** | 113 | 142 | 138 | 94 | 91 | 87 | 69 | 67 | 64 | 75 | 73 | 69 | 121 | 117 | 111 | 95 | 26 | 0.3 |
| **Ethiopia** | -498 | -642 | -444 | -804 | -660 | -616 | -560 | -443 | -410 | -33 | 90 | 98 | -1493 | -1300 | -1229 | -596 | 455 | 0.8 |
| **Gambia** | -464 | -439 | -424 | -465 | -449 | -472 | -328 | -309 | -328 | -75 | -57 | -69 | -842 | -829 | -851 | -427 | 251 | 0.6 |
| **Ghana** | 289 | 362 | 437 | 151 | 211 | 226 | 253 | 297 | 302 | 488 | 513 | 506 | -202 | -122 | -89 | 241 | 218 | 0.9 |
| **Guinea** | 915 | 675 | 968 | 403 | 631 | 644 | 395 | 603 | 611 | 584 | 808 | 754 | 347 | 633 | 646 | 641 | 170 | 0.3 |
| **Guinea Bissau** | 910 | 993 | 913 | 801 | 714 | 811 | 700 | 679 | 734 | 708 | 688 | 711 | 822 | 580 | 742 | 767 | 104 | 0.1 |
| **Ivory Coast** | 947 | 873 | 1044 | 586 | 737 | 773 | 513 | 645 | 680 | 700 | 780 | 785 | 546 | 734 | 780 | 742 | 138 | 0.2 |
| **Liberia** | 52 | -1336 | 211 | -1273 | 39 | 32 | -686 | 292 | 290 | -204 | 358 | 387 | -2774 | -934 | -942 | -433 | -859 | 2.0 |
| **Mali** | -113 | -123 | -24 | -147 | -81 | 27 | -103 | -80 | 4 | -48 | 12 | 83 | -451 | -305 | -124 | -98 | 129 | 1.3 |
| **Mauritania** | -474 | -475 | -505 | -502 | -512 | -527 | -454 | -449 | -453 | -382 | -406 | -424 | -640 | -663 | -696 | -504 | 90 | 0.2 |
| **Niger** | 117 | 183 | 247 | -151 | -114 | -148 | -290 | -271 | -303 | -7 | 58 | 42 | -315 | -246 | -265 | -98 | 184 | 1.9 |
| **Nigeria** | -95 | -150 | -77 | -267 | -204 | -165 | -132 | -84 | -52 | 135 | 162 | 184 | -663 | -581 | -535 | -168 | 248 | 1.5 |
| **Senegal** | -119 | -134 | -202 | -240 | -297 | -234 | -205 | -249 | -193 | -25 | -51 | -26 | -421 | -508 | -429 | -222 | 141 | 0.6 |
| **Sierra Leone** | 277 | -728 | 334 | -767 | 183 | 176 | -493 | 217 | 215 | -173 | 252 | 260 | -1376 | -115 | -95 | -122 | 487 | 4.0 |
| **Sudan** | -44 | -49 | 10 | -144 | -97 | -91 | -177 | -139 | -108 | -30 | 14 | -7 | -281 | -211 | -227 | -105 | 87 | 0.8 |
| **Togo** | 916 | 998 | 1022 | 827 | 838 | 842 | 794 | 788 | 789 | 870 | 887 | 894 | 547 | 568 | 581 | 811 | 139 | 0.2 |
| **Mean**[1] | 1 | -84 | 47 | -212 | -107 | -76 | -119 | -39 | -12 | -131 | 200 | 213 | -570 | -428 | -384 | -96 | 217 | 2.3 |

[1]Weighted mean using national population data as weight.

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

**Figures**

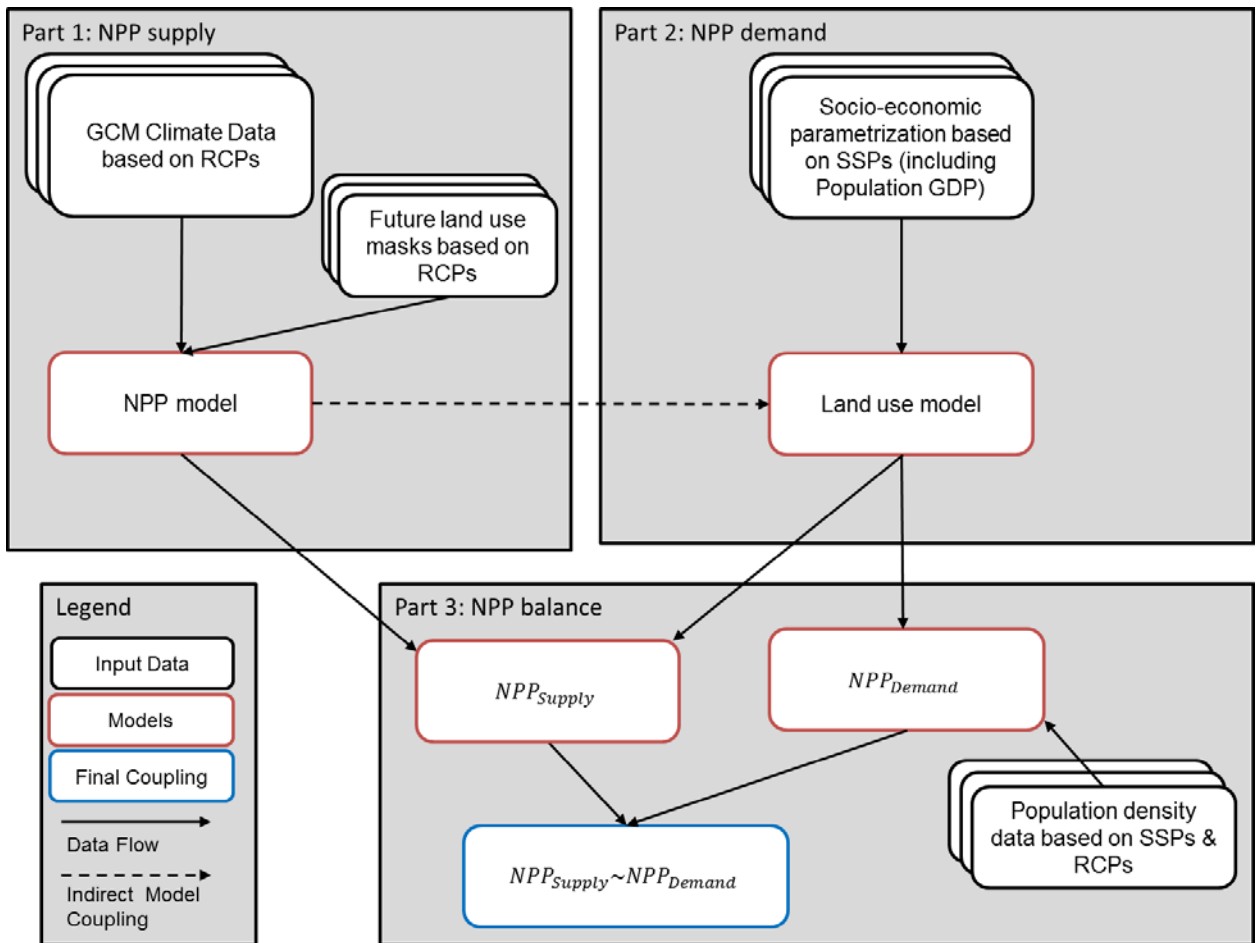

5    **Fig. 1** Conceptual logic of the modelling framework. The framework is based on three components enclose by three grey

boxes: (1) $NPP_{supply}$, (2) $NPP_{demand}$ and (3) $NPP_{balance}$. The white boxes indicate data inputs originating from

modelling studies (as referenced in section 2.2). The main models and equations are given in the boxes outlined in red,

where solid arrows show the data flow. The dashed arrow between *NPP model* (section 2.1.1) and *Land use model* (section

2.1.2) represents an indirect model coupling for areas of cropland and pasture. The box outlined in blue indicates the final

10    coupling allowing the assessment of $NPP_{supply}$ and $NPP_{demand}$.

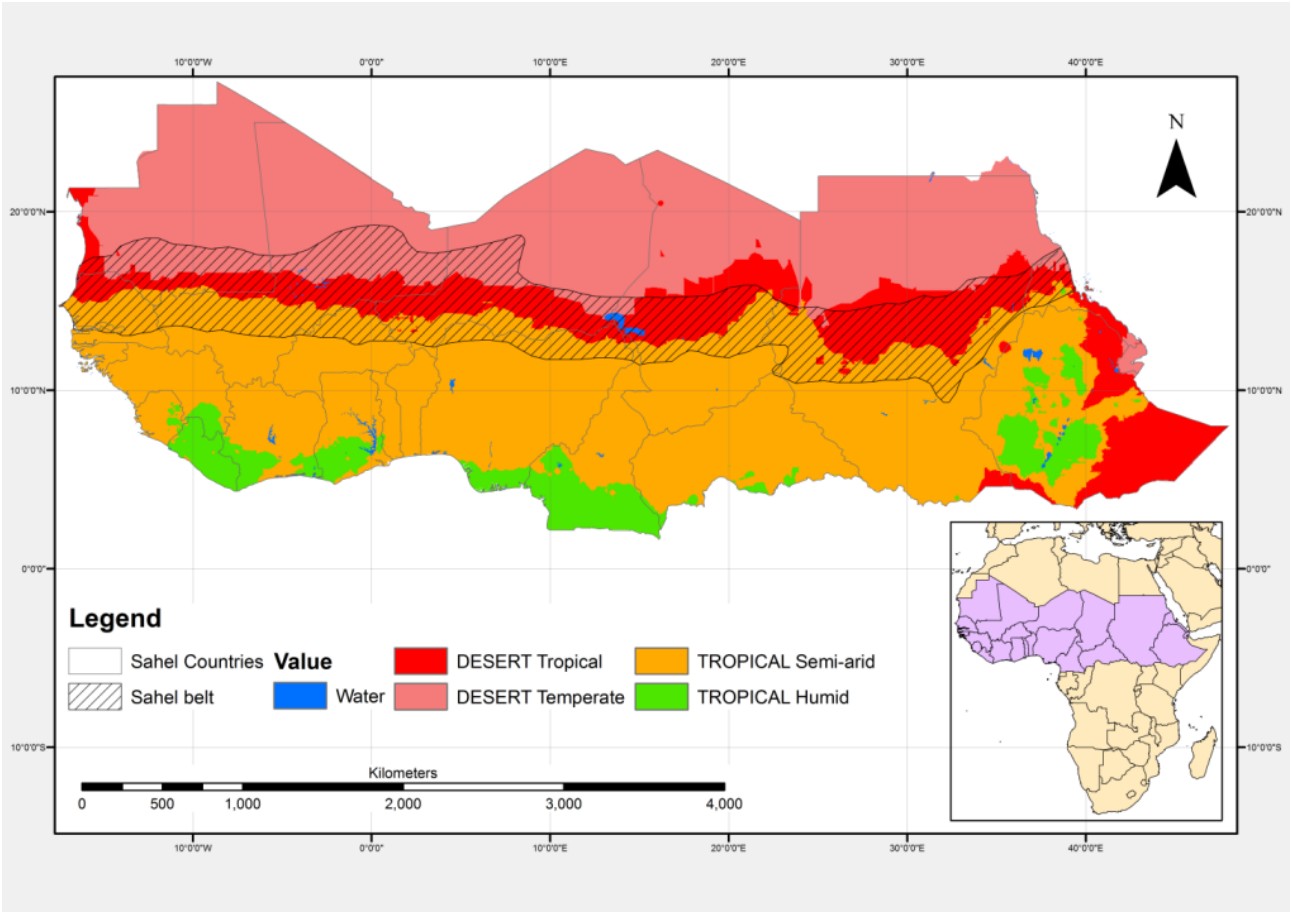

**Fig. 2** Major Biome Map from year 2000 for greater Sahel region. The hatched area shows the traditionally-defined Sahel, where annual rainfall ranges from 100mm to 600mm. The Major Biome Map is based on Reich and Eswaran (2002).

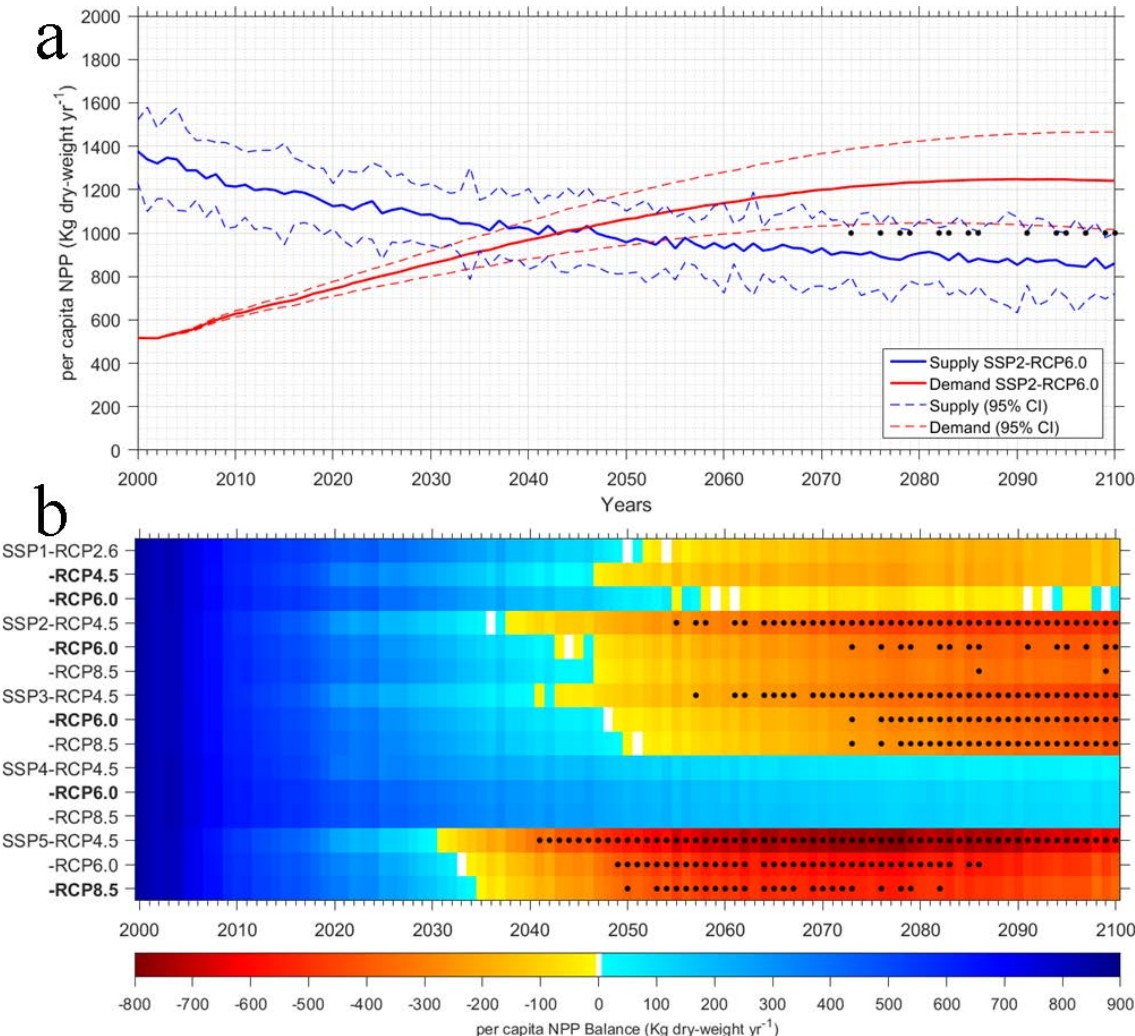

**Fig. 3** The per capita NPP supply, demand and balance for the entire Sahel region over the time period. **3a)** shows NPP supply (red) and demand (blue). The solid curves illustrate the mean of the SSP2-RCP6.0 combination. The dashed blue curves show supply uncertainty (95% confidence interval around the mean) based on the five GCMs NPP results. The dashed red curves show demand uncertainty (95% confidence interval around the mean) based on the uncertainty related to the interpretation and quantification of SSP2. **3b)** shows the different magnitudes of the NPP balance and the varying onsets of shortage across all SSP-RCP combinations. Black dots illustrate years with a shortage outside of the 95% confidence intervals. The combinations are grouped according to the socio-economic scenarios (y-axis). The RCPs are ordered from low to high radiative forcing in each SSP group. The temporal trajectory is shown along the x-axis and the colouring indicates the sign of the annual NPP balance. Blues show a surplus of the NPP supply while yellow to red represent small to very large NPP shortages (i.e. the gap between supply and demand). SSP-RCP combinations in bold indicate the most likely SSP-RCP pairs based on Table 1.

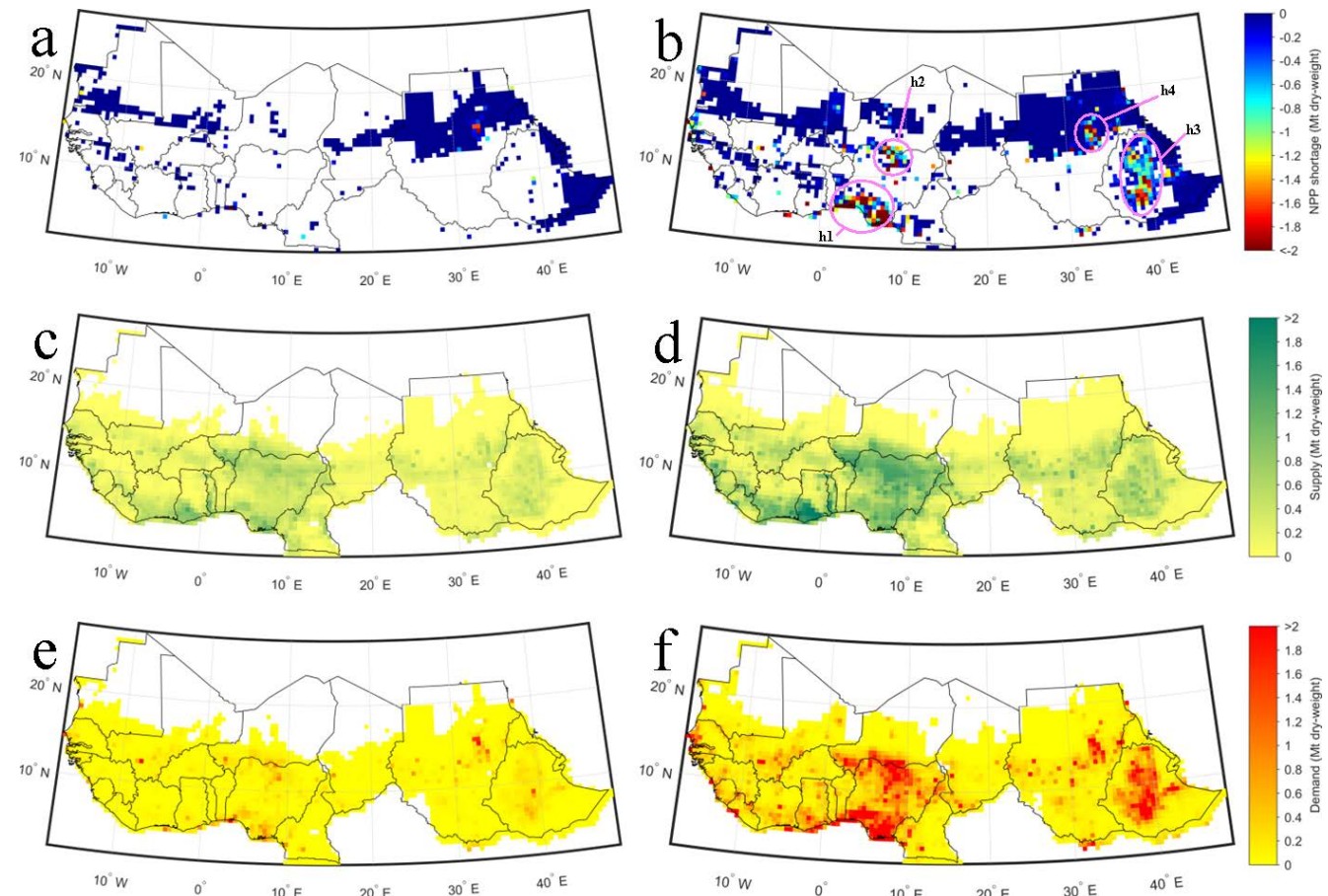

**Fig 4** Maps of NPP shortage (**a,b**), NPP supply (**c,d**) and NPP demand (**e,f**) for the year 2000 (left panels) and SSP2-RCP6.0 year 2050 (right panels). The hotspots of large NPP shortage are marked with circles in 4b, where h1 is in the area around Lagos (Nigeria) and the Niger delta; h2 is in the Nigerian hinterlands (close to Kano); h3 is in the Ethiopian highlands (close to Addis Ababa); and h4 is in the area surrounding Khartoum (Sudan). In 4a we excluded all areas with a surplus in the NPP balance.

**Tables**

**Table 1** Scenario matrix translated into quantitative probabilities (see also Engström et al. (2016b).

|  | RCP 2.6 | RCP 4.5 | RCP 6 | RCP 8.5 | Sum |
|---|---|---|---|---|---|
| **SSP1** | 0.0909 | 0.4545 | 0.4545 | 0.0000 | 1 |
| **SSP2** | 0.0000 | 0.0909 | 0.6818 | 0.2273 | 1 |
| **SSP3** | 0.0000 | 0.1667 | 0.5000 | 0.3333 | 1 |
| **SSP4** | 0.0000 | 0.3704 | 0.5556 | 0.0741 | 1 |
| **SSP5** | 0.0000 | 0.0741 | 0.3704 | 0.5556 | 1 |

**Table 2** Rules of combining $NPP_{cereal\_balance}$ and $NPP_{grazing\_balance}$ to determine the final balance of NPP demand and supply.

| Combination rule | $NPP_{cereal\_balance}$ | $NPP_{grazing\_balance}$ | $NPP_{balance}$ |
|---|---|---|---|
| 1 | <0 | ≥0 | $NPP_{cereal\_balance}$ |
| 2 | ≥0 | ≥0 | $NPP_{cereal\_balance}$ |
| 3 | <0 | <0 | $NPP_{cereal\_balance}$+ $NPP_{grazing\_balance}$ |
| 4 | ≥0 | <0 | $NPP_{cereal\_balance}$+ $NPP_{grazing\_balance}$ |

**Table 3** Summary of the Shared Socio-economic Pathway key characteristics (population development, economic growth, consumption & diet, policy orientation and technological change) based on (Engström et al., 2016; O'Neill et al., 2017).

| Pathway | Key characteristics |
|---|---|
| **SSP1:** **Sustainability - Taking the green road** | Relatively low population development |
| | Medium to high economic growth |
| | Low growth in material consumption, low-meat diets |
| | Towards sustainable development |
| | Rapid technology development and transfer |
| **SSP2:** **Middle of the road** | Medium population development |
| | Medium (but uneven) economic growth |
| | Material-intensive consumption, medium meat consumption |
| | Weak focus on sustainability |
| | Medium technology development and slow transfer |
| **SSP3:** **Regional rivalry - A rocky road** | High population development |
| | Slow economic growth |
| | Material-intensive consumption |
| | Oriented toward security |
| | Slow technology development and transfer |
| **SSP4:** **Inequality - A road divided** | Relatively high population development |
| | Low to medium economic growth |
| | Elites: high consumption, rest: low consumption |
| | Toward the benefit of the political and business elite |
| | Rapid technology transfer in high-tech sectors, but slow in other, little transfer within countries to poorer people |
| **SSP5:** **Fossil-fuel development - Taking the highway** | Relatively low population development |
| | High economic growth |
| | Materialisms, status consumption, meat-rich diets |
| | Toward development, free markets, human capital |
| | Rapid technology change and transfer |

**Table 4** Per capita NPP balance, NPP supply, NPP demand and population for SSP2-RCP6 for 2000 and 2050. All NPP is given in dry-weight (DW).

| Country | Per capita NPP balance [kg DW yr$^{-1}$] | | Per capita NPP supply [kg DW yr$^{-1}$] | | Per capita NPP demand [kg DW yr$^{-1}$] | | Total Population [millions] | |
|---|---|---|---|---|---|---|---|---|
| | 2000 | 2050 | 2000 | 2050 | 2000 | 2050 | 2000 | 2050 |
| **Benin** | 867 | -267 | 1341 | 607 | 474 | 874 | 8 | 25 |
| **Burkina Faso** | 737 | 147 | 933 | 316 | 196 | 169 | 12 | 46 |
| **Cameroon** | 1740 | 456 | 2127 | 1173 | 387 | 717 | 16 | 40 |
| **Chad** | 1220 | 326 | 1878 | 1484 | 658 | 1157 | 8 | 26 |
| **Djibouti** | -134 | -119 | 0 | 0 | 134 | 120 | 1 | 2 |
| **Eritrea** | 218 | 91 | 333 | 221 | 124 | 130 | 4 | 12 |
| **Ethiopia** | 366 | -660 | 825 | 779 | 459 | 1439 | 67 | 149 |
| **Gambia** | 431 | -449 | 1137 | 632 | 706 | 1082 | 1 | 3 |
| **Ghana** | 1216 | 211 | 1490 | 1291 | 274 | 1080 | 19 | 48 |
| **Guinea** | 1371 | 631 | 1773 | 1697 | 402 | 1066 | 8 | 22 |
| **Guinea Bissau** | 1720 | 714 | 2319 | 1648 | 599 | 934 | 1 | 3 |
| **Ivory Coast** | 1513 | 737 | 1795 | 1549 | 282 | 811 | 17 | 41 |
| **Liberia** | 975 | 39 | 1186 | 1312 | 212 | 1273 | 3 | 10 |
| **Mali** | 818 | -81 | 1929 | 1191 | 1111 | 1272 | 11 | 43 |
| **Mauritania** | -401 | -512 | 1129 | 1043 | 1530 | 1555 | 3 | 8 |
| **Niger** | 2163 | -114 | 3437 | 1426 | 1274 | 1540 | 11 | 55 |
| **Nigeria** | 738 | -204 | 1059 | 719 | 321 | 923 | 123 | 386 |
| **Senegal** | 369 | -297 | 925 | 539 | 556 | 837 | 10 | 28 |
| **Sierra Leone** | 565 | 183 | 759 | 949 | 194 | 767 | 4 | 12 |
| **Sudan** | 986 | -97 | 2517 | 1512 | 1530 | 1609 | 29 | 96 |
| **Togo** | 1900 | 838 | 2171 | 1491 | 271 | 653 | 5 | 11 |
| **Mean[1]** | 860 | -107 | 1377 | 957 | 517 | 1064 | 361 | 1066 |

5  [1]Weighted mean using national population data as weight.