# Peer review of "Future supply and demand of net primary production in the Sahel"

_Earth System Dynamics, 2016_

## Referee Comment (RC1) · T.A.M. Pugh (Referee) · 22 Dec 2016

Sallaba et al. present a coupled modelling system comprising a land-use model and an emulator of NPP outputs from the LPJ-GUESS vegetation model. They apply this coupled system for the Sahel to assess the likelihood of local food demand exceeding local supply during the 21st century, finding that this is the case in all but one SSP scenario, with many SSP-RCP combinations resulting in the lower 95% confidence bound of demand exceeding the upper 95% confidence bound of supply. $CO_2$ fertilisation and intensification of cropping were found to be important drivers of supply, but population-driven increases in demand where most influential. Overall, I find the manuscript to be well conceived, fairly clearly written and informative, and I recommend publication if the following concerns/queries can be addressed

Major comments

I presume the LPJ-GUESS simulations used to calibrate the BME model were potential natural vegetation (would help if this was explicitly stated)? In which case I wonder how effectively NPP of natural ecosystems can be used as a proxy for NPP of agricultural ones. NPP is not independent of plant type, and the distinction between natural vegetation, which may well be woody, and cereal and pasture vegetation may be particularly relevant in the Sahel, where the deeper roots of trees may have access to water resources that herbaceous plants cannot use. Can the authors demonstrate that such effects are not large, both in the LPJ-GUESS model and also based on any observations in the Sahel or analogous ecosystems?

Whilst the BME model is evaluated against LPJ-GUESS, any evaluation of the extent to which LPJ-GUESS can accurately represent actual NPP in the Sahel region is lacking. The references given (pg. 4 l. 14) did not address this ecosystem and also used a version of the model lacking carbon-nitrogen interactions, which leads to quite different vegetation simulations for the Sahel (Smith et al., 2014). Evaluation of the model response for the Sahel is necessary to give credence to the comparisons of supply and demand, which strongly depend on simulated absolute values for NPP. Whilst there is no gold-standard NPP (or GPP) dataset to compare against, comparison against NPP from the ESMs used to assess uncertainty, along with comparison of GPP against the alternative approaches of Jung et al. (2011) and Zhao et al. (2005) could go a long way towards increasing confidence. Alternatively (or additionally), FAO yield statistics could be used to evaluate the "yields" calculated here. Although none of these sources of comparison are likely to be low in uncertainty in the Sahel region, as it stands we have no idea how well LPJ-GUESS performs in this region - and current DGVMs cover a wide range of possibilities at regional scales (Sitch et al., 2015).

On the theme of evaluation. I'm not clear from the manuscript if PLUM land-use simulations are normalised in some way to the dataset of Hurtt et al. (2011) in 2000, or if they represent a purely "PLUM version" of the Sahel land-use in 2000. The former would

raise the question of how much the model drifts from the observed towards its preferred state at the start of the simulations. The latter suggests the need for a comparison of the PLUM initial state with current observation-based estimates (such as Hurtt et al., 2011). I realise there are significant difficulties in modelling actual land-use, but surely the size of any discrepancies and the resulting implications should be discussed?

Minor comments

pg. 2 l. 31. Why does a 31% population increase lead to a 100% increase in NPP requirement? What information is missing here?

pg. 6 l. 16. I'm confused about the cropland cover, I thought it was taken from PLUM? How is Hurtt being used here?

pg. 6 l. 23. Surely the total amount of NPP for human appropriation must be the sum of NPPcereal_demand and NPPgrazing_demand, not just NPPcereal_demand alone? As parts of both cereal and grazing demand contribute to animal raising, the current definition is inconsistent. Was it meant to be something like "total amount of annual NPP for human appropriation via cropland"?

The SSP-RCP scenario likelihoods seem rather important. Rather than referring the reader to another paper, maybe you could include them in this analysis? For instance along the right y-axis of Fig. 3b?

pg. 7 l. 29-33. This text reads as if it was originally located before the first paragraph of 2.1.3, and some of the text would seem to be more logically located there, where this likelihood matrix is first mentioned.

pg. 9 l. 11. I would say that the shortfalls in SSP5-RCP6.0 and SSP5-RCP8.5 are pretty sustained. They just don't run to the end of the century. Consider rephrasing?

More generally, regarding the discussion of "shortfalls", it seems strange that you only consider shortfalls to occur when the 95% confidence limits do not overlap (and demand is higher of course). To my mind this lack of overlap of the confidence limits

suggests very high likelihood of shortfalls, but the best guess result shows shortfalls occurring for a larger number of scenarios. For instance, on pg. 11, l. 26 it is stated that "statistically significant shortages never develop" in the context of SSP1, but that doesn't seem quite right. Assuming non-skewed distributions of uncertainty (big assumption, I know), then when the best estimate of demand exceeds the best estimate of supply there is a more than even chance of shortages occurring, but it's not possible to say with high certainty that a shortage will occur until the 95% limits no longer overlap. Consider rephrasing also?

pg. 9 l. 22. Reference to Table 3 here?

pg. 12 l. 3. Regarding, "so strong efforts should be made to reduce these gaps", this is too simplistic. Efforts to close yield gaps have other environmental and socio-economic consequences which are not addressed here, meaning that this statement cannot be supported by the presented evidence. I suggest to remove this recommendation. Going beyond this however, can you say anything about the potential additional yield by closing yield gaps in this region, and whether such efforts could alleviate the shortages simulated? Maybe PLUM can provide the necessary data?

pg. 12 l. 24. Where is the attribution of supply increases to additional rainfall and CO2 fertilisation shown in the results?

pg. 13 l. 7. The relative attribution of supply growth to climate/co2 and closure of yield gaps would be very informative, allowing the results to be interpreted more subtly. Your approach seems to be suitable to make this isolation.

pg. 13 l. 12. I would take the opposite view. The extent to which models appropriately represent CO2 fertilisation is not clear, and the difference in NPP trends between models is very large (e.g. Friend et al., 2014; Körner, 2006; Pugh et al., 2016; Rosenzweig et al., 2014). Therefore, I think it is fair to say that we have no more confidence in the trends than we do in the absolute levels. Moreover, the reference here to Fig. A2 does nothing to support the point, as the point of comparison is an LPJ-GUESS simulation,

not observations.

pg. 13 l. 22. You could also briefly mention irrigation water availability projections here (Elliott et al., 2014).

Grammatical/typographical

pg. 1, l. 20. "surplus, while" pg. 1 l. 23. "diet" pg. 2 l. 13. "global food security is not ensured" pg. 2 l. 16. "world, where" pg. 2 l. 19. "own land, where", also full stop missing after "pastoralism" pg. 4 l 32. Should "estimates to the total area", read " estimates to sum over the total area"? I don't think you translated NPP to total area literally? pg. 5 l. 22. Replace "Furthermore" with "Therefore" pg. 5 l. 32. "choice, and the" pg. 6 l. 13, 14, 20. "Fig. 2" should be "Fig. 1"? Also there are several boxes in red in Fig. 1 so "box outlined in red" is of limited use, and the distinction between cereal and pasture products can't be seen in the picture. pg. 8 l. 4. "Hence, one" pg. 10 l. 2. Only two countries are listed. pg. 12 l. 26. "mobilization is one method local" pg. 12 l. 31. "increase" pg. 14 l. 2. I think this would read better as "the Sahel is likely to experience NPP shortages in most SSP scenarios due to" pg. 14 l. 7. Reference formatting. pg. 14 l. 25. "show" rather than "assume"? pg. 15 l. 2. "will outstrip supply during the 21st century". pg. 15 l.12. "unfolds, a relatively"

References

Elliott, J., Deryng, D., Müller, C., Frieler, K., Konzmann, M., Gerten, D., Glotter, M., Flörke, M., Wada, Y., Best, N., Eisner, S., Fekete, B.M., Folberth, C., Foster, I., Gosling, S.N., Haddeland, I., Khabarov, N., Ludwig, F., Masaki, Y., Olin, S., Rosenzweig, C., Ruane, A.C., Satoh, Y., Schmid, E., Stacke, T., Tang, Q., Wisser, D., 2014. Constraints and potentials of future irrigation water availability on agricultural production under climate change. Proc. Natl. Acad. Sci. U. S. A. 111, 3239–44. doi:10.1073/pnas.1222474110

Friend, A.D., Lucht, W., Rademacher, T.T., Keribin, R., Betts, R., Cadule, P., Ciais,

P., Clark, D.B., Dankers, R., Falloon, P.D., Ito, A., Kahana, R., Kleidon, A., Lomas, M.R., Nishina, K., Ostberg, S., Pavlick, R., Peylin, P., Schaphoff, S., Vuichard, N., Warszawski, L., Wiltshire, A., Woodward, F.I., 2014. Carbon residence time dominates uncertainty in terrestrial vegetation responses to future climate and atmospheric CO2. Proc. Natl. Acad. Sci. U. S. A. 111, 3280–3285. doi:10.1073/pnas.1222477110

Hurtt, G.C., Chini, L.P., Frolking, S., Betts, R. a., Feddema, J., Fischer, G., Fisk, J.P., Hibbard, K., Houghton, R. a., Janetos, a., Jones, C.D., Kindermann, G., Kinoshita, T., Klein Goldewijk, K., Riahi, K., Shevliakova, E., Smith, S., Stehfest, E., Thomson, a., Thornton, P., Vuuren, D.P., Wang, Y.P., 2011. Harmonization of land-use scenarios for the period 1500–2100: 600 years of global gridded annual land-use transitions, wood harvest, and resulting secondary lands. Clim. Change 109, 117–161. doi:10.1007/s10584-011-0153-2

Jung, M., Reichstein, M., Margolis, H. a., Cescatti, A., Richardson, A.D., Arain, M.A., Arneth, A., Bernhofer, C., Bonal, D., Chen, J., Gianelle, D., Gobron, N., Kiely, G., Kutsch, W., Lasslop, G., Law, B.E., Lindroth, A., Merbold, L., Montagnani, L., Moors, E.J., Papale, D., Sottocornola, M., Vaccari, F., Williams, C., 2011. Global patterns of land-atmosphere fluxes of carbon dioxide, latent heat, and sensible heat derived from eddy covariance, satellite, and meteorological observations. J. Geophys. Res. 116, G00J07. doi:10.1029/2010JG001566

Körner, C., 2006. Plant CO2 responses: an issue of definition, time and resource supply. New Phytol. 172, 393–411.

Pugh, T.A.M., Müller, C., Arneth, A., Haverd, V., Smith, B., 2016. Key knowledge and data gaps in modelling the influence of CO2 concentration on the terrestrial carbon sink. J. Plant Physiol. 203, 3–15. doi:10.1016/j.jplph.2016.05.001

Rosenzweig, C., Elliott, J., Deryng, D., Ruane, A.C., Müller, C., Arneth, A., Boote, K.J., Folberth, C., Glotter, M., Khabarov, N., Neumann, K., Piontek, F., Pugh, T. a M., Schmid, E., Stehfest, E., Yang, H., Jones, J.W., 2014. Assessing agricultural risks of

climate change in the 21st century in a global gridded crop model intercomparison. Proc. Natl. Acad. Sci. U. S. A. 111, 3268–73. doi:10.1073/pnas.1222463110

Sitch, S., Friedlingstein, P., Gruber, N., Jones, S.D., Murray-Tortarolo, G., Ahlström, a., Doney, S.C., Graven, H., Heinze, C., Huntingford, C., Levis, S., Levy, P.E., Lomas, M., Poulter, B., Viovy, N., Zaehle, S., Zeng, N., Arneth, a., Bonan, G., Bopp, L., Canadell, J.G., Chevallier, F., Ciais, P., Ellis, R., Gloor, M., Peylin, P., Piao, S.L., Le Quéré, C., Smith, B., Zhu, Z., Myneni, R., 2015. Recent trends and drivers of regional sources and sinks of carbon dioxide. Biogeosciences 12, 653–679. doi:10.5194/bg-12-653-2015

Smith, B., Wårlind, D., Arneth, a., Hickler, T., Leadley, P., Siltberg, J., Zaehle, S., 2014. Implications of incorporating N cycling and N limitations on primary production in an individual-based dynamic vegetation model. Biogeosciences 11, 2027–2054. doi:10.5194/bg-11-2027-2014

Zhao, M., Ann, F., Nemani, R.R., Running, S.W., 2005. Improvements of the MODIS terrestrial gross and net primary production global data set. Remote Sens. Environ. 95, 164–176. doi:10.1016/j.rse.2004.12.011

---

## Referee Comment (RC2) · Anonymous Referee #2 · 13 Jan 2017

Dear Editor,

This manuscript does NOT satisfies your editorial criteria as described at http://www.earth-system-dynamics.net/peer_review/review_criteria.html

This manuscript perhaps intends to make contributions to regional studies of the socio-economic implications of global change, particularly about the Sahel and its delicate balance between supply and demand of natural resources, with a focus on its implications for food production; however some of its methods are flawed and the use of information weak.

This paper deals with very delicate topics that deserve honour and credit, but using the wrong tools to address them, for which the authors deserve no mercy.

[Figure]

Therefore I recommend the rejection of this manuscript.

FURTHER COMMENTS

LPJ and the like models are normally very rough on their predictions, if you simplify them more, then your results might be useless.

Pg. 4 line 25, you evaluate the performance of your model against another model (LPJ)? Why this is good science deserving publication? This is bad science. Have you thought about doing it against data?

You do a regional level study using GCM data? Not good practice. See what other Swedish colleagues do with regional data there: http://www.smhi.se/en/research/research-departments/climate-research-rossby-centre2-552/an-ensemble-of-cordex-africa-climate-projections-simulated-by-rca4-1.25312

What you intend to argue, deriving insights from NPP into food production related arguments, is very weak in methodological terms, and although your rationale and arguments are sensible, the methods you use disqualify the support you use for the argumentation.

Then you use a convenient "technology improvement factor"? and close the yield gap with it? I am sorry, again, this is bad science, and it should not be published.

---

## Referee Comment (RC3) · Anonymous Referee #3 · 18 Apr 2017

This manuscript aims to forecast the spatiotemporal supply and demand of net primary productivity (NPP) across the Sahel region for the full 21st century. The authors utilize a simple vegetation production model forced by four RCP projections and a simple socioeconomic model based on assumptions derived from five shared socioeconomic pathways to quantify spatiotemporal variability in NPP supply and demand, respectively. Results indicate widespread NPP shortfalls across the region by 2050 due to population increases and shifts toward diets rich in animal products. The authors conclude that the UN sustainable development goals for ending hunger are at high risk for failure. Overall, I feel the manuscript presents a useful framework for addressing the eminent grand challenge of meeting future demand in the face of climate change, dietary shifts, and population growth. The manuscript is well written and the methodological detail is clear and sufficient. The findings are of interest to a broad audience

including land managers and policy makers. Yet, there are apparent major flaws in the modeling approach that must be addressed before I can recommend publication of this manuscript. My main issue is that . . . I recommend ... Please find my general and specific comments below.

Major Comments: 1. The introduction is very well-written and does an excellent job of framing the question and establishing the importance of the work.

2. Page 4, line 27: It is stated that the authors used 0.5 degree climate data from five GCMs, and [CO2] based on four RCPs (Representative Concentration Pathways). The way it is phased it is unclear which RCPs were used to generate the climate projections for each GCM. The authors should have used climate data derived from runs across the 4 RCPs for each of the 5 models. Please clarify the text if this is the case. If not, please explain more fully why the climate data were not derived for all RCPs.

3. Robustly representing future NPP trajectories is challenging due to the many potential counteracting feedbacks. The authors show a good fit with LPJ NPP simulations, but do not consider observational data or alternative runs of the LPJ model itself. I recommend further comparison against both census derived yield trends (Rey et al. 2013) and satellite-derived yield trends (Running et al. 2004). For instance, the authors could consider runs in which the CO2 fertilization effect is turned off. Currently all the NPP trends considered in the paper are increasing due to CO2 fertilization (page 9, line 24). This is an area of debate and may be counter to observational data (see Smith et al. 2016, Oberneier et al. 2016, and Ort & Long et al. 2014). Thus, I wonder if a scenario in which CO2 fertilization effects are isolated and removed would be a more realistic lower boundary on what to expect for the region? I would imagine very large increases in the NPP debt (without large irrigation efforts), much larger than what is currently considered in the paper.

Running, S.W., Nemani, R.R., Heinsch, F.A., Zhao, M., Reeves, M. & Hashimoto, H. (2004). A Continuous Satellite-Derived Measure of Global Terrestrial Primary Production. Bioscience, 54, 547–560.

Ray, D.K., Mueller, N.D., West, P.C. & Foley, J.A. (2013). Yield Trends Are Insufficient to Double Global Crop Production by 2050. PLoS One, 8, e66428.

Smith, W.K., Reed, S.C., Cleveland, C.C., Ballantyne, A.P., Anderegg, W.R.L., Wieder, W.R., Liu, Y.Y. & Running, S.W. (2016). Large divergence of satellite and Earth system model estimates of global terrestrial CO2 fertilization. Nat. Clim. Chang., 6, 306–310.

Obermeier, W.A., Lehnert, L.W., Kammann, C.I., Müller, C., Grünhage, L., Luterbacher, J., Erbs, M., Moser, G., Seibert, R., Yuan, N. & Bendix, J. (2016). Reduced CO2 fertilization effect in temperate C3 grasslands under more extreme weather conditions. Nat. Clim. Chang., 7, 137–142.

Ort, D.R. & Long, S.P. (2014). Limits on Yields in the Corn Belt. Science (80-. )., 344, 484–485. McGrath, J.M. & Lobell, D.B. (2013). Regional disparities in the CO 2 fertilization effect and implications for crop yields. Environ. Res. Lett., 8, 14054.

4. Page 5, line 4: When the fractional agricultural landcover estimates from Hurtt et al (2011) were applied, was it assumed that natural and agricultural NPP were similar? If so, this assumption should be revisited after considering differences between agricultural vs. natural NPP for the region. For instance, the authors could compare census-based estimates of crop productivity with their estimates as a reality check. Smith et al. 2014 (see reference below), found that agricultural productivity for the region is significantly lower than natural productivity. If this potential reality is not considered, then the scenarios in this manuscript may be overly optimistic.

Smith, W., Cleveland, C.C., Reed, S.C. & Running, S.W. (2014). Agricultural conversion without external water and nutrient inputs reduces terrestrial vegetation productivity. Geophys. Res. Lett., 41, 449–455.

5. I would recommend revisiting all crop allocation parameters based on those reported by Monfreda et al. (2008). Given the high variability in crop specific harvest fractions,

it seems it may be necessary to parameterize the model for each individual crop grown in the region.

Monfreda, C., Ramankutty, N. & Foley, J.A. (2008). Farming the planet: 2. Geographic distribution of crop areas, yields, physiological types, and net primary production in the year 2000. Global Biogeochem. Cycles, 22, 1–19.

6. Page 12 line 24-26: This statement is not representative of the literature (see below references). I would suggest more nuanced discussion of the potential limitations of the supply approach used in this analysis. For instances, how much did $CO_2$ fertilization drive increases? How uncertain are the precipitation estimates? Were nutrient constraints considered and if so what are the management implications? If not, how might nutrient constraints limit NPP? How will increases in atmospheric water demand (Vapor pressure deficit) affect yields and productivity? Could increased drought and desertification also represent a potential scenario had the $CO_2$ sensitivity been adjusted? The way that this section is currently written is a gross over extension of the simplified NPP modeling that the paper is based on.

Smith, W.K., Reed, S.C., Cleveland, C.C., Ballantyne, A.P., Anderegg, W.R.L., Wieder, W.R., Liu, Y.Y. & Running, S.W. (2016). Large divergence of satellite and Earth system model estimates of global terrestrial $CO_2$ fertilization. Nat. Clim. Chang., 6, 306–310.

Obermeier, W.A., Lehnert, L.W., Kammann, C.I., Müller, C., Grünhage, L., Luterbacher, J., Erbs, M., Moser, G., Seibert, R., Yuan, N. & Bendix, J. (2016). Reduced $CO_2$ fertilization effect in temperate C3 grasslands under more extreme weather conditions. Nat. Clim. Chang., 7, 137–142.

Ort, D.R. & Long, S.P. (2014). Limits on Yields in the Corn Belt. Science (80-. )., 344, 484–485. McGrath, J.M. & Lobell, D.B. (2013). Regional disparities in the CO 2 fertilization effect and implications for crop yields. Environ. Res. Lett., 8, 14054.

Wieder, W.R., Cleveland, C.C., Smith, W.K. & Todd-Brown, K. (2015). Future productivity and carbon storage limited by terrestrial nutrient availability. Nat. Geosci., 1–5.

Minor Comments: 1. Page 3, Line 17: "Three different aggregation levels are considered, including Sahel, the country, and the local". Please define what is meant by local level. Pixel level? What resolution? 2. Page 12 line 12: missing end of parentheses.

―――――――――――――――――――――

---

## Author Comment (AC2) · 29 Jun 2017

We thank Dr. Pugh for providing constructive and insightful comments on our manuscript. Below, we provide a point-by-point response to each comment. Note that we have already made minor changes in our manuscript suggested by Dr. Pugh, and submit a marked up manuscript to show these. We also indicate changes made in our responses.

Thomas Pugh: I presume the LPJ-GUESS simulations used to calibrate the BME model were potential natural vegetation (would help if this was explicitly stated)? In which case I wonder how effectively NPP of natural ecosystems can be used as a proxy for NPP of agricultural ones. NPP is not independent of plant type, and the dis-

tinction between natural vegetation, which may well be woody, and cereal and pasture vegetation may be particularly relevant in the Sahel, where the deeper roots of trees may have access to water resources that herbaceous plants cannot use. Can the authors demonstrate that such effects are not large, both in the LPJ-GUESS model and also based on any observations in the Sahel or analogous ecosystems?

Authors' Response: The LPJ-GUESS (C-N) simulations used to calibrate BME were based on potential natural vegetation and we have already made this change in the manuscript. In order to test how effectively the NPP of natural ecosystems can be can be used as a proxy for the NPP of agricultural ones we ran LPJ-GUESS managed land (C-N version) for the period 1970 to 2010 and compared this to LPJ-GUESS (C-N, and used to develop BME), for the greater Sahel region defined in our manuscript (note that for comparison purposes, we also provide runs of LPJ-GUESS (C-only), BME and MOD-17). The results (see Figure 1) of this experiment show that mean NPP derived from LPJ-GUESS ml over the region underestimates mean NPP derived from LPJ-GUESS by 2.4% (0.02 kg dry-weight m-2 yr-1), though both models show similar levels of interannual variability and trend (see Figure 1). The implication of this experiment is that there is a demonstrable reduction in NPP when land management is taken into consideration, but the effect is relatively minor. The one caveat is that modelled crop yield in LPJ-GUESS (C-N) produces estimates of potential yield for various crops, rather than actual yield, therefore representing an upper limit to actual yield. Lindeskog et al. (2013) show that LPJ-GUESS managed land (C-version) overestimates actual yield derived from FAO country-level crop statistics. Smith et al. (2014a) also report that natural systems are more productive than agricultural systems in sub-Saharan Africa. We conclude that our results are likely in the upper range for NPP in the Sahel, and that our analysis across the scenarios regarding the advent of supply shortfalls is optimistic. We will be willing to add this point to the discussion in a revised version of our manuscript.

Thomas Pugh: Whilst the BME model is evaluated against LPJ-GUESS, any evaluation

of the extent to which LPJ-GUESS can accurately represent actual NPP in the Sahel region is lacking. The references given (pg. 4 l. 14) did not address this ecosystem and also used a version of the model lacking carbon-nitrogen interactions, which leads to quite different vegetation simulations for the Sahel (Smith et al., 2014). Evaluation of the model response for the Sahel is necessary to give credence to the comparisons of supply and demand, which strongly depend on simulated absolute values for NPP. Whilst there is no gold-standard NPP (or GPP) dataset to compare against, comparison against NPP from the ESMs used to assess uncertainty, along with comparison of GPP against the alternative approaches of Jung et al. (2011) and Zhao et al. (2005) could go a long way towards increasing confidence. Alternatively (or additionally), FAO yield statistics could be used to evaluate the "yields" calculated here. Although none of these sources of comparison are likely to be low in uncertainty in the Sahel region, as it stands we have no idea how well LPJ-GUESS performs in this region - and current DGVMs cover a wide range of possibilities at regional scales (Sitch et al., 2015).

Authors' Response: Firstly, thank-you very much for highlighting these validation issues. We compare total yearly means of NPP from LPJ-GUESS (N-C) to NPP derived from the MOD-17 processing stream for the period 2000 to 2006 for the greater Sahel region as defined in our manuscript. Our results show that MOD-17 derived NPP underestimates modelled NPP from LPJ-GUESS N-C by 43% (0.37 kg dry-weight m-2 yr-1) (Figure 1). Despite this, the R2-value between the two series is 0.8 suggesting similarity in both interannual variability and trend. Ardö (2015) also reports that that average annual MOD-17 NPP underestimates LPJ-GUESS (C version) for Africa for 2000-2010 and attributes this to the fact that autotrophic respiration is considerably higher for MOD17 compared to LPJ-GUESS, due to large temperature sensitivity in the MOD17 algorithm, differences in the biome-specific parameterizations in MOD-17 as well as specification of plant functional types in LPJ-GUESS.

We also gauged LPJ-GUESS (N-C) and BME performance for estimating NPP with NPP field-measurements from Michaletz et al. (2016) and Luyssaert et al. (2009) at the

biome level (see Sallaba et al., 2015) for the Major Biome Classification of Reich and Eswaran (2002) including the biomes found in the Sahel (Desert Temperate, Tropical Semi-arid and Tropical Humid). Note that since only two observations were available for our study area (see Figure 2) this evaluation demonstrates the ability of both LPJ-GUESS and BME to replicate NPP for Sahel biomes but found elsewhere in the world.

Before we combined the Michaletz et al. (2016) and Luyssaert et al. (2009) datasets, we removed sites with no records of combined above- and below-ground NPP measurements. After we merged the data, we checked the final assembly of NPP measurements for duplicates and removed them. The final dataset consists of 1561 samples (i.e. 1247 samples from Michaletz et al. (2016) and 314 samples from Luyssaert et al. (2009)) representing total NPP measurements across the terrestrial biosphere (sample sizes are 18, 6, and 12 for Sahel biomes of Desert Temperate, Tropical Semi-arid and Tropical Humid, respectively) from 1959-2006. Both LPJ-GUESS (N-C) and BME were driven with CRU TS 3.21 climate data (Harris et al. 2014, Trenberth et al. 2014) that has global coverage across the time period. We calculated mean values of the NPP field-measurements and the modelled NPP estimates located in the respective biomes, following Smith et al. (2014b). We aggregated to the biome-level to account for the difference in scale between in situ NPP measurements and modelled grid cell NPP estimates (being grid cell averages).

Finally, we determined the overall model performance, biome-by-biome, with the coefficient of determination ($R^2$ value) and the root mean square error (RMSE). Additionally, we investigated model agreement with performance ratios (hereafter referred to as 'Q') by dividing mean biome NPP estimates (for both models) with mean biome NPP observations. Model overestimation in comparison to in situ NPP measurements is indicated by $Q > 1$ and underestimation by $Q < 1$. Good model performance is classified with a Q range between 0.9-1.1 assuming an error of $\pm$ 10% following Sallaba et al. (2015). However, we further defined an acceptable model performance error range of $\pm20\%$ (i.e. Q = 0.8-1.25) given the limitations of using LPJ-GUESS (C-N) standard modelling

protocol, PNV and CRU climate observations, and especially the simplicity of BME.

LPJ-GUESS (N-C) performs reasonably well in simulating NPP at the overall biome level (R2 = 0.71 and RMSE = 0.16) but the model performance varies notably across the biomes (see Figure 3 and Table 1). In general, LPJ-GUESS (N-C) yields acceptable model agreement in seven (with good performance in four biomes) out of thirteen biomes. At the same time, the model underestimates NPP in three biomes while it overestimates NPP in two biomes as shown in Figure 3.

For Greater Sahel biomes: LPJ-GUESS (N-C) exhibits good skill in simulating NPP in the Tropical humid (Q = 0.96, see Table 1) where it also captures satisfactorily the variability of the NPP measurements. LPJ-GUESS (N-C) underestimates NPP for the tropical semi-arid biome (Q = 0.75) showing reduced NPP variation compared to the observations. Performance is reduced for Desert temperate (Q =0.56).

BME performance is acceptable at the overall biome level (R2 = 0.57 and RMSE = 0.26) but varies substantially for individual biomes (see Figure 4). Overall, BME model agreement is reasonable in four biomes (with good performance in two biomes). At the same time, BME overestimates NPP in two biomes while it underestimates plant growth in six biomes. The variability in in- situ NPP measurements cannot be captured by BME in the majority of biomes except in the tropical humid and tundra permafrost biomes (see vertical and horizontal lines connected to the diamonds in Figure 4).

For Greater Sahel biomes: BME yields acceptable agreement in estimating NPP in the tropical semi-arid and tropical humid biomes (Q = 0.84, 0.81 respectively) but accuracy drops more water limited biomes of desert temperate (Q = 0.28).

Overall, BME mimics the behavior of LPJ-GUESS (N-C) , shown by a good model agreement of R2 = 0.71 and moderate RMSE = 0.12 kg C m-2 yr-1 between the average biome NPP estimates of BME and LPJ-GUESS (N-C). BME yields on average less NPP in the majority of biomes compared to the observations. In sum, a comparison with MOD-17 shows that LPJ-GUESS N-C (and BME) overestimates total mean annual

NPP in the greater Sahel region (2000-2006) while a validation involving ground measurements for the same biomes found in the Sahel (but observations mostly from other locations) show that LPJ-GUESS (N-C) and BME underestimate NPP. Differences are due to a combination of spatial aggregation/sampling issues (e.g. low sample sizes for biomes typically found in the Sahel, that CRU data do not necessarily represent site-level climate, and the uncertain assessment below-ground and short-lived above-ground plant matter at the site level) as well differing assumptions between the MOD-17 processing stream and LPJ-GUESS (N-C) (particularly respiration). We conclude that LPJ-GUESS (N-C) and BME replicate ground observations of NPP at similar orders of magnitude at the biome level. This underscores the fact that LPG-GUESS (N-C) and BME should be restricted to biome-level applications (or coarser) while applications on the grid cell level should be limited to explorations of patterns and trends, which is the reason why, in our manuscript, we emphasize an aggregated level.

We would be happy to include, in the appendix of a new version of our manuscript, a complete description of this validation exercise.

Thomas Pugh: On the theme of evaluation. I'm not clear from the manuscript if PLUM land-use simulations are normalised in some way to the dataset of Hurtt et al. (2011) in 2000, or if they represent a purely "PLUM version" of the Sahel land-use in 2000. The former would raise the question of how much the model drifts from the observed towards its preferred state at the start of the simulations. The latter suggests the need for a comparison of the PLUM initial state with current observation-based estimates (such as Hurtt et al., 2011). I realise there are significant difficulties in modelling actual land-use, but surely the size of any discrepancies and the resulting implications should be discussed?

Authors' Response: The Hurtt et al. (2011) data for the year 2000 is used as basis, and we will make sure to clarify this in the updated manuscript. In the table below the scaling factors for the year 2000 are shown, these numbers will be added to the Appendix, possibly to Table C1. The scaling factors are the per country ratios Hurtt:PLUM.

Thomas Pugh: pg. 2 l. 31. Why does a 31% population increase lead to a 100% increase in NPP requirement? What information is missing here?

Authors' Response: The line from our original manuscript is reproduced here:"Abdi et al. (2014) also showed that 19% of the NPP supply in the Sahel was able to satisfy demand for the year 2000 but this increased to 41% in 2010 due to a 31% increase in the population." Abdi et al. (2014) point out that NPP demand increased at an annual rate of 2.2% between 2000 and 2010 while the supply was near constant. So, in relative terms, the doubling in NPP demand is simply because there is less NPP supply to service the increase in population. We have already clarified this in our manuscript.

Thomas Pugh: pg. 6 l. 16. I'm confused about the cropland cover, I thought it was taken from PLUM? How is Hurtt being used here?

Authors' Response: Please refer the previous response.

Thomas Pugh: pg. 6 l. 23. Surely the total amount of NPP for human appropriation must be the sum of NPPcereal_demand and NPPgrazing_demand, not just NPPcereal_demand alone? As parts of both cereal and grazing demand contribute to animal raising, the current definition is inconsistent. Was it meant to be something like "total amount of annual NPP for human appropriation via cropland"?

Authors' Response: We have already changed this sentence to read "total amount of annual NPP for human appropriation via cropland." Indeed, we explicitly distinguish between the demand of cereal and pasture products. Cereal demand is given in Equation 1 of the manuscript, while grazing demand is given in Equation 9 (not Equation 8 as stated in the first version, Appendix A3 – we have changed this too). Cereal-based and grazing-based supply-demand balances are then computed separately. They are then summed according to Table 1 of the manuscript in order to determine final balances of supply and demand of NPP.

Thomas Pugh: The SSP-RCP scenario likelihoods seem rather important. Rather than referring the reader to another paper, maybe you could include them in this analysis? For instance along the right y-axis of Fig. 3b?

Authors' Response: Table 3 shows the scenario likelihoods, and is the same as Table 4 found in Engström et al. (2016). We would be happy to include them in a new version of the manuscript. Note that these likelihoods refer to the most consistent SSP-RCP combinations (e.g. it is more likely that the sustainability assumptions for SSP1 would yield greenhouse gas concentrations in line with RCP4.5/6 rather than RCP2.6/8.5).

Thomas Pugh: pg. 7 l. 29-33. This text reads as if it was originally located before the first paragraph of 2.1.3, and some of the text would seem to be more logically located there, where this likelihood matrix is first mentioned.

Authors' Response: We have already moved this information to the suggested location in our manuscript.

Thomas Pugh: pg. 9 l. 11. I would say that the shortfalls in SSP5-RCP6.0 and SSP5-RCP8.5 are pretty sustained. They just don't run to the end of the century. Consider rephrasing? More generally, regarding the discussion of "shortfalls", it seems strange that you only consider shortfalls to occur when the 95% confidence limits do not overlap (and demand is higher of course). To my mind this lack of overlap of the confidence limits suggests very high likelihood of shortfalls, but the best guess result shows shortfalls occurring for a larger number of scenarios. For instance, on pg. 11, l. 26 it is stated that "statistically significant shortages never develop" in the context of SSP1, but that doesn't seem quite right. Assuming non-skewed distributions of uncertainty (big assumption, I know), then when the best estimate of demand exceeds the best estimate of supply there is a more than even chance of shortages occurring, but it's not possible to say with high certainty that a shortage will occur until the 95% limits no longer overlap. Consider rephrasing also?

Authors' Response: These items have been rephrased, here we produce a suggestion
for a rewritten portion of the results that, if acceptable, we would be happy to include a revised manuscript:

"Per capita demand exceeds supply in the early 2040s for SSP2-RCP6.0 after which a very high likelihood for shortfalls begin in 2070 (see black dots in Fig. 3a showing non-overlapping 95% confidence limits). By 2050, per capita demand almost doubles while per capita supply drops by almost 30% for the same scenario. Across the scenarios, differences in the timing of the start of persistent supply shortfalls with high certainty (see black dots in Fig. 3b) are observed. Three of these high likelihood shortfalls begin at 2050 or before (SSP5 scenarios – see black dots in Fig. 3b) while an additional six display shortfalls with high certainty by the end of the 21st century (black dots in Fig. 3a, b). Out of these nine, two scenarios never achieve a sustained run of shortfalls (SSP2-RCP6.0, SPP2-RCP8.5).). In total, there is better than an even chance for shortfalls before 2050 for 9 scenarios (exceptions are SSP1-RCP2.6, SSP1-RCP6.0, and all SSP4 scenarios).

Variations in the timing of onset and end of supply shortfalls are generally greater between the SSPs than between the RCPs (Fig. 3b). For SSP2 and SSP3 scenarios, onsets of high likelihood supply shortfall range from the early 2050s to the mid-2070s (even chance from late 2030s to early 2050s). The SSP5 family shows the largest deficits of high likelihood shortfalls beginning in the 2040s-2050s (even chance from the early 2030s), and after several decades of deepening begin to diminish in the 2080s. Shortfalls with high certainty never emerge for SSP1 (even chance from the early 2050s) while the SSP4 scenarios show sustained but diminishing surplus throughout."

Thomas Pugh: pg. 9 l. 22. Reference to Table 3 here?

Authors' Response: We have already referred Table 3 at this location in our manuscript.

Thomas Pugh: pg. 12 l. 3. Regarding, "so strong efforts should be made to reduce these gaps", this is too simplistic. Efforts to close yield gaps have other environmental

and socio-economic consequences which are not addressed here, meaning that this statement cannot be supported by the presented evidence. I suggest to remove this recommendation. Going beyond this however, can you say anything about the potential additional yield by closing yield gaps in this region, and whether such efforts could alleviate the shortages simulated? Maybe PLUM can provide the necessary data?

Authors' Response: We will remove this recommendation in a revised manuscript. And thank you very much for suggesting this experiment, which we have now conducted. We find that closing production gaps in the greater Sahel for the year 2050 (the mid-century point of reference given in our manuscript), for the scenario SSP2-6.0, would result in a change in mean per capita NPP balance from -107 kg DW yr-1 (see Table 3 in the manuscript) to 9 kg DW yr-1 – though the balance for many countries will still be negative, but reduced in magnitude. We conclude that closing yield gaps in the region could indeed alleviate the simulated shortages by mid-century. We would be happy to briefly treat this aspect in the discussion of a revised manuscript.

Thomas Pugh: pg. 12 l. 24. Where is the attribution of supply increases to additional rainfall and CO2 fertilisation shown in the results?

Authors' Response: Fig 5 shows that for SSP2-RCP6, CO2 contributes far more to the increase in NPP compared to rainfall for the greater Sahel region. I order to produce the combined effect of CO2 and rainfall, we compared a simulation where both variables taken from the scenario were compared with a simulation where both were held constant from the year 2000 through to 2050. In order to isolate the CO2 (rainfall) effect, we compared a simulation where rainfall (CO2) was held constant with the simulation where both were held constant. We performed these simulations for RCP 6.0 for all GCMS. The mean of the scenarios are shown in Figure 5. We can add these findings to the results section of our manuscript.

Thomas Pugh: pg. 13 l. 7. The relative attribution of supply growth to climate/co2 and closure of yield gaps would be very informative, allowing the results to be interpreted

more subtly. Your approach seems to be suitable to make this isolation.

Authors' Response: Fig 5 shows that for SSP2, the reduction in yield gap between 2000 and 2050 contributes slightly more to the increase in NPP than CO2 for RCP 6.0, and in turn much more than rainfall for the same climate scenario. We can update our manuscript to account for the yield gap effect.

Thomas Pugh: pg. 13 l. 12. I would take the opposite view. The extent to which models appropriately represent CO2 fertilisation is not clear, and the difference in NPP trends between models is very large (e.g. Friend et al., 2014; Körner, 2006; Pugh et al., 2016; Rosenzweig et al., 2014). Therefore, I think it is fair to say that we have no more confidence in the trends than we do in the absolute levels. Moreover, the reference here to Fig. A2 does nothing to support the point, as the point of comparison is an LPJ-GUESS simulation, not observations.

Authors' Response: Thanks very much for highlighting issues with the trends. Please see our response to Anonymous Review # 3 for a broader discussion of the trends (e.g. responses to comments #3 and #6). We suggest modifying our sentence in a revised version to:

"Uncertainty exists with respect to the total magnitude and trends of simulated NPP supply (given the lack of ground truth for the region, and that differences in NPP trends between models is very large (e.g. Friend et al., 2014; Körner et al., 2006; Pugh et al., 2016; Rosenzweig et al., 2014). Therefore, our emphasis is on the structural analysis of NPP supply and demand across a range of scenarios. This also serves to demonstrate the usefulness of our overall approach for this application."

Thomas Pugh: pg. 13 l. 22. You could also briefly mention irrigation water availability projections here (Elliott et al., 2014).

Authors' Response: We suggest the following alteration a revised version of the manuscript

"However, Elliott et al. (2014) underscore that freshwater limitations in the dryer regions of the globe could limit agricultural production, and even lead to the reversion of irrigated farmland to rainfed farmland thereby negatively affecting food production. Conventional agricultural intensification can result in environmental degradation, vulnerability to pests, and depletion of aquifers (Ceccato et al., 2007; Foley et al., 2005)."

Thomas Pugh: pg. 1, l. 20. "surplus, while" pg. 1 l. 23. "diet" pg. 2 l. 13. "global food security is not ensured" pg. 2 l. 16. "world, where" pg. 2 l. 19. "own land, where", also full stop missing after "pastoralism" pg. 4 l 32. Should "estimates to the total area", read " estimates to sum over the total area"? I don't think you translated NPP to total area literally? pg. 5 l. 22. Replace "Furthermore" with "Therefore" pg. 5 l. 32. "choice, and the" pg. 6 l. 13, 14, 20. "Fig. 2" should be "Fig. 1"? Also there are several boxes in red in Fig. 1 so "box outlined in red" is of limited use, and the distinction between cereal and pasture products can't be seen in the picture. pg. 8 l. 4. "Hence, one" pg. 10 l. 2. Only two countries are listed. pg. 12 l. 26. "mobilization is one method local" pg. 12 l. 31. "increase" pg. 14 l. 2. I think this would read better as "the Sahel is likely to experience NPP shortages in most SSP scenarios due to" pg. 14 l. 7. Reference formatting. pg. 14 l. 25. "show" rather than "assume"? pg. 15 l. 2. "will outstrip supply during the 21st century". pg. 15 l.12. "unfolds, a relatively"

Authors Response: These have been fixed.

References in our responses:

Abdi, A. M., Seaquist, J., Tenenbaum, D. E., Eklundh, L., and Ardö, J.: The supply and demand of net primary production in the Sahel, Environmental Research Letters, 9, 094003, 2014.

Ardö, J. Comparison between remote sensing and a dynamic vegetation model for estimating terrestrial primary production of Africa. Carbon Balance and Management, 10(8), 2015.

[Figure]

Ceccato, P., Cressman, K., Giannini, A., and Trzaska, S.: The desert locust upsurge in West Africa (2003-2005): Information on the desert locust early warning system and the prospects for seasonal climate forecasting, International Journal of Pest Management, 53, 7-13, 2007.

Elliott, J., Deryng, D., Müller, C., Frieler, K., Konzmann, M., Gerten, D., Glotter, M., Flörke, M., Wada, Y., Best, N., Eisner, S., Fekete, B.M., Folberth, C., Foster, I., Gosling, S.N., Haddeland, I., Khabarov, N., Ludwig, F., Masaki, Y., Olin, S., Rosenzweig, C., Ruane, A.C., Satoh, Y., Schmid, E., Stacke, T., Tang, Q., Wisser, D., 2014. Constraints and potentials of future irrigation water availability on agricultural production under climate change. Proc. Natl. Acad. Sci. U. S. A. 111, 3239–44. doi:10.1073/pnas.1222474110

Engström, K., Olin, S., Rounsevell, M. D. A., Brogaard, S., van Vuuren, D. P., Alexander, P., Murray-Rust, D., and Arneth, A.: Assessing uncertainties in global cropland futures using a conditional probabilistic modelling framework, Earth System Dynamics, 7, 893–915, 2016.

Foley, J. A., Defries, R., Asner, G. P., Barford, C., Bonan, G., Carpenter, S. R., Chapin, F. S., Coe, M. T., Daily, G. C., Gibbs, H. K., Helkowski, J. H., Holloway, T., Howard, E. A., Kucharik, C. J., Monfreda, C., Patz, J. A., Prentice, I. C., Ramankutty, N., and Snyder, P. K.: Global consequences of land use, Science, 309, 570-574, 2005.

Friend, A.D., Lucht, W., Rademacher, T.T., Keribin, R., Betts, R., Cadule, P., Ciais, P., Clark, D.B., Dankers, R., Falloon, P.D., Ito, A., Kahana, R., Kleidon, A., Lomas, M.R., Nishina, K., Ostberg, S., Pavlick, R., Peylin, P., Schaphoff, S., Vuichard, N., Warszawski, L., Wiltshire, A., Woodward, F.I., 2014. Carbon residence time dominates uncertainty in terrestrial vegetation responses to future climate and atmospheric $CO_2$. Proc. Natl. Acad. Sci. U. S. A. 111, 3280–3285. doi:10.1073/pnas.1222477110

Harris, I., P. D. Jones, T. J. Osborn, and D. H. Lister. 2014. Updated high-resolution grids of monthly climatic observations – the CRU TS3.10 Dataset. International Journal of Climatology 34:623-642. Hurtt, G. C., Chini, L. P., Frolking, S., Betts, R. A., Feddema, J., Fischer, G., Fisk, J. P., Hibbard, K., Houghton, R. A., Janetos, A., Jones, C. D., Kindermann, G., Kinoshita, T., Goldewijk, K. K., Riahi, K., Shevliakova, E., Smith, S., Stehfest, E., Thomson, A., Thornton, P., van Vuuren, D. P., and Wang, Y. P.: Harmonization of land-use scenarios for the period 1500-2100: 600 years of global gridded annual land-use transitions, wood harvest, and resulting secondary lands, Climatic Change, 109, 117-161, 2011.

Körner, C., 2006. Plant CO2 responses: an issue of definition, time and resource supply. New Phytol. 172, 393–411. Pugh, T.A.M., Müller, C., Arneth, A., Haverd, V., Smith, B., 2016. Key knowledge and data gaps in modelling the influence of CO2 concentration on the terrestrial carbon sink. J. Plant Physiol. 203, 3–15. doi:10.1016/j.jplph.2016.05.001

Lindeskog, M., Arneth, A., Bondeau, A., Waha, K., Seaquist, J., Olin, S., and Smith, B.: Implications of accounting for land use in simulations of ecosystem carbon cycling in Africa, Earth Syst. Dynam., 4, 385-407, doi:10.5194/esd-4-385-2013, 2013.

Luyssaert, S., I. Inglima, and M. Jung. 2009. Global Forest Ecosystem Structure and Function Data for Carbon Balance Research. Global Forest Ecosystem Structure and Function Data for Carbon Balance Research. Data set. Available on-line [http://daac.ornl.gov/] from Oak Ridge National Laboratory Distributed Active Archive Center, Oak Ridge, Tennessee, U.S.A. doi:10.3334/ORNLDAAC/949.

Michaletz, S. T., D. Cheng, A. J. Kerkhoff, and B. J. Enquist. 2014. Convergence of terrestrial plant production across global climate gradients. Nature 512:39-43

Pugh, T.A.M., Müller, C., Arneth, A., Haverd, V., Smith, B., 2016. Key knowledge and data gaps in modelling the influence of CO2 concentration on the terrestrial carbon sink. J. Plant Physiol. 203, 3–15. doi:10.1016/j.jplph.2016.05.001

Reich, P. F., and H. Eswaran. 2002. Global resources. In: Lal, R. (ed.). Encyclopedia

of Soil Science, pp. 607-611. Marcel Dekker, New York.

Rosenzweig, C., Elliott, J., Deryng, D., Ruane, A.C., Müller, C., Arneth, A., Boote, K.J., Folberth, C., Glotter, M., Khabarov, N., Neumann, K., Piontek, F., Pugh, T. a M., Schmid, E., Stehfest, E., Yang, H., Jones, J.W., 2014. Assessing agricultural risks of climate change in the 21st century in a global gridded crop model intercomparison. Proc. Natl. Acad. Sci. U. S. A. 111, 3268–73. doi:10.1073/pnas.1222463110

Sallaba, F., D. Lehsten, J. Seaquist, and M. T. Sykes. 2015. A rapid NPP meta-model for current and future climate and CO2 scenarios in Europe. Ecological Modelling 302:29-41.

Smith, W., Cleveland, C.C., Reed, S.C. & Running, S.W. (2014a). Agricultural conversion without external water and nutrient inputs reduces terrestrial vegetation productivity. Geophys. Res. Lett., 41, 449–455.

Smith, B., Wårlind, D., Arneth, a., Hickler, T., Leadley, P., Siltberg, J., Zaehle, S., 2014b. Implications of incorporating N cycling and N limitations on primary production in an individual-based dynamic vegetation model. Biogeosciences 11, 2027–2054. doi:10.5194/bg-11-2027-2014

Trenberth, K. E., A. Dai, G. van der Schrier, P. D. Jones, J. Barichivich, K. R. Briffa, and J. Sheffield. 2014. Global warming and changes in drought. Nature Clim. Change 4:17-22.

Please also note the supplement to this comment:
https://www.earth-syst-dynam-discuss.net/esd-2016-58/esd-2016-58-AC2-supplement.pdf

―――――――――――――――

[Figure]

[Figure]

**Fig. 1.** Total mean annual NPP for the greater Sahel with runs of different versions of LPJ-GUESS, BME, as well as MOD-17.

[Figure]

**Fig. 2.** Map of the Major Biome Classification based on Reich and Eswaran (2002). The red and green points are the locations of the NPP field-data from Michaletz et al. (2016) and Luyssaert et al. (2009)

[Figure]

**Fig. 3.** Comparison of LPJ-GUESS (N-C) through NPP estimates and NPP field-measurements at the biome level using biome mean NPP values and their standard deviation. The different colours represent MBC

[Figure]

**Fig. 4.** Comparison of BME NPP estimates and NPP field-measurements on biome level using biome mean values as well as biome standard deviation of the means. The different colours represent MBC biomes

[Figure]

**Fig. 5.** The relative contributions of CO2, precipitation and yield gap closure to the increased NPP over the greater Sahel region. Results for CO2 and precipitation are from the RCP 6.0 and yield ga

| Biome (sample size) | Field-data mean NPP [kg C m$^{-2}$ yr$^{-1}$] | LPJ-GUESS mean NPP [kgC m$^{-2}$ yr$^{-1}$] | LPJ-GUESS Q | BME mean NPP [kgC m$^{-2}$ yr$^{-1}$] | BME Q |
|---|---|---|---|---|---|
| TUNDRA Permafrost (78) | 0.30 | 0.44 | 1.46 | 0.24 | 0.79 |
| TUNDRA Interfrost (62) | 0.32 | 0.56 | 1.75 | 0.44 | 1.36 |
| BOREAL Semi-arid (19) | 0.54 | 0.45 | 0.83 | 0.49 | 0.91 |
| BOREAL Humid (405) | 0.42 | 0.62 | 1.48 | 0.56 | 1.32 |
| TEMPERATE Semi-arid (179) | 0.71 | 0.57 | 0.80 | 0.45 | 0.63 |
| TEMPERATE Humid (729) | 0.59 | 0.54 | 0.91 | 0.56 | 0.95 |
| MEDITERRANEAN Warm (36) | 0.95 | 0.78 | 0.83 | 0.52 | 0.55 |
| MEDITERRANEAN Cold (9) | 0.90 | 0.85 | 0.94 | 0.41 | 0.45 |
| DESERT Temperate (18) | 0.31 | 0.17 | 0.56 | 0.09 | 0.28 |
| DESERT Cold (13) | 0.42 | 0.20 | 0.48 | 0.24 | 0.57 |
| TROPICAL Semi-arid (6) | 1.23 | 0.92 | 0.75 | 0.84 | 0.68 |
| TROPICAL Humid (12) | 0.97 | 0.93 | 0.96 | 0.81 | 0.84 |
| Ice (3) | 0.50 | 0.45 | 0.90 | - | - |

**Fig. 6.** Table 1 Comparison between mean biome NPP field-measurements, LPJ-GUESS (N-C), BME NPP estimates; and their Q as model performance measure. Sahel biomes are underlined.

[revised manuscript text omitted]

---

## Author Comment (AC3) · 29 Jun 2017

We thank RC3 for their insightful and constructive comments. What follows is a point-by-point treatment of these comments. Minor changes suggested by RC3 have already been implemented and indicated below.

RC3: 1. The introduction is very well-written and does an excellent job of framing the question and establishing the importance of the work.

Authors' Response: Thank-you very much. It is good to know that the effort we made crafting the introduction does not go unnoticed.

RC3: 2. Page 4, line 27: It is stated that the authors used 0.5 degree climate data from five GCMs, and [CO2] based on four RCPs (Representative Concentration Pathways).

[Figure]

The way it is phased it is unclear which RCPs were used to generate the climate projections for each GCM. The authors should have used climate data derived from runs across the 4 RCPs for each of the 5 models. Please clarify the text if this is the case. If not, please explain more fully why the climate data were not derived for all RCPs.

Authors' Response: We will clarify this in the updated manuscript and explicitly write the RCPs (2.6, 4.5, 6.0 and 8.5) and GCMs (HADLEY, GFDL, IPSL, MIROC and NorESM) that we used. We will also clarify that we ran the model for all RCPs with the selected GCMs.

RC: 3. Robustly representing future NPP trajectories is challenging due to the many potential counteracting feedbacks. The authors show a good fit with LPJ NPP simulations, but do not consider observational data or alternative runs of the LPJ model itself. I recommend further comparison against both census derived yield trends (Rey et al. 2013) and satellite-derived yield trends (Running et al. 2004). For instance, the authors could consider runs in which the CO2 fertilization effect is turned off. Currently all the NPP trends considered in the paper are increasing due to CO2 fertilization (page 9, line 24). This is an area of debate and may be counter to observational data (see Smith et al. 2016, Oberneier et al. 2016, and Ort & Long et al. 2014). Thus, I wonder if a scenario in which CO2 fertilization effects are isolated and removed would be a more realistic lower boundary on what to expect for the region? I would imagine very large increases in the NPP debt (without large irrigation efforts), much larger than what is currently considered in the paper.

Authors' Response: We thank this reviewer for underscoring these issues with the trends, and for suggesting some ways forward. In order to address these comments, we compare BME simulations with MOD-17 data for the period 2000-2006. Thereafter, we compare these results with country level census yield trends (1989-2008) for 4 major crops (rice, maize, wheat, and soybean) from appendix Data S1 of Ray et al. (2013), for some countries found in our study area. Finally, we compare BME trajectories of

[Figure]

NPP with and without CO2 fertilization for all scenarios for the period 2000-2100 in order to account for the fertilization effect. We also direct AR#3 to responses made to Thomas Pugh (R#1) that shed some additional light on these issues.

Our results show that between 2000 to 2006, MOD-17 derived NPP underestimate modelled NPP from BME by 42% (difference of 0.38 kg dry-weight m-2 yr-1), on average (Figure 1). Trends are similar, showing yearly increases of 0.55% (BME) and 0.51% (MOD-17) for the 6 year period of overlap. For the entire length of each series (1970-2006 for BME, and 2000-2010 for MOD-17), slopes indicate yearly increases of 0.40% and 0.62% respectively. Regarding magnitude differences, Ardö (2015) reports that that average annual MOD-17 NPP underestimated LPJ-GUESS carbon version) for Africa for 2000-2010 and attributes this to the fact that autotrophic respiration is considerably higher for MOD17 compared to LPJ-GUESS, due to large temperature sensitivity in the MOD-17 algorithm, differences in the biome-specific parameterizations in MOD17 as well as specification of plant functional types in LPJ-GUESS. Country-level yield trends for rice (Benin, Burkina Faso, Chad, Ghana, Guinea, Guinea-Bissau, Ivory Coast, Liberia, Mali, Nigeria, Senegal, Sierra Leone, Togo), maize (Benin, Burkina Faso, Cameroon, Chad, Ethiopia, Ghana, Guinea, Ivory Coast, Mali, Nigeria, Senegal, Togo), wheat (Cameroon, Chad, Eritrea, Ethiopia, Mali, Mauritania, Niger, Nigeria, Sudan) and soybean (Benin, Burkina Faso, and Nigeria) ranged from -5.98 to 2.80 (mean of -0.002), -0.94 to 4.08 (mean of 1.400), -2.58 to 3.1 (mean of 1.280) and 1.15 to 3.98 (mean of 2.280) respectively. Though the mean of BME and MOD-17 trends fall within most of these ranges for specific crops, it is impossible to generalize due to the number of uncertainties involved in this comparison (e.g. spatial/temporal sampling, and the fact that BME and MOD-17 represent natural vegetation and a mix of natural vegetation and crops, respectively).

If the effect of CO2 fertilization is negated, per capita demand has an equal chance of exceeding per capita supply in 2036 for the SSP2-6.0 scenario (Figure 2a), as opposed to 2043 if CO2 fertilization in included (Figure 3a in the manuscript), with a very high

likelihood of continuous supply shortfall beginning in 2056 as indicated by the black dots (as opposed to 2073 with CO2 fertilization). The effect on all other scenarios is an earlier shift to the onset of supply shortfalls, by about 10 years (Figure 3b). Supply shortfalls with high likelihood of occurrence (black dots showing non-overlapping 95% confidence intervals) are similarly shifted, and occur with greater consistency and frequency.

Therefore the results in our manuscript provide an upper bound for NPP supply, and the analysis of the balance of supply and demand contained there may provide an optimistic account of the timing and duration of shortfalls across all scenarios.

We can provide an abbreviated account of this in a revised version of the manuscript.

RC3: 4. Page 5, line 4: When the fractional agricultural landcover estimates from Hurtt et al (2011) were applied, was it assumed that natural and agricultural NPP were similar? If so, this assumption should be revisited after considering differences between agricultural vs. natural NPP for the region. For instance, the authors could compare census based estimates of crop productivity with their estimates as a reality check. Smith et al. 2014 (see reference below), found that agricultural productivity for the region is significantly lower than natural productivity. If this potential reality is not considered, then the scenarios in this manuscript may be overly optimistic.

Authors' Response: Yes, it is true that we considered the NPP to be equal for all land covers. However, by using a relatively low (0.235) harvest index, we have implicitly accounted for at least some of that lower productivity. We will add to the discussion that our estimates of cropland NPP are in the upper range. Thank you for the reference.

RC3: 5. I would recommend revisiting all crop allocation parameters based on those reported by Monfreda et al. (2008). Given the high variability in crop specific harvest fractions, it seems it may be necessary to parameterize the model for each individual crop grown in the region.

Authors' Response: It is true that the representation of crop allometry in the current setup is simplistic and cannot likely capture all the variability present in agricultural landscapes in the region. But the approach taken here, by having one root-to-shoot ratio and harvest index for all crops in the region is consistent with the underlying theme of the study, e.g. have a simplistic modelling framework to be able to explore supply-demand outcomes with a minimum of input data. Furthermore, as we do not know how these cropping systems would develop in the future across the scenarios, we think a simple approach is the safest bet.

RC3: 6. Page 12 line 24-26: This statement is not representative of the literature (see below references). I would suggest more nuanced discussion of the potential limitations of the supply approach used in this analysis. For instances, how much did CO2 fertilization drive increases? How uncertain are the precipitation estimates? Were nutrient constraints considered and if so what are the management implications? If not, how might nutrient constraints limit NPP? How will increases in atmospheric water demand (Vapor pressure deficit) affect yields and productivity? Could increased drought and desertification also represent a potential scenario had the CO2 sensitivity been adjusted? The way that this section is currently written is a gross over extension of the simplified NPP modeling that the paper is based on.

Authors' Response: Thank-you very much for directing us to the salient literature. In response to this, we suggest this following alteration to the section 'Additional Perspectives' and present it here:

"Our finding that supply increases for all SSP-RCP scenarios, partly due to increasing rainfall and CO2 fertilization. However, recent observational evidence suggests that the effect of CO2 fertilization on plant growth may be constrained by counteracting feedbacks associated with increasing atmospheric moisture demand and nutrient availability (e.g. Smith et al., 2016; Wieder et al. 2015). For example, NPP is reduced under warmer and dryer conditions due to moisture stress, particularly in temperate and arid ecosystems. Future trends NPP trends in the Sahel could therefore be strongly deter-

mined by changes in the frequencies of wet years versus dry years, with the dry years counteracting the CO2 fertilization effect. Furthermore, nutrient supply rates may not be able to keep up with extra demand associated with CO2 fertilization, and leading to a depletion of soil nutrients, as current evidence suggests. This could also curtail the CO2 fertilization effect, particularly in the more southerly parts of our study area, where nutrients tend to become a limiting factor.

All of this suggests that that the current trend of Sahel greening identified from satellite sensor-based mapping studies (e.g. Eklundh and Olsson, 2003; Hickler et al., 2005; Seaquist et al., 2009) may may not continue into the future, even if with increasing rainfall.

Livestock mobilization is one way local populations generally employ to manage risk (e.g. Herrmann et al. (2014). This strategy may help regulate supply shortfalls locally, and over the short term. Even if the Sahel were to continue to green (increase in NPP supply) this would not necessarily imply an increase in the amount of usable NPP or an enhancement in health and well-being. Recent studies in the Sahel show that much of the recent greening, at least in some regions, is due to undesirable shifts in species composition (e.g. Herrmann et al. (2014)), reductions in biodiversity and an increases in woody biomass (e.g. Brandt et al. (2015))."

RC3: Minor Comments: 1. Page 3, Line 17: "Three different aggregation levels are considered, including Sahel, the country, and the local". Please define what is meant by local level. Pixel level? What resolution? 2. Page 12 line 12: missing end of parentheses.

Authors' Response: with the term local level, we mean the level of the cell, with a resolution of 0.5 x 0.5 degrees. This allows the inspection of supply and demand variations at a sub-national level, across cells. We have already clarified this in our manuscript and have fixed the typo.

CAPTION FOR FIGURE 2

The per capita NPP supply, demand and balance for the entire Sahel region over the time period without CO2 fertilization. 2a) shows NPP supply (red) and demand (blue). The solid curves illustrate the mean of the SSP2-RCP6.0 combination. The dashed blue curves show supply uncertainty (95% confidence interval around the mean) based on the five GCMs NPP results. The dashed red curves show demand uncertainty (95% confidence interval around the mean) based on the uncertainty related to the interpretation and quantification of SSP2. 2b) shows the different magnitudes of the NPP balance and the varying onsets of shortage across all SSP-RCP combinations. Black dots illustrate years with a shortage outside of the 95% confidence intervals. The combinations are grouped according to the socio-economic scenarios (y-axis). The RCPs are ordered from low to high radiative forcing in each SSP group. The temporal trajectory is shown along the x-axis and the colouring indicates the sign of the annual NPP balance. Blues show a surplus of the NPP supply while yellow to red represent small to very large NPP shortages (i.e. the gap between supply and demand). SSP-RCP combinations in bold indicate the most likely SSP-RCP pairs based on Tables 3 and 4 of Engström et al. (2016b).

References in our responses

Brandt, M., Mbow, C., Diouf, A. A., Verger, A., Samimi, C., and Fensholt, R.: Ground- and satellite-based evidence of the biophysical mechanisms behind the greening Sahel, Global Change Biology, 21, 1610-1620, 2015.

Eklundh, L. and Olsson, L.: Vegetation index trends for the African Sahel 1982–1999, Geophysical Research Letters, 30, 8, 1430, 2003.

Herrmann, S. M., Sall, I., and Sy, O.: People and pixels in the Sahel: a study linking coarse-resolution remote sensing observations to land users' perceptions of their changing environment in Senegal, Ecology and Society, 19, 2014.

Hickler, T., Eklundh, L., Seaquist, J. W., Smith, B., Ardo, J., Olsson, L., Sykes, M. T., and Sjostrom, M.: Precipitation controls Sahel greening trend, Geophysical Research

Letters, 32, L21415, 2005.

Obermeier, W.A., Lehnert, L.W., Kammann, C.I., Müller, C., Grünhage, L., Luterbacher, J., Erbs, M., Moser, G., Seibert, R., Yuan, N. & Bendix, J. (2016). Reduced $CO_2$ fertilization effect in temperate C3 grasslands under more extreme weather conditions. Nat. Clim. Chang., 7, 137–142.

Ray, D.K., Mueller, N.D., West, P.C. & Foley, J.A. (2013). Yield Trends Are Insufficient to Double Global Crop Production by 2050. PLoS One, 8, e66428.

Seaquist, J. W., Hickler, T., Eklundh, L., Ardo, J., and Heumann, B. W.: Disentangling the effects of climate and people on Sahel vegetation dynamics, Biogeosciences, 6, 469-477, 2009.

Smith, W.K., Reed, S.C., Cleveland, C.C., Ballantyne, A.P., Anderegg, W.R.L., Wieder, W.R., Liu, Y.Y. & Running, S.W. (2016). Large divergence of satellite and Earth system model estimates of global terrestrial $CO_2$ fertilization. Nat. Clim. Chang., 6, 306–310

Wieder, W.R., Cleveland, C.C., Smith, W.K. & Todd-Brown, K. (2015). Future productivity and carbon storage limited by terrestrial nutrient availability. Nat. Geosci., 1–5.

Please also note the supplement to this comment:
https://www.earth-syst-dynam-discuss.net/esd-2016-58/esd-2016-58-AC3-supplement.pdf
* * *
[Figure]

**Fig. 1.** Total mean annual NPP over the greater Sahel region estimated by BME (red) and MODIS (blue) with trends for the two estimates (dashed lines).

[Figure]

**Fig. 2.** Please see end of main body of text, just before references, for caption (too long to place here)

**Supplement:**

[revised manuscript text omitted]

---

## Author Response (AR1)

We thank Dr. Pugh for providing constructive and insightful comments on our manuscript. Below, we provide a point-by-point response to each comment, together with a new marked-up version of our manuscript.

**Thomas Pugh:** I presume the LPJ-GUESS simulations used to calibrate the BME model were potential natural vegetation (would help if this was explicitly stated)? In which case I wonder how effectively NPP of natural ecosystems can be used as a proxy for NPP of agricultural ones. NPP is not independent of plant type, and the distinction between natural vegetation, which may well be woody, and cereal and pasture vegetation may be particularly relevant in the Sahel, where the deeper roots of trees may have access to water resources that herbaceous plants cannot use. Can the authors demonstrate that such effects are not large, both in the LPJ-GUESS model and also based on any observations in the Sahel or analogous ecosystems?

**Authors' Response:** Page 4, lines 13- 14 of our revised manuscript now reads "LPJ-GUESS is a state-of-the-art dynamic global potential natural vegetation model that incorporates carbon and nitrogen interactions (Smith et al., 2014)."

In order to test how effectively the NPP of natural ecosystems can be can be used as a proxy for the NPP of agricultural ones we ran LPJ-GUESS managed land (C-N version) for the period 1970 to 2010 and compared this to LPJ-GUESS (C-N, and used to develop BME), for the greater Sahel region defined in our manuscript (note that for comparison purposes, we also provide runs of LPJ-GUESS, BME and MOD-17 and LPJ-GUESS C only). Accordingly, we provide some new discussion in Appendix A of our revised manuscript, to take into consideration these aspects (see new Section A.2.2, p. 30, lines 12-16 and p. 31, lines 1-6.

"In order to test how effectively the NPP of natural ecosystems can be can be used as a proxy for the NPP of agricultural ones we ran LPJ-GUESS managed land (Olin et al., 2015) for the period 1970 to 2006 and compared this to LPJ-GUESS (used to develop BME) for the entire Sahel region. The results (see Fig. A5) of this experiment show that mean NPP derived from LPJ-GUESS ml over the region underestimates mean NPP derived from BME by 0.7% (0.006 dry-weight m$^{-2}$ yr$^{-1}$) and LPJ-GUESS by 2.4% (0.020 kg dry-weight m$^{-2}$ yr$^{-1}$), though all models show similar levels of interannual variability and trend (see Fig. A5). The implication of this experiment is that there is a demonstrable reduction in NPP when land management is taken into consideration, but the effect is relatively minor. Lindeskog et al. (2013) show that LPJ-GUESS managed land (C-version) overestimated actual yield derived from FAO country-level crop statistics and Smith et al. (2014b) also report that natural systems are more productive than agricultural systems in sub-Saharan Africa. We conclude with that possibility that our results are in the upper range for NPP found in the Sahel."

Note that we do not use the 'C-N' designation for specifying LPJ-GUESS version in the manuscript, as it is stated from the beginning that this is the version we use to develop the BME, based on Smith et al. (2014).

[Figure]

**Fig. A5.** Regional annual NPP Annual means of NPP for BME, LPJ-GUESS, LPJ-GUESS C (carbon only) and LPJ-GUESS ml (managed land) (1970 to 2006) and MODIS (2000-2010) for the greater Sahel region.

**Thomas Pugh:**  Whilst the BME model is evaluated against LPJ-GUESS, any evaluation of the extent to which LPJ-GUESS can accurately represent actual NPP in the Sahel region is lacking. The references given (pg. 4 l. 14) did not address this ecosystem and also used a version of the model lacking carbon-nitrogen interactions, which leads to quite different vegetation simulations for the Sahel (Smith et al., 2014). Evaluation of the model response for the Sahel is necessary to give credence to the comparisons of supply and demand, which strongly depend on simulated absolute values for NPP. Whilst there is no gold-standard NPP (or GPP) dataset to compare against, comparison against NPP from the ESMs used to assess uncertainty, along with comparison of GPP against the alternative approaches of Jung et al. (2011) and Zhao et al. (2005) could go a long way towards increasing confidence. Alternatively (or additionally), FAO yield statistics could be used to evaluate the "yields" calculated here. Although none of these sources of comparison are likely to be low in uncertainty in the Sahel region, as it stands we have no idea how well LPJ-GUESS performs in this region - and current DGVMs cover a wide range of possibilities at regional scales (Sitch et al., 2015).

**Authors' Response:** Firstly, thank-you very much for highlighting these validation issues. In a revamped and much extended version of Appendix A.2.1 (pp. 24 – 29), we now include a global-level biome-by-biome validation of both LPJ-GUESS (C-N) and BME where we highlight the results for Sahel biomes):

**"A.2.1 Biome Level Model Validation**

We validate biome-level LPJ-GUESS and BME performance for estimating NPP of natural vegetation with NPP field-measurements from Michaletz et al. (2016) and Luyssaert et al. (2009) (see Sallaba et al., 2015) for the Major Biome Classification of Reich and Eswaran (2002) including the biomes found in the Sahel (desert temperate, tropical semi-arid and tropical humid – no observations were available for desert tropical). Note that since only two observations were available for our study area (see Fig. A1) this evaluation demonstrates the ability of both LPJ-GUESS and BME to replicate NPP for Sahel biomes found elsewhere in the world.

Before we combined the Michaletz et al. (2016) and Luyssaert et al. (2009) datasets, we removed sites with no records of combined above- and below-ground NPP measurements. After we merged the data, we checked the final assembly of NPP measurements for duplicates and removed them. The final dataset consists of 1561 samples (i.e. 1247 samples from Michaletz et al. (2016) and 314 samples from Luyssaert et al. (2009)) representing total NPP measurements across the terrestrial biosphere (sample sizes are 18, 6, and 12 for Sahel biomes of desert temperate, tropical semi-arid and tropical humid, respectively) from 1959-2006. Both LPJ-GUESS and BME were driven with CRU TS 3.21 climate data (Harris et al. 2014, Trenberth et al. 2014) that has global coverage across the time period.

We calculated mean values of the NPP field-measurements and the modelled NPP estimates located in the respective biomes, following Smith et al. (2014b). We aggregated to the biome-level to account for the difference in scale between in situ NPP measurements and modelled grid cell NPP estimates (being grid cell averages).

Finally, we determined the overall model performance, biome-by-biome, with the coefficient of determination ($R^2$ value) and the root mean square error (RMSE). Additionally, we investigated model agreement with performance ratios (hereafter referred to as 'Q') by dividing mean biome NPP estimates (for both models) with mean biome NPP observations. Model overestimation in comparison to in situ NPP measurements is indicated by Q > 1 and underestimation by Q < 1. Good model performance is classified with a Q range between 0.9-1.1 assuming an error of ± 10% following Sallaba et al. (2015). However, we further defined an acceptable model performance error range of ±20% (i.e. Q = 0.8-1.25) given the limitations of using LPJ-GUESS standard modelling protocol, PNV and CRU climate observations, and especially the simplicity of BME.

[Figure]

**Legend**

[revised manuscript text omitted]

We also include the comparison with the results of the MODIS processing stream p. 31 lines 7 – 18:

"We also compare total yearly means of NPP from BME and LPJ-GUESS to NPP derived from the MOD17A3 processing stream (using MOD17A3 data obtained from the NASA Earth Observation System repository at the University of Montana at www.ntsg.umt.edu) for the period 2000 to 2006 for the greater Sahel region (Running, 2004). We averaged resampled MODIS NPP from 1km to the spatial resolution of the BME estimates (0.5 x 0.5 degrees) and excluded urban areas. We removed below-ground NPP and plant parts unable to be consumed by applying the same R:S and harvest index as described in Section 2.1.1. Lastly, we calculated mean values of MODIS NPP estimates from 2000 to 2010 for each grid cell covering the study area. Our results show that between 2000 and 2006 MODIS-derived NPP underestimate BME-derived NPP by 42% (difference of 0.38 kg dry-weight $m^{-2}$ $yr^{-1}$), on average (Figure A5). Ardö (2015) also reports that that average annual MODIS NPP underestimates LPJ-GUESS (C version only) for Africa for 2000-2010 and attributes this to the fact that autotrophic respiration is considerably higher for MODIS NPP compared to LPJ-GUESS, due to large temperature sensitivity in the MODIS algorithm, differences in the biome-specific parameterizations for MODIS as well as specification of plant functional types in LPJ-GUESS."

**Thomas Pugh:** On the theme of evaluation. I'm not clear from the manuscript if PLUM land-use simulations are normalised in some way to the dataset of Hurtt et al. (2011) in 2000, or if they represent a purely "PLUM version" of the Sahel land-use in 2000. The former would raise the question of how much the model drifts from the observed towards its preferred state at the start of the simulations. The latter suggests the need for a comparison of the PLUM initial state with current observation-based estimates (such as Hurtt et al., 2011). I realise there are significant difficulties in modelling actual land-use, but surely the size of any discrepancies and the resulting implications should be discussed?

**Authors' Response:** The Hurtt et al. (2011) data for the year 2000 is used as basis, as stated on p. 5 lines 12-15:

"We estimated crop- and grassland scaling factors for each country by dividing the PLUM-predicted land-use area with the total land-use area provided by the Hurtt et al. (2011) dataset (Table C1). We then applied the scaling factors to the Hurtt et al. (2011) land-use data and multiplied the resulting crop- and grassland areas with the NPP estimates to obtain annual $NPP_{cereal\_supply}$ and $NPP_{grazing\_supply}$ (kg DW $cell^{-1}$ $yr^{-1}$)."

We have also added the country specific factors (along with country-specific land area from FAO) to Table C1 of the appendix:

**Table C1** Per capita NPP supply and demand of countries in the greater Sahel region for 2000 and 2050. Portions of food and feed (including grazing) in per capita NPP demand for SSP2-RCP6.0. All NPP is given in dry-weight (DW). Hurtt:PLUM scaling factors and land areas (from FAO) are also included.

| Hurtt:PLUM scaling factors | Land area from FAOSTAT |
|---|---|
| **2000** | 1000 ha |
| **0.89** | 11062 |
| **0.90** | 27360 |
| **1.04** | 47271 |
| **1.00** | 125920 |
| **0.00** | 2318 |
| **1.10** | 10100 |
| **0.98** | 1000000 |
| **1.58** | 1000 |
| **1.03** | 22754 |
| **1.73** | 24572 |
| **1.25** | 2812 |
| **0.98** | 31800 |
| **0.91** | 9632 |
| **0.97** | 122019 |
| **0.97** | 103070 |
| **1.01** | 126670 |
| **1.04** | 91077 |
| **0.74** | 19253 |
| **0.99** | 7162 |
| **0.98** | 237600 |
| **1.10** | 5439 |
| **-** | - |

We have also added the following discussion, p. 17, lines 32-33 and p. 18, lines 1-3.

"Finally, we note that country-specific scaling factors used to convert PLUM output to per pixel changes using the Hurtt et al. (2011) data set for the year 2000 did not depart substantially from 1 (scaling factors for the larger countries were all within 10%, and the area weighted mean of the scaling factors was 0.95), but a few smaller countries in West Africa diverge by more than 25% (<0.80 or > 1.25) (see Table C1). We expect these to have only marginal influence on the results at the regional level, but could have a larger impact on localities along the West African coast  (Fig. 4 and Fig. B1)."

**Thomas Pugh:** pg. 2 l. 31. Why does a 31% population increase lead to a 100% increase in NPP requirement? What information is missing here?

**Authors' Response:** p2 lines 31-33 now read:

"Since the NPP demand increased at an annual rate of 2.2% over the period while the supply was near constant, the near doubling in NPP demand implies, in relative terms, that there was less NPP supply to service the increase in population."

**Thomas Pugh:** pg. 6 l. 16. I'm confused about the cropland cover, I thought it was taken from PLUM? How is Hurtt being used here?

**Authors' Response:** Please refer the previous response.

**Thomas Pugh:** pg. 6 l. 23. Surely the total amount of NPP for human appropriation must be the sum of $NPP_{cereal\_demand}$ and $NPP_{grazing\_demand}$, not just $NPP_{cereal\_demand}$ alone? As parts of both cereal and grazing demand contribute to animal raising, the current definition is inconsistent. Was it meant to be something like "total amount of annual NPP for human appropriation via cropland"?

**Authors' Response:** this clause, on p. 7, line 4 now reads

"total amount of annual NPP for human appropriation via cropland."

Indeed, we explicitly distinguish between the demand of cereal and pasture products. Cereal demand is given in Equation 1 of the manuscript, while grazing demand is given in Equation 9 (not Equation 8 as stated in the first version, Appendix A3 – we have changed this too). Cereal-based and grazing-based supply-demand balances are then computed separately. They are then summed according to Table 1 in the manuscript in order to determine final balances of supply and demand of NPP.

**Thomas Pugh:** The SSP-RCP scenario likelihoods seem rather important. Rather than referring the reader to another paper, maybe you could include them in this analysis? For instance along the right y-axis of Fig. 3b?

**Authors' Response:** -We now provided Table 1 in our revised manuscript, referred to on p. 7 line 17. Table 1 shows the scenario likelihoods, and is the same as Table 4 found in Engström et al. (2016b). Note that these likelihoods refer to the most consistent SSP-RCP combinations (e.g. it is more likely that the sustainability assumptions for SSP1 would yield greenhouse gas concentrations in line with RCP4.5/6 rather than RCP2.6/8.5).

**Table 1** Scenario matrix translated into quantitative probabilities (see also Engström et al. (2016b).

|       | RCP 2.6 | RCP 4.5 | RCP 6  | RCP 8.5 | Sum |
|-------|---------|---------|--------|---------|-----|
| **SSP1** | 0.0909  | 0.4545  | 0.4545 | 0.0000  | 1   |
| **SSP2** | 0.0000  | 0.0909  | 0.6818 | 0.2273  | 1   |
| **SSP3** | 0.0000  | 0.1667  | 0.5000 | 0.3333  | 1   |
| **SSP4** | 0.0000  | 0.3704  | 0.5556 | 0.0741  | 1   |
| **SSP5** | 0.0000  | 0.0741  | 0.3704 | 0.5556  | 1   |

**Thomas Pugh:**  pg. 7 l. 29-33. This text reads as if it was originally located before the first paragraph of 2.1.3, and some of the text would seem to be more logically located there, where this likelihood matrix is first mentioned.

**Authors' Response:** This information is now moved to the suggested location, p. 7 lines 18-21.

**Thomas Pugh:** pg. 9 l. 11. I would say that the shortfalls in SSP5-RCP6.0 and SSP5-RCP8.5 are pretty sustained. They just don't run to the end of the century. Consider rephrasing? More generally, regarding the discussion of "shortfalls", it seems strange that you only consider shortfalls to occur when the 95% confidence limits do not overlap (and demand is higher of course). To my mind this lack of overlap of the confidence limits suggests very high likelihood of shortfalls, but the best guess result shows shortfalls occurring for a larger number of scenarios. For instance, on pg. 11, l. 26 it is stated that "statistically significant shortages never develop" in the context of SSP1, but that doesn't seem quite right. Assuming non-skewed distributions of uncertainty (big assumption, I know), then when the best estimate of demand exceeds the best estimate of supply there is a more than even chance of shortages occurring, but it's not possible to say with high certainty that a shortage will occur until the 95% limits no longer overlap. Consider rephrasing also?

**Authors' Response:** These items have been rephrased and here we produce a rewritten portion of the results, found on p. 10, lines 6-19:

"Per capita demand exceeds supply in the early 2040s for SSP2-RCP6.0 after which a very high likelihood for shortfalls begins in 2070 (see black dots in Fig. 3a showing non-overlapping 95% confidence limits). By 2050, per capita demand almost doubles while per capita supply drops by almost 30% for the same scenario. Across the scenarios, differences in the timing of the start of persistent supply shortfalls with high statistical certainty are observed (see black dots in Fig. 3b). Three of these high likelihood shortfalls begin at 2050 or before (SSP5 scenarios – see black dots in Fig. 3b) while an additional six display shortfalls with high certainty by the end of the 21[st] century (black dots in Fig. 3a, b). Out of these nine, two scenarios never achieve a sustained run of shortfalls (SSP2-RCP6.0, SPP2-RCP8.5). In total, there is better than an even chance for shortfalls before 2050 for 9 scenarios (exceptions are SSP1-RCP2.6, SSP1-RCP6.0, and all SSP4 scenarios.

Variations in the timing of onset and end of supply shortfalls are generally greater between the SSPs than between the RCPs (Fig. 3b). For SSP2 and SSP3 scenarios, onsets of high likelihood supply shortfall range from the early 2050s to the mid-2070s (even chance from late 2030s to early 2050s). The SSP5 family shows the largest deficits of high likelihood shortfalls beginning in the 2040s-2050s (even chance from the early 2030s), and after several decades of deepening begin to diminish in the

2080s. Shortfalls with high certainty never emerge for SSP1 (even chance from the early 2050s) while the SSP4 scenarios show sustained but diminishing surplus throughout."

**Thomas Pugh:** pg. 9 l. 22. Reference to Table 3 here?

**Authors' Response:** We now reference this table on p. 11 line 6 of our revised manuscript. Note that a new Table 1 means that Table 3 of the original manuscript is now Table 4 in the revised manuscript.

**Thomas Pugh:** pg. 12 l. 3. Regarding, "so strong efforts should be made to reduce these gaps", this is too simplistic. Efforts to close yield gaps have other environmental and socio-economic consequences which are not addressed here, meaning that this statement cannot be supported by the presented evidence. I suggest to remove this recommendation. Going beyond this however, can you say anything about the potential additional yield by closing yield gaps in this region, and whether such efforts could alleviate the shortages simulated? Maybe PLUM can provide the necessary data?

**Authors' Response:** The recommendation has now been removed. We have now added the following information on p. 14, lines 3-5 of our revised manuscript:

"The closure of yield gaps by 2050 (for scenario SSP2-6.0) would result in a change in mean per capita NPP balance from -107 kg DW yr-1 (see Table 3) to 9 kg DW yr-1. Though the balance for many countries will still be negative, the magnitudes of shortfalls could be reduced. Thus, closing yield gaps in the region could indeed alleviate the simulated shortages."

**Thomas Pugh:** pg. 12 l. 24. Where is the attribution of supply increases to additional rainfall and CO2 fertilisation shown in the results?

**Authors' Response:** We have conducted this experiment and added the following to p. 15, lines 14-17 of our revised manuscript:

"In order to isolate the $CO_2$ (rainfall) effect on NPP increase for RCP6.0, we compared a simulation where rainfall ($CO_2$) was held constant with a simulation where both were held constant for the period 2000-2050 for all GCMs. We found that supply increases mostly due to $CO_2$ fertilization (see Fig. B2), with very little attributed to rainfall. However, yield gap closure from SSP2 contributes most to NPP increase (Fig. B2)."

The $CO_2$ fertilization effect increases with the magnitude of climate change and explains the smaller shortages in SSP-RCP8.5 scenarios compared to SSP-RCP4.5 scenarios (Fig. 3b)."

[Figure]

**Fig. B2**. The relative contributions of $CO_2$, precipitation and yield gap closure to the increase in NPP over the greater Sahel region, 2000-2050. Results for CO2 and precipitation are from RCP 6.0 and yield gap is from SSP2. Simulated climate and $CO_2$ effects shown here are mean effects over the five GCMs (GFDL,MIROC,Hadley,NorESM, IPSL).

We have now removed the original statement alluding to the fact that NPP increases are attributed to CO2 fertilization and rainfall from this section entirely.

**Thomas Pugh:** pg. 13 l. 7. The relative attribution of supply growth to climate/co2 and closure of yield gaps would be very informative, allowing the results to be interpreted more subtly. Your approach seems to be suitable to make this isolation.

**Authors' Response:** See previous comment.

**Thomas Pugh:** pg. 13 l. 12. I would take the opposite view. The extent to which models appropriately represent CO2 fertilisation is not clear, and the difference in NPP trends between models is very large (e.g. Friend et al., 2014; Körner, 2006; Pugh et al., 2016; Rosenzweig et al., 2014). Therefore, I think it is fair to say that we have no more confidence in the trends than we do in the absolute levels. Moreover, the reference here to Fig. A2 does nothing to support the point, as the point of comparison is an LPJ-GUESS simulation, not observations.

**Authors' Response:** Thanks very much for highlighting issues with the trends. Please see our response to RC3 for a broader discussion of the trends (e.g. responses to comments #3 and #6). We have now removed this sentence entirely from this section and have modified Fig. A2 (now Fig. A5 in the revised manuscript (see a previous comment), and in order to meet this critique (and that of RC3), we have added the following to Section 4.6 (Uncertainties), p. 17, lines 13-31. This section also refers to our new Fig. B7 where we rerun supply-demand scenarios with CO2 turned off:

"Additional uncertainty exists with respect to the total magnitude and trends of simulated NPP supply, given the lack of ground truth for the region, and that differences in NPP trends between other models is very large (e.g. Friend et al., 2014; Körner et al., 2006; Pugh et al., 2016; Rosenzweig et al., 2014). Indeed, recent observational evidence suggests that the effect of $CO_2$ fertilization on plant growth may be constrained by counteracting feedbacks associated with increasing atmospheric moisture demand and nutrient availability (e.g. Smith et al., 2016; Wieder et al. 2015). For example, NPP is reduced under warmer and dryer conditions due to moisture stress, particularly in temperate and arid ecosystems. Future trends NPP trends in the Sahel could therefore be strongly determined by changes in the frequencies of wet years versus dry years, with the dry years counteracting the $CO_2$ fertilization effect. Furthermore, nutrient supply rates may not be able to keep up with extra demand associated with $CO_2$ fertilization, and leading to a depletion of soil nutrients, as current evidence suggests. This could also curtail the $CO_2$ fertilization effect, particularly in the more southerly parts of our study area, where nutrients tend to become a limiting factor. We performed a simple experiment negating the $CO_2$ fertilization effect in order to gauge its impact on supply-demand balance on all scenarios. For the SSP2-RCP6.0, per capita demand has an equal chance of exceeding per capita supply in 2036 for the SSP2-6.0 scenario as opposed to 2043 if $CO_2$ fertilization in included (Fig. B7), with a very high likelihood of continuous supply shortfall beginning in 2056, as opposed to 2073 with $CO_2$ fertilization. The effect on all other scenarios is an earlier shift to the onset of supply shortfalls, by about 10 years, compared to Fig. 3b (see Fig. B7). Supply shortfalls with high likelihood of occurrence (black dots showing non-overlapping 95% confidence intervals) are similarly shifted, and occur with greater consistency and frequency. All of this suggests that the NPP increases found in our current analysis are likely optimistic, due the potential overestimation of the $CO_2$ fertilization effect, as well as the fact that BME is based on potential natural vegetation."

[Figure]

**Fig. B7** Per capita NPP supply, demand and balance for the greater Sahel (2000-2100) without $CO_2$ fertilization. **B7a)** shows NPP supply (red) and demand (blue). The solid curves illustrate the mean of the SSP2-RCP6.0 combination. The dashed blue curves show supply uncertainty (95% confidence interval around the mean) based on the five GCMs NPP results. The dashed red curves show demand uncertainty (95% confidence interval around the mean) based on the uncertainty related to the interpretation and quantification of SSP2. **B7b)** shows the different magnitudes of the NPP balance and the varying onsets of shortage across all SSP-RCP combinations. Black dots illustrate years with a shortage outside of the 95% confidence intervals. Combinations are grouped according to the socio-economic scenarios (y-axis). The RCPs are ordered from low to high radiative forcing in each SSP group. The temporal trajectory is shown along the x-axis and the colouring indicates the sign of the annual NPP balance. Blues show a surplus of the NPP supply while yellow to red represent small to very large the gaps between supply and demand). SSP-RCP combinations in bold indicate the most likely SSP-RCP pairs based on Table 1.

**Thomas Pugh:** pg. 13 l. 22. You could also briefly mention irrigation water availability projections here (Elliott et al., 2014).

**Authors' Response:** We have now appended our manuscript with the following, found on p. 16 lines 3-6:

"However, Elliott et al. (2014) underscore that freshwater limitations in the dryer regions of the globe could limit agricultural production, and even lead to the reversion of irrigated farmland to rainfed farmland thereby negatively affecting food production."

**Thomas Pugh:** pg. 1, l. 20. "surplus, while" pg. 1 l. 23. "diet" pg. 2 l. 13. "global food security is not ensured" pg. 2 l. 16. "world, where" pg. 2 l. 19. "own land, where", also full stop missing after "pastoralism" pg. 4 l 32. Should "estimates to the total area", read " estimates to sum over the total area"? I don't think you translated NPP to total area literally? pg. 5 l. 22. Replace "Furthermore" with "Therefore" pg. 5 l. 32. "choice, and the" pg. 6 l. 13, 14, 20. "Fig. 2" should be "Fig. 1"? Also there are several boxes in red in Fig. 1 so "box outlined in red" is of limited use, and the distinction between cereal and pasture products can't be seen in the picture. pg. 8 l. 4. "Hence, one" pg. 10 l. 2. Only two countries are listed. pg. 12 l. 26. "mobilization is one method local" pg. 12 l. 31. "increase" pg. 14 l. 2. I think this would read better as "the Sahel is likely to experience NPP shortages in most SSP scenarios due to" pg. 14 l. 7. Reference formatting. pg. 14 l. 25. "show" rather than "assume"? pg. 15 l. 2. "will outstrip supply during the 21st century". pg. 15 l.12. "unfolds, a relatively"

**Authors Response:** These typos have been fixed.

**References in our responses**

Abdi, A. M., Seaquist, J., Tenenbaum, D. E., Eklundh, L., and Ardö, J.: The supply and demand of net primary production in the Sahel, Environmental Research Letters, 9, 094003, 2014.

Ardö, J. Comparison between remote sensing and a dynamic vegetation model for estimating terrestrial primary production of Africa. Carbon Balance and Management, 10(8), 2015.

Ceccato, P., Cressman, K., Giannini, A., and Trzaska, S.: The desert locust upsurge in West Africa (2003-2005): Information on the desert locust early warning system and the prospects for seasonal climate forecasting, International Journal of Pest Management, 53, 7-13, 2007.

Elliott, J., Deryng, D., Müller, C., Frieler, K., Konzmann, M., Gerten, D., Glotter, M., Flörke, M., Wada, Y., Best, N., Eisner, S., Fekete, B.M., Folberth, C., Foster, I., Gosling, S.N., Haddeland, I., Khabarov, N., Ludwig, F., Masaki, Y., Olin, S., Rosenzweig, C., Ruane, A.C., Satoh, Y., Schmid, E., Stacke, T., Tang, Q., Wisser, D., 2014. Constraints and potentials of future irrigation water availability on agricultural production under climate change. Proc. Natl. Acad. Sci. U. S. A. 111, 3239–44. doi:10.1073/pnas.1222474110

Engström, K., Olin, S., Rounsevell, M. D. A., Brogaard, S., van Vuuren, D. P., Alexander, P., Murray-Rust, D., and Arneth, A.: Assessing uncertainties in global cropland futures using a conditional probabilistic modelling framework, Earth System Dynamics, 7, 893–915, 2016.

Foley, J. A., Defries, R., Asner, G. P., Barford, C., Bonan, G., Carpenter, S. R., Chapin, F. S., Coe, M. T., Daily, G. C., Gibbs, H. K., Helkowski, J. H., Holloway, T., Howard, E. A., Kucharik, C. J., Monfreda, C., Patz, J. A., Prentice, I. C., Ramankutty, N., and Snyder, P. K.: Global consequences of land use, Science, 309, 570-574, 2005.

Friend, A.D., Lucht, W., Rademacher, T.T., Keribin, R., Betts, R., Cadule, P., Ciais, P., Clark, D.B., Dankers, R., Falloon, P.D., Ito, A., Kahana, R., Kleidon, A., Lomas, M.R., Nishina, K., Ostberg, S., Pavlick, R., Peylin, P., Schaphoff, S., Vuichard, N., Warszawski, L., Wiltshire, A., Woodward, F.I., 2014. Carbon residence time dominates uncertainty in terrestrial vegetation responses to future climate and atmospheric CO2. Proc. Natl. Acad. Sci. U. S. A. 111, 3280–3285. doi:10.1073/pnas.1222477110

Harris, I., P. D. Jones, T. J. Osborn, and D. H. Lister. 2014. Updated high-resolution grids of monthly climatic observations – the CRU TS3.10 Dataset. International Journal of Climatology 34:623-642.

Hurtt, G. C., Chini, L. P., Frolking, S., Betts, R. A., Feddema, J., Fischer, G., Fisk, J. P., Hibbard, K., Houghton, R. A., Janetos, A., Jones, C. D., Kindermann, G., Kinoshita, T., Goldewijk, K. K., Riahi, K., Shevliakova, E., Smith, S., Stehfest, E., Thomson, A., Thornton, P., van Vuuren, D. P., and Wang, Y. P.: Harmonization of land-use scenarios for the period 1500-2100: 600 years of global gridded annual land-use transitions, wood harvest, and resulting secondary lands, Climatic Change, 109, 117-161, 2011.

Körner, C., 2006. Plant CO2 responses: an issue of definition, time and resource supply. New Phytol. 172, 393–411. Pugh, T.A.M., Müller, C., Arneth, A., Haverd, V., Smith, B., 2016. Key knowledge and data gaps in modelling the influence of CO2 concentration on the terrestrial carbon sink. J. Plant Physiol. 203, 3–15. doi:10.1016/j.jplph.2016.05.001

Lindeskog, M., Arneth, A., Bondeau, A., Waha, K., Seaquist, J., Olin, S., and Smith, B.: Implications of accounting for land use in simulations of ecosystem carbon cycling in Africa, Earth Syst. Dynam., 4, 385-407, doi:10.5194/esd-4-385-2013, 2013.

Luyssaert, S., I. Inglima, and M. Jung. 2009. Global Forest Ecosystem Structure and Function Data for Carbon Balance Research. Global Forest Ecosystem Structure and Function Data for Carbon Balance Research. Data set. Available on-line [http://daac.ornl.gov/] from Oak Ridge National Laboratory Distributed Active Archive Center, Oak Ridge, Tennessee, U.S.A. doi:10.3334/ORNLDAAC/949.

Michaletz, S. T., D. Cheng, A. J. Kerkhoff, and B. J. Enquist. 2014. Convergence of terrestrial plant production across global climate gradients. Nature 512:39-43

Pugh, T.A.M., Müller, C., Arneth, A., Haverd, V., Smith, B., 2016. Key knowledge and data gaps in modelling the influence of CO2 concentration on the terrestrial carbon sink. J. Plant Physiol. 203, 3–15. doi:10.1016/j.jplph.2016.05.001

Reich, P. F., and H. Eswaran. 2002. Global resources. In: Lal, R. (ed.). Encyclopedia of Soil Science, pp. 607-611. Marcel Dekker, New York.

Rosenzweig, C., Elliott, J., Deryng, D., Ruane, A.C., Müller, C., Arneth, A., Boote, K.J., Folberth, C., Glotter, M., Khabarov, N., Neumann, K., Piontek, F., Pugh, T. a M., Schmid, E., Stehfest, E., Yang, H., Jones, J.W., 2014. Assessing agricultural risks of climate change in the 21st century in a global gridded crop model intercomparison. Proc. Natl. Acad. Sci. U. S. A. 111, 3268–73. doi:10.1073/pnas.1222463110

Sallaba, F., D. Lehsten, J. Seaquist, and M. T. Sykes. 2015. A rapid NPP meta-model for current and future climate and CO2 scenarios in Europe. Ecological Modelling 302:29-41.

Smith, W., Cleveland, C.C., Reed, S.C. & Running, S.W. (2014a). Agricultural conversion without external water and nutrient inputs reduces terrestrial vegetation productivity. Geophys. Res. Lett., 41, 449–455.

Smith, B., Wårlind, D., Arneth, a., Hickler, T., Leadley, P., Siltberg, J., Zaehle, S., 2014b. Implications of incorporating N cycling and N limitations on primary production in an individual-based dynamic vegetation model. Biogeosciences 11, 2027–2054. doi:10.5194/bg-11-2027-2014

Trenberth, K. E., A. Dai, G. van der Schrier, P. D. Jones, J. Barichivich, K. R. Briffa, and J. Sheffield. 2014. Global warming and changes in drought. Nature Clim. Change 4:17-22.

We thank RC2 for challenging us with some thought-provoking critique. What follows is a point-by-point response to these comments, together with a new marked-up version of our manuscript.

**RC2:** This manuscript does NOT satisfies your editorial criteria as described at http://www.earth-system-dynamics.net/peer_review/review_criteria.html This manuscript perhaps intends to make contributions to regional studies of the socioeconomic implications of global change, particularly about the Sahel and its delicate balance between supply and demand of natural resources, with a focus on its implications for food production; however some of its methods are flawed and the use of information weak. This paper deals with very delicate topics that deserve honour and credit, but using the wrong tools to address them, for which the authors deserve no mercy. Therefore I recommend the rejection of this manuscript.

**Authors' Response:** Thanks very much for your general opinion. Please read through our responses below and our responses to the first and third reviewers which we hope will allay some of your concerns.

**RC2:** LPJ and the like models are normally very rough on their predictions, if you simplify them more, then your results might be useless.

**Authors' Response:** We concur that 'LPJ and the like models' can be 'very rough on their predictions,' which is the reason for taking an exploratory approach (rather than a predictive one) in this study. We emphasize a structural analysis of NPP supply-demand outcomes across a range of scenarios, using simplified models that can easily be coupled across sectors. One of the purposes of the manuscript is to demonstrate such a framework. We re-iterate our rationale, taken from p. 3 lines 3-12 of our original manuscript, and p. 3 lines 5-14 of our revised manuscript:

"Developing such tools requires coupling of specific models that address different sectors, such as a model for supply and a model for demand that can be run across multiple future climate, socio-economic and CO2 concentration scenarios. However, the supply-demand system in the Sahel is complex and the future cannot be precisely evaluated. This is because there are many uncertainties associated with the assumptions that underpin the natural and socioeconomic drivers that lead to particular supply-demand balances. As such, an exploratory modelling approach is required, where an emphasis is placed on a structured analysis across a range of outcomes. This approach capitalizes on future indeterminacy for developing adaptive policy insights (e.g. Kwakkel and Pruyt (2013)). As the goal of exploratory frameworks is not prediction, they often employ parsimonious or simplified versions of more complex models (often referred to as meta-models in the latter case) that run across a range of scenarios (e.g. Harrison et al. (2016)). Another benefit of using such simplified models lies in the ease to which they can be coupled to other sectoral models (e.g. Kebede et al. (2015))."

**RC2:** Pg. 4 line 25, you evaluate the performance of your model against another model (LPJ)? Why this is good science deserving publication? This is bad science. Have you thought about doing it against data?

**Authors' Response:** The rationale for evaluating the performance of BME against LPJ-GUESS is to verify that BME (a meta-model based on LPJ-GUESS) captures the magnitude, interannual variation and trends in LPJ-GUESS across the historical climate record. In accordance with specific requests by

Thomas Pugh (RC1) and RC3, we have also compared our LPJ-GUESS and BME NPP with MOD-17 (absolute values, interannual variability, and trends) for the years 2000-2006, as well as against trends in crop yields found in the literature. We have also performed a validation of LPJ-GUESS and BME for all biomes used to develop BME. Please see our responses to R#1 and AR#3 for details, and our revised Appendix A2 (pp. 24-33).

**RC2:** You do a regional level study using GCM data? Not good practice. See what other Swedish colleagues do with regional data there: http://www.smhi.se/en/research/research-departments/climate-research-rossbycentre2-552/an-ensemble-of-cordex-africa-climate-projections-simulated-by-rca4- 1.25312

**Authors' Response:** PLUM is a global scale model (see Engström et al. 2016a, 2016b) that links all countries via international trade to help regulate the balance of feed and food. The implication is that the supply and demand generated in any one country or world region (e.g. the greater Sahel) is a function of supply-demand dynamics across the globe. GCM output is therefore consistent with the level of organization at which PLUM operates (global) and requires global level climate projections. Spatial resolution may certainly be a factor but as the Sahel does not exhibit large topographical variation we hypothesize that the effect of downscaling will not be large. Indeed, Blanke et al. (2016) also conclude that there is no large gain in LPJ-GUESS simulations of C and N stocks when using regionally downscaled, bias corrected climate products compared to GCM simulations, at least for Europe.

**RC2:** What you intend to argue, deriving insights from NPP into food production related arguments, is very weak in methodological terms, and although your rationale and arguments are sensible, the methods you use disqualify the support you use for the argumentation. Then you use a convenient "technology improvement factor"? and close the yield gap with it? I am sorry, again, this is bad science, and it should not be published.

**Authors' Response:** Please see our responses to your previous comments, as well as our responses to the other two reviewers that deal with various methodological issues. We now clarify on p. 5, lines 17-22:

"The technology improvement factor is the aggregate result of parameterizing three technology related parameters (trends in technology, change in yield with GDP per capita, as well as how agricultural management practices are transferred both within and between countries) that are consistent with the scenario storyline of each SSP. Parameter ranges have been empirically determined based on analysis of data between the years 1995 and 2005. Yield gaps are not necessarily closed, but are decreased (see Engström et al. 2016b for more detail)."

**References in our responses**

Blanke, J.H., Lindeskog, M., Lindström, J., and Lehsten, V. Effect of climate data on simulated carbonand nitrogen balances for Europe, Journal of Geophysical Research: Biogeosciences, 121, 1352-1371, 2016.

Engström, K., Olin, S., Rounsevell, M. D. A., Brogaard, S., van Vuuren, D. P., Alexander, P., Murray-Rust, D., and Arneth, A.: Assessing uncertainties in global cropland futures using a conditional probabilistic modelling framework, Earth System Dynamics, 7, 893–915, 2016b.

Engström, K., Rounsevell, M. D. A., Murray-Rust, D., Hardacre, C., Alexander, P., Cui, X. F., Palmer, P. I., and Arneth, A.: Applying Occam's razor to global agricultural land use change, Environmental Modelling & Software, 75, 212-229, 2016a.

Harrison, P. A., Dunford, R. W., Holman, I. P., and Rounsevell, M. D. A.: Climate change impact modelling needs to include cross-sectoral interactions, Nature Clim. Change, advance online publication, 2016.

Kebede, A. S., Dunford, R., Mokrech, M., Audsley, E., Harrison, P. A., Holman, I. P., Nicholls, R. J., Rickebusch, S., Rounsevell, M. D. A., Sabaté, S., Sallaba, F., Sanchez, A., Savin, C., Trnka, M., and Wimmer, F.: Direct and indirect impacts of climate and socio-economic change in Europe: a sensitivity analysis for key land- and water-based sectors, Climatic Change, 128, 261-277, 2015.

Kwakkel, J. H. and Pruyt, E.: Exploratory Modeling and Analysis, an approach for model-based foresight under deep uncertainty, Technological Forecasting and Social Change, 80, 419-431, 2013.

We also thank RC3 for their insightful and constructive comments. What follows is a point-by-point treatment of these comments. We respond to each comment in turn, while at the same time provide a marked up manuscript with changes.

**RC3:** 1. The introduction is very well-written and does an excellent job of framing the question and establishing the importance of the work.

**Authors' Response:** Thank-you very much. It is good to know that the effort we made crafting the introduction does not go unnoticed.

**RC3:** 2. Page 4, line 27: It is stated that the authors used 0.5 degree climate data from five GCMs, and [CO2] based on four RCPs (Representative Concentration Pathways). The way it is phased it is unclear which RCPs were used to generate the climate projections for each GCM. The authors should have used climate data derived from runs across the 4 RCPs for each of the 5 models. Please clarify the text if this is the case. If not, please explain more fully why the climate data were not derived for all RCPs.

**Authors' Response:** This section in our revised manuscript now reads (p. 5, lines 3-6):

"We forced BME with climate data (spatial resolution 0.5 x 0.5 degrees) from five GCMs (General Circulation Models, including HADLEY, GFDL, IPSL, MIROC and NorESM), and [$CO_2$] based on four RCPs (Representative Concentration Pathways, including 2.6, 4.5, 6.0 and 8.5) to estimate annual total NPP in kg dry-weight $m^{-2}$ $yr^{-1}$. (DW, dry-weight). We used climate data derived from runs across the 4 RCPs for each of the 5 models."

**RC 3**: Robustly representing future NPP trajectories is challenging due to the many potential counteracting feedbacks. The authors show a good fit with LPJ NPP simulations, but do not consider observational data or alternative runs of the LPJ model itself. I recommend further comparison against both census derived yield trends (Rey et al. 2013) and satellite-derived yield trends (Running et al. 2004). For instance, the authors could consider runs in which the CO2 fertilization effect is turned off. Currently all the NPP trends considered in the paper are increasing due to CO2 fertilization (page 9, line 24). This is an area of debate and may be counter to observational data (see Smith et al. 2016, Oberneier et al. 2016, and Ort & Long et al. 2014). Thus, I wonder if a scenario in which CO2 fertilization effects are isolated and removed would be a more realistic lower boundary on what to expect for the region? I would imagine very large increases in the NPP debt (without large irrigation efforts), much larger than what is currently considered in the paper.

**Authors' Response:** We thank this reviewer for underscoring these issues with the trends, and for suggesting some ways forward. In order to address these comments, we compare BME simulations with MODIS data for the period 2000-2006 (see new section, Appendix 2.2, p. 31, lines 7-18):

"We also compare total yearly means of NPP from BME and LPJ-GUESS to NPP derived from the MOD17A3 processing stream (using MOD17A3 data obtained from the NASA Earth Observation System repository at the University of Montana at www.ntsg.umt.edu) for the period 2000 to 2006 for the greater Sahel region (Running, 2004). We averaged resampled MODIS NPP from 1km to the spatial resolution of the BME estimates (0.5 x 0.5 degrees) and excluded urban areas. We removed below-ground NPP and plant parts unable to be consumed by applying the same R:S and harvest index as described in Section 2.1.1. Lastly, we calculated mean values of MODIS NPP estimates from

[revised manuscript text omitted]

**RC3:** 4. Page 5, line 4: When the fractional agricultural landcover estimates from Hurtt et al (2011) were applied, was it assumed that natural and agricultural NPP were similar? If so, this assumption should be revisited after considering differences between agricultural vs. natural NPP for the region. For instance, the authors could compare census based estimates of crop productivity with their estimates as a reality check. Smith et al. 2014 (see reference below), found that agricultural

productivity for the region is significantly lower than natural productivity. If this potential reality is not considered, then the scenarios in this manuscript may be overly optimistic.

**Authors' Response:** Yes, it is true that we considered the NPP to be equal for all land covers. However, by using a relatively low (0.235) harvest index, we have implicitly accounted for at least some of that lower productivity.

In several sections, we have now pointed out that that our estimates of NPP are optimistic (eg. p. 17, lines 29-31):

"All of this suggests that the NPP increases found in our current analysis are likely optimistic, due the potential overestimation of the $CO_2$ fertilization effect, as well as the fact that BME is based on potential natural vegetation. "

Appendix A,  p. 31 lines 2-6:

"The implication of this experiment is that there is a demonstrable reduction in NPP when land management is taken into consideration, but the effect is relatively minor. Lindeskog et al. (2013) show that LPJ-GUESS managed land (C-version) overestimated actual yield derived from FAO country-level crop statistics and Smith et al. (2014b) also report that natural systems are more productive than agricultural systems in sub-Saharan Africa. We conclude with that possibility that our results are in the upper range for NPP found in the Sahel."

Appendix A,  p. 32, lines 7-9:

"We therefore conclude that BME and LPJ-GUESS replicate ground observations of NPP at similar orders of magnitude at the biome level, but may be overestimated due to the fact that natural systems are usually more productive than agricultural ones."

**RC3:** 5. I would recommend revisiting all crop allocation parameters based on those reported by Monfreda et al. (2008). Given the high variability in crop specific harvest fractions, it seems it may be necessary to parameterize the model for each individual crop grown in the region.

**Authors' Response:** It is true that the representation of crop allometry in the current setup is simplistic and cannot likely capture all the variability present in agricultural landscapes in the region. But the approach taken here, by having one root-to-shoot ratio and harvest index for all crops in the region is consistent with the underlying theme of the study, e.g.  have a simplistic modelling framework to be able to explore supply-demand outcomes with a minimum of input data. Furthermore, as we do not know how these cropping systems would develop in the future across the scenarios, we think a simple approach is the safest bet.

**RC3:** 6.  Page 12 line 24-26: This statement is not representative of the literature (see below references). I would suggest more nuanced discussion of the potential limitations of the supply approach used in this analysis. For instances, how much did CO2 fertilization drive increases? How uncertain are the precipitation estimates? Were nutrient constraints considered and if so what are the management implications? If not, how might nutrient constraints limit NPP? How will increases in atmospheric water demand (Vapor pressure deficit) affect yields and productivity? Could increased drought and desertification also represent a potential scenario had the CO2 sensitivity been

adjusted? The way that this section is currently written is a gross over extension of the simplified NPP modeling that the paper is based on.

**Authors' Response**: Thank-you very much for directing us to the salient literature. In response to this, we have removed these statements from Section 4.4 (Additional Perspectives) and treat these issues in an expanded Section 4.6 (Uncertainties). Please see p. 17, lines 13-31:

"Additional uncertainty exists with respect to the total magnitude and trends of simulated NPP supply, given the lack of ground truth for the region, and that differences in NPP trends between other models is very large (e.g. Friend et al., 2014; Körner et al., 2006; Pugh et al., 2016; Rosenzweig et al., 2014). Indeed, recent observational evidence suggests that the effect of $CO_2$ fertilization on plant growth may be constrained by counteracting feedbacks associated with increasing atmospheric moisture demand and nutrient availability (e.g. Smith et al., 2016; Wieder et al. 2015). For example, NPP is reduced under warmer and dryer conditions due to moisture stress, particularly in temperate and arid ecosystems. Future trends NPP trends in the Sahel could therefore be strongly determined by changes in the frequencies of wet years versus dry years, with the dry years counteracting the $CO_2$ fertilization effect. Furthermore, nutrient supply rates may not be able to keep up with extra demand associated with $CO_2$ fertilization, and leading to a depletion of soil nutrients, as current evidence suggests. This could also curtail the $CO_2$ fertilization effect, particularly in the more southerly parts of our study area, where nutrients tend to become a limiting factor. We performed a simple experiment negating the $CO_2$ fertilization effect in order to gauge its impact on supply-demand balance on all scenarios. For the SSP2-RCP6.0, per capita demand has an equal chance of exceeding per capita supply in 2036 for the SSP2-6.0 scenario as opposed to 2043 if $CO_2$ fertilization in included (Fig. B7), with a very high likelihood of continuous supply shortfall beginning in 2056, as opposed to 2073 with $CO_2$ fertilization. The effect on all other scenarios is an earlier shift to the onset of supply shortfalls, by about 10 years, compared to Fig. 3b (see Fig. B7). Supply shortfalls with high likelihood of occurrence (black dots showing non-overlapping 95% confidence intervals) are similarly shifted, and occur with greater consistency and frequency. All of this suggests that the NPP increases found in our current analysis are likely optimistic, due the potential overestimation of the $CO_2$ fertilization effect, as well as the fact that BME is based on potential natural vegetation."

**RC3:** Minor Comments: 1. Page 3, Line 17: "Three different aggregation levels are considered, including Sahel, the country, and the local". Please define what is meant by local level. Pixel level? What resolution? 2. Page 12 line 12: missing end of parentheses.

**Authors' Response:** We clarify this on p. 3, lines 19-20 of our marked up manuscript:

"Three different aggregation levels are considered, including Sahel, the country, and the local (cell level with a spatial resolution of $0.5^{o}$ x $0.5^{o}$)."

We have also fixed the typos.

**References in our responses**

Brandt, M., Mbow, C., Diouf, A. A., Verger, A., Samimi, C., and Fensholt, R.: Ground- and satellite-based evidence of the biophysical mechanisms behind the greening Sahel, Global Change Biology, 21, 1610-1620, 2015.

Eklundh, L. and Olsson, L.: Vegetation index trends for the African Sahel 1982–1999, Geophysical Research Letters, 30, 8, 1430, 2003.

Herrmann, S. M., Sall, I., and Sy, O.: People and pixels in the Sahel: a study linking coarse-resolution remote sensing observations to land users' perceptions of their changing environment in Senegal, Ecology and Society, 19, 2014.

Hickler, T., Eklundh, L., Seaquist, J. W., Smith, B., Ardo, J., Olsson, L., Sykes, M. T., and Sjostrom, M.: Precipitation controls Sahel greening trend, Geophysical Research Letters, 32, L21415, 2005.

Obermeier, W.A., Lehnert, L.W., Kammann, C.I., Müller, C., Grünhage, L., Luterbacher, J., Erbs, M., Moser, G., Seibert, R., Yuan, N. & Bendix, J. (2016). Reduced $CO_2$ fertilization effect in temperate C3 grasslands under more extreme weather conditions. Nat. Clim. Chang., 7, 137–142.

[revised manuscript text omitted]

---

## Author Response (AR2)

Response to Reviewer:

Thanks very much for your final recommendation. We are very happy that the manuscript is accepted as is.

We have, however, made two changes:

1. Page 1, Line 5, J.W. Seaquist was changed to Jonathan W. Seaquist
2. Page 39, the image of Table C2 in the original manuscript has been replaced with a table in word.

[revised manuscript text omitted]